# Atomically precise copper clusters with dual sites for highly chemoselective and efficient hydroboration

Teng Jia[1], Jie Ai[1], Xiaoguang Li[2], Miao-Miao Zhang[1,3], Yue Hua[1], Yi-Xin Li[1], Cai-Fang Sun[1], Feng Liu[4], Ren-Wu Huang[1] ✉, Zheng Wang[2] ✉ & Shuang-Quan Zang[1] ✉

The hydroboration of alkynes into vinylboronate esters is a vital transformation, but achieving high chemoselectivity of targeted functional groups and an appreciable turnover number is a considerable challenge. Herein, we develop two dynamically regulating dual-catalytic-site copper clusters ($Cu_4NC$ and $Cu_8NC$) bearing N-heterocyclic thione ligands that endow $Cu_4NC$ and $Cu_8NC$ catalysts with performance. In particular, the performance of microcrystalline $Cu_4NC$ in hydroboration is characterized by a high turnover number (77786), a high chemoselectivity, high recovery and reusability under mild conditions. Mechanistic studies and density functional theory calculations reveal that, compared with the $Cu_8NC$ catalyst, the $Cu_4NC$ catalyst has a lower activation energy for hydroboration, accounting for its high catalytic activity. This work reveals that precisely constructed cluster catalysts with dual catalytic sites may provide a way to substantially improve catalytic properties by fully leveraging synergistic interactions and dynamic ligand effects, thus promoting the development of cluster catalysts.

Vinylboron compounds are potential multifunctional chemical targets that can be used to directly construct various complex organic molecules in a streamlined and efficient manner and are thus highly important compounds in various areas of chemistry[1-4], particularly in materials science, pharmaceuticals and organic synthesis, where these compounds have been used in the Suzuki−Miyaura cross−coupling[5], boronic acid Mannich (BAM)[6], Chan−Lam coupling[7] and Hayashi−Miyaura[8] reactions. Various approaches for synthesizing vinylboron compounds via the hydroboration of alkynes have been investigated. To achieve regioselective and stereoselective synthesis of vinylboronate esters, diverse transition metal catalysts, particularly copper complexes, have been utilized to produce organic boron building blocks, but the turnover number (TON) for hydroboration is usually <5000 under specific conditions, such as dry and inert atmospheres

(Fig. 1a)[9-13]. In addition, the lack of specificity in terms of the functional group reactivity during hydroboration is also a challenge. These challenges occur because the existing transition metal complexes contain a single catalytic center that simultaneously activates alkyne substrates and boron reagents to afford boron compounds, thus greatly limiting the catalytic activity of the transition metal complexes. Dual-catalytic-site catalysts, which can exhibit enhanced catalytic activity through exposed atomic-scale interfaces and the synergistic effect between two contiguous catalytic sites, have been widely applied to various types of catalysis[14-18]. The use of dual-catalytic-site catalysts is a good strategy for overcoming the challenges associated with hydroboration. For example, Li[19] reported that metal−organic framework nanosheets, including dual-catalytic-site complexes containing Cu and N, efficiently promoted hydroboration with a high TON,

[1]Henan Key Laboratory of Crystalline Molecular Functional Materials and College of Chemistry, Zhengzhou University, Zhengzhou 450001, China. [2]Institute for Advanced Study, Shenzhen University, Shenzhen 518060, China. [3]School of Chemistry and Chemical Engineering, Henan University of Technology, Zhengzhou 450001, China. [4]Yunnan Precious Metals Lab Co., LTD, Kunming 650106, China. ✉e-mail: renwuhuang@zzu.edu.cn; chzwang@szu.edu.cn; zangsqzg@zzu.edu.cn

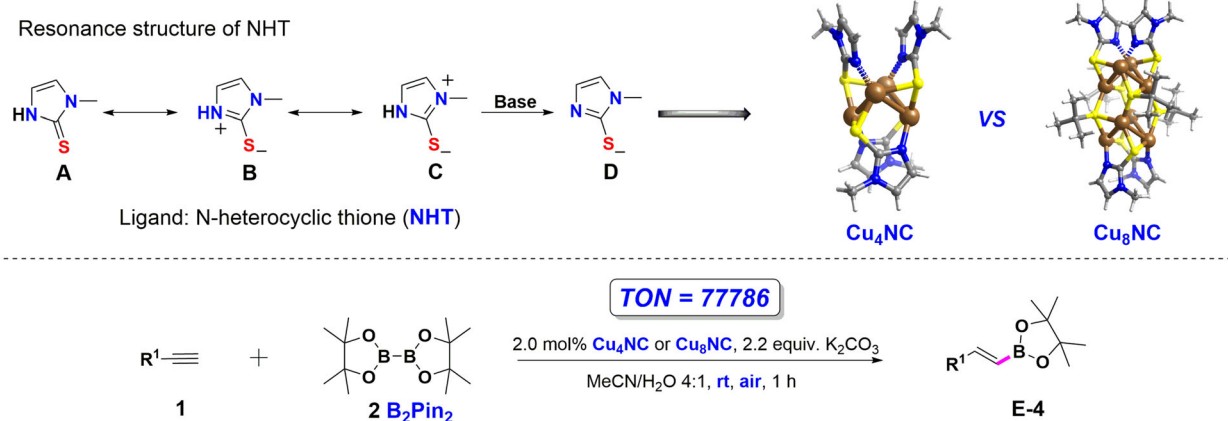

**a** The transition-metal as homogeneous catalysts catalyzed the hydroboration of alkynes.

- Low recovery ratio and low catalytic activity (low TON) for homogeneous catalyst
- Required specific conditions (such as a dry and inert atmosphere, high temperature, etc. )

[M] = Rh(I), Ir(III), Cu(I/II), Fe(II), Co(II), Mg(II) and Ti(IV) salts, etc.

**b This work**: The $Cu_4NC$ and the $Cu_8NC$ as heterogeneous catalysts catalyzed the hydroboration of alkynes.

Resonance structure of NHT

Ligand: N-heterocyclic thione (**NHT**)

$Cu_4NC$ VS $Cu_8NC$

**TON = 77786**

2.0 mol% **$Cu_4NC$ or $Cu_8NC$**, 2.2 equiv. $K_2CO_3$

MeCN/$H_2O$ 4:1, **rt**, **air**, 1 h

- *N*-heterocyclic thione-stabilized copper nanoclusters
- High turnover number of $Cu_4NC$ (77786)
- 28 examples of hydroboration, up to 99% yield
- Excellent regioselectivies, stereoselectivies and chemoselectivies
- C-B coupling under air atmosphere at room temperature
- High recovery ratio for microcrystal $Cu_4NC$ and $Cu_8NC$ catalysts

**Fig. 1 | Representative synthesis strategies for vinylboronate esters. a** Transition metal-catalyzed hydroboration of alkynes. **b** This work: microcrystalline $Cu_4NC$ and $Cu_8NC$, as heterogeneous catalysts, catalyzed the hydroboration of alkynes. $R_1$ represents the type of functional group.

further illustrating that synergistic interactions can tune the catalytic properties. However, the construction of precisely regulated dual-catalytic-site catalysts that fully leverage these synergistic effects to achieve improved catalytic properties remains challenging.

Metal nanoclusters (NCs) containing multiple metal atoms, which have been rapidly developed in the field of chemical catalysis[20–23], are important materials for designing and developing efficient dual-catalytic-site catalysts. Owing to the good chemical purity, well-defined crystal structures, high surface-to-volume ratios, and homogeneous distribution of catalytic sites of such NCs, their targeted modification is convenient and is beneficial for establishing a solid correlation between their structure and performance[24–28]. Compared with traditional dual-catalytic-site catalysts, NCs with dual catalytic sites have the following advantages: (I) Cluster kernels are constructed through metal-metal bonds between different metal atoms, which is beneficial for regulating the catalytic activity[21,24,28]. (II) Dual-catalytic-site NCs protected by dynamic ligands can prevent catalyst passivation and deactivation to maintain their catalytic activity, thus helping to increase the substrate tolerance. (III) Dissociated dynamic ligands are convenient for regulating the chemical environment around dual catalytic sites, laying a solid foundation for improving the selectivity of catalysts. Nevertheless, in previous catalytic studies, the ligands of NCs were usually removed to expose more catalytic sites, leading to changes in the cluster structures and aggregation of clusters, which decreased the structural uniformity. If the dynamic coordination properties of the cluster shell ligands can be effectively utilized to achieve precise dual-catalytic-site control while maintaining the integral cluster structure, then ligand-protected NCs should be the most ideal catalysts. The stability of copper NCs is worse than that of gold NCs and silver NCs, which has caused their development to be slower than that of other materials. Improving the stability of copper NCs

remains challenging. The versatile ligands of N-heterocyclic thiones (NHTs) with zwitterionic resonance structures (Fig. 1b) have been widely applied to various transition metal nanoparticles and homogeneous catalysts to catalyze various kinds of chemical reactions[29–33]. Owing to the ease with which the steric and electronic effects of NHTs can be tuned, the good stability of NHTs and the formation of stable Cu-S bonds between NHT ligands and copper atoms, NHT ligands are ligand stabilizers for copper NC catalysts that can improve the catalytic activity via ligand engineering strategies[32–35]. To date, copper NCs with dual catalytic sites as heterogeneous catalysts have not been reported to catalyze the highly efficient and highly chemoselective hydroboration of alkynes under mild conditions.

Therefore, we designed the bidentate NHT ligand methimazole as a shell ligand for copper NCs. Dynamically regulating dual-catalytic-site (DRDS) copper clusters $[Cu_4(NHT)_4]$ ($Cu_4NC$) and $[Cu_8(NHT)_4(^tBuS)_4]$ ($Cu_8NC$) were efficiently synthesized in the gram scale. The bidentate NHT ligand participates in two different types of bonding modes in the copper clusters, i.e., a reversible Cu-N bond and a stable Cu-S bond between the shell ligands and copper cluster cores, which are beneficial for improving the stability and maintaining the catalytic activity of the clusters. As expected, the microcrystalline $Cu_4NC$ catalyst provided a solid foundation for achieving high efficiency (turnover number (TON) = 77786) (Supplementary Table 1), and regio-, stereo- and chemoselectivity in hydroboration at room temperature in an air atmosphere (Fig. 1b). Microcrystalline $Cu_4NC$ and $Cu_8NC$ catalysts were also recovered and reused. Through single-crystal structure analysis, control experiments, in situ characterization and density functional theory (DFT) calculations, the dual catalytic sites of $Cu_4NC$ and $Cu_8NC$ were further characterized. The hydroboration mechanisms were well elucidated, effectively presenting the relationships between the cluster structure and performance.

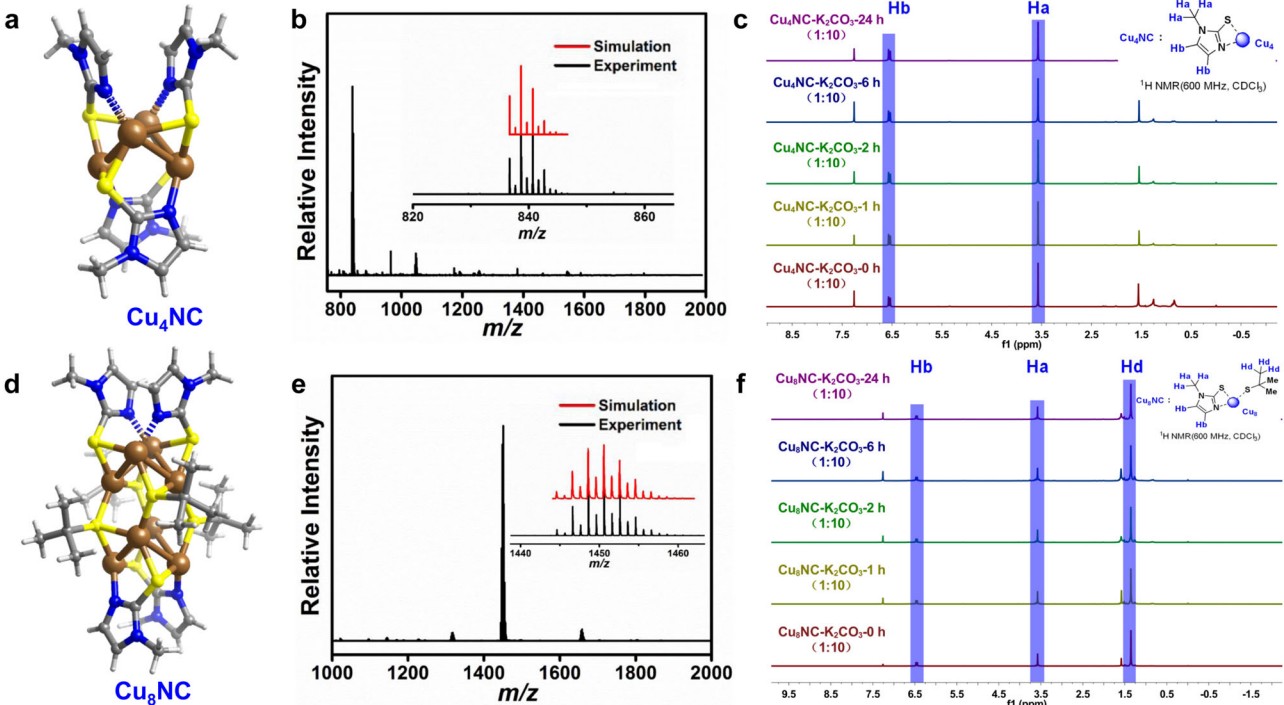

**Fig. 2 | Schematic illustration and characterization of Cu₄NC and Cu₈NC.**
**a** Structure of Cu₄NC. Color legend: brown, Cu; blue, N; yellow, S; gray, C; and white, H. **b** ESI–MS spectra of Cu₄NC dissolved in dimethyl sulfoxide (DMSO) solution and measured in positive ion mode. Inset: experimental (black) and simulated (red) isotopic distributions of Cu₄NC. **c** ¹H NMR spectra of Cu₄NC-K₂CO₃ (1:10) in CDCl₃ (0–24 h). Ha and Hb represent the characteristic H atoms of Cu₄NC.

**d** Structure of Cu₈NC. Color legend: brown, Cu; blue, N; yellow, S; gray, C; and white, H. **e** ESI–MS spectra of Cu₈NC dissolved in DMSO solution and measured in positive ion mode. Inset: experimental (black) and simulated (red) isotopic distributions of Cu₈NC. **f** ¹H NMR spectra of Cu₈NC-K₂CO₃ (1:10) in CDCl₃ (0–24 h). Ha, Hb and Hd represent the characteristic H atoms of Cu₈NC.

## Results

### Catalyst development and characterization

NHT ligands are classic ligand stabilizers and are air- and moisture-stable molecules. The resonance structures of NHT ligands can be drawn as zwitterions, with a negative charge located on the sulfur atom, which indicates that the sulfur anion has a higher associated electron density than a standard thiocarbonyl. The constructed conditions for Cu₄NC and Cu₈NC are under basic conditions. When NHT ligands are under basic conditions, they can undergo tautomerization, in which the thione form structure changes to an enol-like form structure (Fig. 1b, compound D), which implies that the electron density of the sulfur anion is further increased, endowing it with strong interactions with copper. The donation of a pair of electrons by the S anion, as an electron donor, to electron acceptors of copper cluster cores is beneficial for the construction of stable Cu-S bonds. The methimazole molecule, a classic NHT ligand, was designed as a model bidentate ligand with two different coordination modes: flexible and reversible Cu-N bonds and stable Cu-S bonds. The Cu₄NC catalyst was synthesized via the reaction of bidentate NHT ligands and the copper salt [Cu(MeCN)₄]PF₆ under basic conditions through a one-pot method. Similarly, Cu₈NC could be obtained by using ᵗBuSCu as the copper source instead of [Cu(MeCN)₄]PF₆ under basic conditions. Details of the synthesis are presented in the Supplementary Information. Compared with the Cu-N bonds, the Cu-S bonds were more stable. According to DFT calculations of the Cu-S and Cu-N bonds in Cu₄NC and Cu₈NC, the binding energies of the Cu-S bonds (−20.7 kcal/mol and −24.2 kcal/mol, respectively). were lower than the binding energies of Cu-N bonds (−15.0 kcal/mol and −17.6 kcal/mol, respectively) (Supplementary Fig. 71). This result indicates that the Cu-S bond is stronger than the Cu-N bond. Our findings align with previously reported results in the literature[25], which also demonstrated that the Cu-N bond is weaker than the Cu-S bond. Single-crystal X-ray

diffraction (SCXRD) results demonstrated that Cu₄NC was composed of a Cu₄ metal core bridged by four NHT ligands. The NHT ligands contained S and N donors, which could be firmly anchored on the surface of the Cu₄NC cluster via Cu-S bonds and Cu-N bonds, further improving the rigidity and stability of Cu₄NC (Fig. 2a, Supplementary Fig. 5). In the Cu₄ unit, an average Cu-Cu distance of 2.662 Å was observed, which was less than twice the van der Waals radius (2 × 1.40 Å), indicating the occurrence of cuprophilic interactions[36,37]. Structurally, Cu₈NC was found to contain a Cu₈ metal core, four NHT ligands and four ᵗBuS- ligands (Fig. 2d, Supplementary Fig. 6). The skeleton of Cu₈NC consisted of two identical Cu₄ units, which were fused through four thiolate ligands, similar to a cage configuration. The Cu-Cu bond lengths ranged from 2.663 to 2.710 Å, suggesting the occurrence of metal-metal interactions. Furthermore, compared with the overall metal skeleton of Cu₄NC, the thiolate ligands changed the inner structure from a Cu₄ unit to two identical Cu₄ units linked together by Cu-S interactions. SCXRD analysis revealed that the Cu-Cu distances were 2.662 Å and 2.663 Å in the Cu₄NC unit and Cu₈NC unit, respectively, which were slightly shorter than that of the tert-butoxide dicopper species[15]. This shorter distance is beneficial for the formation of key intermediates A and B to achieve synergistic effects on dual catalytic sites.

The research revealed that microcrystalline Cu₄NC and Cu₈NC have moderate solubilities in DCM, CHCl₃ and DMSO solutions, respectively. To confirm the precise molecular masses and formulas of the copper NCs, the clusters were further characterized by electrospray ionization mass spectrometry (ESI–MS) in positive ion mode in DMSO solution (Fig. 2b and e, Supplementary Figs. 9 and 10). The high-abundance peak at m/z 838.7003, can be assigned to [Cu₄NC + Cs]⁺ (calcd m/z 838.6913) (Fig. 2b, Supplementary Fig. 9). The mass spectrum of the Cu₈NC showed an intense peak at 1450.5869, corresponding to the molecular ion [Cu₈NC + Cs]⁺ (calcd m/z 1450.5766)

**Table 1 | Optimization of hydroboration conditions[a,b]**

| Entry | Variations | Yield of E-4a |
|---|---|---|
| 1 | None | 98% |
| 2 | 30 min. | 87% |
| 3 | $N_2$, 30 min. | 86% |
| 4 | $N_2$ | 97% |
| 5 | Dark | 95% |
| 6 | $Cu_8NC$ | 64% |
| 7 | $Cu(MeCN)_4PF_6$ | 43% |
| 8 | 4.0 mol% Cu powder | 27% |
| 9 | HBpin instead of $B_2Pin_2$ | n.r |
| 10 | $Cs_2CO_3$ instead of $K_2CO_3$ | 77% |
| 11 | $Na_2CO_3$ instead of $K_2CO_3$ | 70% |
| 12 | THF as solvent | trace |
| 13 | Dioxane as solvent | trace |
| 14 | EtOH as solvent | 47% |
| 15 | DMF as solvent | trace |
| 16 | MeCN as solvent | trace |
| 17 | DCM/$H_2O$ as solvent | n.r |
| 18 | $CHCl_3$/$H_2O$ as solvent | n.r |
| 19 | 5.0 mol% NHT ligand | n.r |
| 20 | No $K_2CO_3$ | n.r |
| 21 | No copper | n.r |

[a]General reaction conditions: Phenylacetylene **1** (0.2 mmol, 1.0 equiv.), $B_2Pin_2$ **2** (0.44 mmol, 2.2 equiv.), microcrystalline $Cu_4NC$ catal. (0.004 mmol, 2.0 mol%) and $K_2CO_3$ (0.44 mmol, 2.2 equiv.) were added to the mixture solvent (2.0 mL, MeCN/$H_2O$ 4:1) under air atmosphere at room temperature and allowed to react for 1 h. [b]The reported yield was determined by [1]H NMR using 1,3,5-trimethoxybenzene as an internal standard.

(Fig. 2e, Supplementary Fig. 10). The experimental isotopic distribution patterns of these copper NCs were in good agreement with the simulated results. The phase purities of $Cu_4NC$ and $Cu_8NC$ were demonstrated by elemental analysis, powder X-ray diffraction (PXRD) and [1]H nuclear magnetic resonance (NMR) spectroscopy (Supplementary Figs. 7, 8, 13–15, 18–20). Energy-dispersive spectrometry (EDS) revealed the presence of the expected elements in microcrystalline $Cu_4NC$ and $Cu_8NC$ (Supplementary Figs. 11 and 12). The time-dependent UV−vis absorption spectra of these clusters remained unchanged in dichloromethane (DCM) for 72 h of treatment, which confirmed the good stability of the clusters in organic solution (Supplementary Figs. 25 and 26). Furthermore, microcrystalline $Cu_4NC$ and $Cu_8NC$ were also characterized by time-dependent NMR measurements (Fig. 2c and f, Supplementary Figs. 27–30). The [1]H NMR spectra of $Cu_4NC$ and $Cu_8NC$ did not shift for 48 h or 24 h according to the in situ [1]H NMR measurements of the copper NCs in $CDCl_3$ or alkaline $CDCl_3$, illustrating that these copper NCs were stable in neutral and alkaline organic systems (Fig. 2c and f, Supplementary Figs. 27–30). A series of characterizations of $Cu_4NC$ and $Cu_8NC$ fully illustrated that these copper NCs could be efficiently constructed and display good stability, validating the feasibility of using NHT as a bidentate protective agent to stabilize copper NCs.

## Development of the hydroboration of alkynes
To further explore the relationships between the copper NC structures and properties, microcrystalline $Cu_4NC$ and $Cu_8NC$ were used as catalysts to catalyze the hydroboration of alkynes. Initially, under an air atmosphere, the model substrate phenylacetylene (**1a**) (0.2 mmol, 1.0 eq), the microcrystalline $Cu_4NC$ catalyst (2.0 mol%), $K_2CO_3$ (0.44 mmol, 2.2 equiv.) and $B_2Pin_2$ (0.44 mmol, 2.2 eq) were added to a vial to react in a mixture solvent (MeCN/$H_2O$) at room temperature for 30 min, which provided single vinylboron products with a good yield of 87% (Table 1, entry 2). It showed that microcrystalline $Cu_4NC$ can achieve high regio- and stereoselectivity in catalyzing hydroboration. A control experiment was carried out under a $N_2$ atmosphere by substituting a Schlenk tube for the vial to maintain the airtightness of the catalytic system (Table 1, entry 3). The yield of and selectivity for vinylboron products were not obviously different between the control and experimental groups, illustrating that the microcrystalline $Cu_4NC$ catalytic system had the characteristics of high efficiency, high selectivity and strong tolerance to air and protic solvents during hydroboration. In addition, owing to the complete insolubility of microcrystalline $Cu_4NC$ in MeCN and $H_2O$ solutions, microcrystalline $Cu_4NC$ was considered to be a heterogeneous catalyst for hydroboration in the mixture solvent (MeCN/$H_2O$) (Supplementary Figs. 16 and 17). Previously, most metal catalysts used to catalyze the hydroboration of alkynes with $B_2Pin_2$ were applied under an inert atmosphere[12,19,38–50]. Copper NCs used as heterogeneous catalysts, for highly efficient, highly regio- and stereoselective hydroboration of alkynes under mild conditions, have rarely been reported. To overcome this limitation, more investigations of using copper NCs as catalysts for catalyzing the hydroboration of alkynes were performed to construct a platform that can illustrate the precise relationships between copper NC catalysts and their catalytic properties.

To determine the optimal conditions, focused screening was performed, the results of which are summarized in Table 1. When the reaction time was prolonged to 1 h, vinylboronate ester **E-4a** was obtained under air or $N_2$ atmosphere in a yield of 98% or 97%, respectively (Table 1, entries 1 and 4). When the hydroboration catalyzed by microcrystalline $Cu_4NC$ was carried out in the dark, little difference in the yield and selectivity was observed compared with the results obtained under natural light conditions (Table 1, entries 1 and 5). Since the intervention of light had no impact on the catalytic activity, the results showed that light irradiation was not a necessary condition for this organic transformation. The boron radical species might not be involved during the process supported by this reaction[43,51]. Similar to microcrystalline $Cu_4NC$, $Cu_8NC$ was also completely insoluble under the same reaction conditions (Supplementary Figs. 21 and 22). Microcrystalline $Cu_8NC$ catalyzed the hydroboration of alkyne under the reaction condition described in entry 1, offering a vinylboronate ester product in moderate yield (64%) (Table 1, entry 6). These results further support that microcrystalline $Cu_4NC$ has a higher catalytic activity in hydroboration, which may originate from the lower activation energy for hydroboration of microcrystalline $Cu_4NC$, compared to that of microcrystalline $Cu_8NC$. Notably, when the catalyst loading of microcrystalline $Cu_4NC$ was reduced to $7.0 \times 10^{-4}$ mmol, the customized cluster achieved a TON of up to 77786 (Supplementary Table 1), which was higher than those of catalysts reported for the hydroboration of alkynes (Supplementary Table 5)[38-51].

As observed in the control experiments involving other homogeneous and heterogeneous copper salts, the desired products were obtained in poor yields (43% and 27%) under an air atmosphere at room temperature (Table 1, entries 7 and 8). When HBPin replaced $B_2Pin_2$ as the boron source for hydroboration, no reaction was observed under the optimal conditions (Table 1, entry 9). Different types of bases, such as $Cs_2CO_3$ and $Na_2CO_3$, were applied to this reaction, resulting in decreased yields (Table 1, entries 10 and 11). The influence of the solvents on the catalytic activity of microcrystalline $Cu_4NC$ was also investigated. Some solvents, including single strongly polar solvents, single weakly polar solvents and the mixture solvents, hindered the reaction process, resulting in a serious reduction in the yield or no reaction (Table 1, entries 12–18). Control experiments revealed that hydroboration could not be achieved when the NHT ligand was used as the catalyst, under neutral conditions or in the absence of copper catalysts (Table 1, entries 19–21). These results illustrated that the base is crucial for efficiently catalyzing this reaction under mild conditions and that the hydroboration of alkynes is not a spontaneous reaction under an air atmosphere at room temperature.

After the experiments, the microcrystalline $Cu_4NC$ or $Cu_8NC$ catalysts was filtered, dried and collected. The recycled microcrystalline copper NCs were further characterized via PXRD, UV−vis spectroscopy, $^1H$ NMR spectroscopy and ESI–MS (Supplementary Figs. 31–38), and the results revealed that the characteristic peaks of the copper NCs after catalysis well matched the characteristic peaks of the fresh copper NCs, confirming that the microcrystalline $Cu_4NC$ and $Cu_8NC$ remained intact under air, basic and organic solvents during hydroboration. The filtered supernatant was subsequently characterized via inductively coupled plasma mass spectrometry (ICP-MS), which revealed that no $Cu^+$ ions were found in the reaction solution, suggesting that the $Cu^+$ ions did not leach from the metal cluster catalysts into the solution over the course of the reaction. In addition, three parallel experiments involving hydroboration catalyzed by microcrystalline $Cu_4NC$ were also carried out. The parallel experiments were stopped at the 10th minute, 30th minute and 60th minute. The microcrystalline $Cu_4NC$ catalyst was filtered to recycle it. The recycled microcrystalline $Cu_4NC$ catalyst was further characterized via PXRD (Supplementary Fig. 96). The characteristic peaks of the microcrystalline $Cu_4NC$ after catalysis at different times well matched those of the microcrystalline $Cu_4NC$ before catalysis, confirming that

structure of the recycled catalyst was maintained. These results further confirmed that microcrystalline $Cu_4NC$ and $Cu_8NC$ were stable catalysts.

Under the optimal reaction conditions, the scope of alkyne substrates suitable for hydroboration transformations was investigated, and these results are summarized in Fig. 3. Various alkynes, including aryl and aliphatic alkynes, were considered as coupling partners with $B_2Pin_2$. For aryl alkynes (**1a–1r**), the catalytic system was quite well-tolerated, and performed well for electron-rich and electron-deficient aromatic substrates, resulting in yields (up to 99%). When 4-cyanophenylacetylene (**1p**) and 3-ethynylbenzaldehyde (**1q**) were used as alkyne substrates, vinylboronate esters **E-4p** and **E-4q**, respectively, were obtained as single products, which illustrated that microcrystalline $Cu_4NC$ was a highly regio-, stereo- and chemoselective heterogeneous catalyst capable of achieving highly chemoselective hydroboration between $C \equiv C$ bonds and $C \equiv N$ bonds (or $C = O$ bonds). N-heterocyclic and S-heterocyclic aromatic alkynes also readily afforded single vinylboronate esters (**E-4s–E-4v**) in yields (86%-97%). Furthermore, aromatic alkynes possessing functional groups with high steric hindrance, ferrocene functional groups and diethynyl functional groups were investigated. The results showed that these substrates worked well, offering single organoboronate products (**E-4v–E-4z**) in yields (80%-96%). Ethynyl oestradiol, which is an aliphatic alkyne with high steric hindrance, also provided the desired single organoboronate compound (**E-4aa**) in yield of 85%. These results further support that reversible Cu-N bonds can dissociate from the metal core of $Cu_4NC$ to reduce the steric effect of the catalytic sites, thus improving the catalytic activity and enhancing the substrate adaptability. No reactions were observed with the internal alkynes bis(4-tolyl) acetylene, diphenylacetylene and 1-phenylpropyne, indicating a high activation barrier for internal alkynes.

The chemoselectivity between $C \equiv C$ bonds and $C = C$ bonds was further explored via control experiments catalyzed by microcrystalline $Cu_4NC$. Equal amounts of phenylacetylene **1a** and styrene **3a** as substrates were added to the reaction system, and the reaction was catalyzed by the microcrystalline $Cu_4NC$ catalyst. Single vinylboronate ester **E-4a** was acquired in 94% yield, whereas alkylboronate ester **5a** was not observed in the hydroboration. In the intramolecular control experiment, 1-ethynyl-4-vinylbenzene as a substrate afforded only a single vinylboronate ester **E-4ab** in yield of 95%. These results showed that the microcrystalline $Cu_4NC$ catalyst possessed high chemoselectivity between alkynes and alkenes for substrates containing $C \equiv C$ bonds under mild conditions; thus, alkynes were the active substrates, and alkenes were inert substrates in the hydroboration catalyzed by microcrystalline $Cu_4NC$. When the reaction substrates contained both $C \equiv C$ bonds and $C = C$ bonds, the microcrystalline $Cu_4NC$ catalyst had high value in the special hydroboration of $C \equiv C$ bond functional groups.

## Mechanistic studies and Density Functional Theory (DFT) calculations

To further elucidate the mechanism of hydroboration catalyzed by microcrystalline $Cu_4NC$ and $Cu_8NC$ and determine why the microcrystalline $Cu_4NC$ catalyst has higher catalytic activity than microcrystalline $Cu_8NC$, a series of control experiments and density functional theory (DFT) calculations were conducted. First, deuterium experiments of hydroboration catalyzed by microcrystalline $Cu_4NC$ were explored under the optimal conditions with either $CD_3CN/H_2O$ or $CH_3CN/D_2O$ as the mixture solvent (Figs. 4a, b, Supplementary Figs. 42–46). When $CD_3CN/H_2O$ was used as the catalytic system solvent, the normal vinylboronate ester product **E-4a** was afforded in 98% yield and the deuterated vinylboronate ester was not observed, indicating that the $H_a$ atom did not come from $CD_3CN$ (Fig. 4a, Supplementary Fig. 42). When $H_2O$ was replaced with $D_2O$, two kinds of deuterated products were observed (Fig. 4b, Supplementary

**a Substrate scope in the hydroboration**

**b Chemoselectivities in the hydroboration**

**Fig. 3 | Performance of the Cu₄NC-catalyzed hydroboration of alkynes. a** Scope of alkynes for the Hydroboration. General reaction conditions: alkyne **1** (0.2 mmol, 1.0 equiv.), $B_2Pin_2$ **2** (0.44 mmol, 2.2 equiv.), the microcrystalline $Cu_4NC$ catal. (0.004 mmol, 2.0 mol%) and $K_2CO_3$ (0.44 mmol, 2.2 equiv.) were added to the mixture solvent (2.0 mL, MeCN/$H_2O$ 4:1) under air atmosphere at room temperature to react for 1 h. Isolated yields are given. [a]Reaction time: 2 h. [b]4,4′-Diethynyl-1,1′-biphenyl 1z (0.1 mmol) was used. **b** Investigation on chemoselectivity in the $Cu_4NC$-catalyzed intermolecular and intramolecular hydroboration of alkynes and alkenes. Intermolecular hydroboration reaction conditions: phenylacetylene **1a** (0.2 mmol, 1.0 equiv.), styrene **3a** (0.2 mmol, 1.0 equiv.), $B_2Pin_2$ **2** (0.44 mmol, 2.2 equiv.), the microcrystalline $Cu_4NC$ catal. (0.004 mmol, 2.0 mol%) and $K_2CO_3$ (0.44 mmol, 2.2 equiv.) were added to the mixture solvent (2.0 mL, MeCN/$H_2O$ 4:1) under air atmosphere at room temperature to react for 1 h. Isolated yields are given. Intramolecular hydroboration reaction conditions: 1-ethynyl-4-vinylbenzene **1ab** (0.2 mmol, 1.0 equiv.), $B_2Pin_2$ **2** (0.44 mmol, 2.2 equiv.), the microcrystalline $Cu_4NC$ catal. (0.004 mmol, 2.0 mol%) and $K_2CO_3$ (0.44 mmol, 2.2 equiv.) were added to the mixture solvent (2.0 mL, MeCN/$H_2O$ 4:1) under air atmosphere at room temperature to react for 1 h. Isolated yields are given.

**Fig. 4 | Investigation on the mechanism of the Cu₄NC-catalyzed hydroboration of alkynes. a** Deuterium experiments of the Cu₄NC-catalyzed hydroboration reaction in the mixture solvent (CD₃CN/H₂O). **b** Deuterium experiments of the Cu₄NC-catalyzed hydroboration reaction in the mixture solvent (CH₃CN/D₂O). **c** Deuterium experiments of the Cu₄NC-catalyzed hydroboration of phenylacetylene-d₁. **d, e** Radical scavenging experiments of Cu₄NC-catalyzed hydroboration. **f** Investigation on the chemoselectivity in Cu₄NC-catalyzed hydroboration.

Figs. 42–46). The $D_a$ and $D_b$ atoms clearly originated from $D_2O$. The $D_a$ atom was introduced into the deuterated products during the elimination reaction in hydroboration. Notably, the manner in which the $D_b$ atom was introduced into the deuterated products remains unknown. Two hypotheses are presented here. The first hypothesis concerning the source of the β-site $D_b$, is that copper catalysts easily activated the C≡C bond of alkynes to form an intermediate π-metal–alkyne complex, which enhanced the acidity of the alkyne proton, facilitating the formation of copper acetylides under basic conditions. In this case, copper acetylides, as active species, participated in hydroboration. During the elimination reaction, deuterated products containing $D_b$ atoms were easily obtained. The second hypothesis concerning the source of the β-site $D_b$, is that partial phenylacetylene was directly deuterated to phenylacetylene-d₁ under the action of the strong base $K_2CO_3$ and protic solvent $D_2O$.

To confirm which of the above hypotheses regarding the source of the β-site $D_b$ is correct, a series of control experiments were carried out. When phenylacetylene-d₁ was used instead of normal phenylacetylene (Fig. 4c, Supplementary Figs. 42, 47–50), the normal product and deuterated product were obtained, which further proved that the $D_b$ atom of the β-site was easily replaced by the H atoms of $H_2O$. When CH₃CN/D₂O was used as the mixture solvent in the absence of $B_2Pin_2$, and phenylacetylene (**1a**) and $K_2CO_3$ were added to the reaction system catalyzed by microcrystalline Cu₄NC, providing phenylacetylene-d₁ in 96% yield (Supplementary Figs. 60–63). Another control experiment was performed: phenylacetylene (**1a**) and $K_2CO_3$ were directly added to the CH₃CN/H₂O mixture solvent under the reaction conditions without the addition of the microcrystalline Cu₄NC catalyst, affording phenylacetylene-d₁ in 94% yield (Supplementary Figs. 60 and 67–69). These results indicated that the deuteration of phenylacetylene was independent of microcrystalline Cu₄NC, and the second hypothesis is thus more reasonable, in which the partial phenylacetylene was directly deuterated to phenylacetylene-d₁ under the strong base $K_2CO_3$. When phenylacetylene and phenylacetylene-d₁

were applied to the hydroboration reaction catalyzed by microcrystalline Cu₄NC in the CH₃CN/D₂O mixture solvent, two deuterated products were obtained. Based on these results, relevant control experiments were also conducted with Cu₈NC to gain more insight into the hydroboration of these copper clusters (Supplementary Figs. 51–60 and 64–66). As expected, the process and principle of deuterium experiments with the microcrystalline Cu₈NC catalyst were consistent with those of microcrystalline Cu₄NC.

To further explore the possibility of the presence of boron radical species, BHT (4.0 equiv.) or o-cresol (4.0 equiv.) was added to the hydroboration reaction systems catalyzed by microcrystalline Cu₄NC under standard conditions, and the single vinylboronate ester **E-4a** was still obtained in yield of 98% or 97%, which further verified that the radical species may not participate in the process[43,51] (Fig. 4d, e, Supplementary Fig. 39). When both the C≡C bonds of alkynes and the C=O bonds of aldehydes were simultaneously present within the microcrystalline Cu₄NC-catalyzed reaction system, the single vinylboronate ester product **E-4a** was still generated in yield of 97%, and benzaldehyde was almost completely recovered, proving that microcrystalline Cu₄NC is a specific catalyst for the hydroboration of alkynes (Fig. 4f, Supplementary Figs. 40 and 41). To verify the high regio-, stereo- and chemoselectivity in hydroboration catalyzed by microcrystalline Cu₄NC, we monitored the hydroboration process via in situ ¹⁹F NMR spectroscopy and found that the characteristic peak of 2-F-PA at -110.10 ppm gradually disappeared and that only the characteristic peak of vinylboronate ester **E-4g** at -117.65 ppm gradually appeared (Supplementary Fig. 70), which demonstrated that microcrystalline Cu₄NC is an efficient and highly selective catalyst for hydroboration.

In-depth understanding of the mechanism of hydroboration catalyzed by microcrystalline Cu₄NC or Cu₈NC was obtained via in situ Fourier transform infrared (FT-IR) spectroscopy and in situ Raman spectroscopy analysis (Supplementary Figs. 83–89). As shown in Supplementary Figs. 87 and 89, the top view of the three-dimensional surface of the in situ FT-IR spectra provided the selected individual

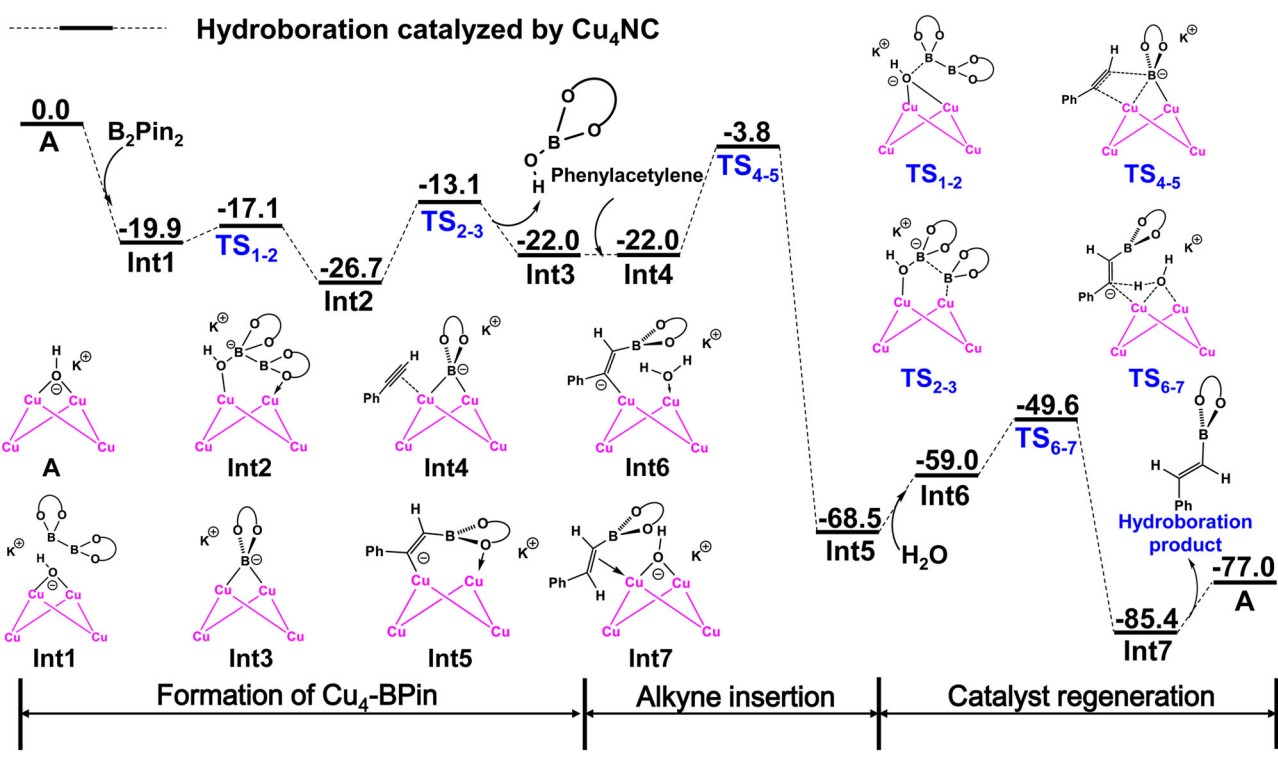

**Fig. 5 | DFT-calculated Gibbs free energy profile.** DFT-calculated Gibbs free energy profile (ΔG in kcal/mol) for the hydroboration catalyzed by microcrystalline $Cu_4NC$.

spectra of key signals for the hydroboration reaction catalyzed by microcrystalline $Cu_4NC$ or $Cu_8NC$. The IR peak intensities of B-B bonds (1124 cm⁻¹ for stretching vibrations) and B-O bonds (1284 cm⁻¹ for stretching vibrations) in $B_2Pin_2$ gradually decreased as the reaction proceeded, illustrating the progressive consumption of the $B_2Pin_2$ substrate[52–54]. Moreover, the IR peaks appeared at 1148 cm⁻¹, 1352 cm⁻¹ and 1624 cm⁻¹, which could be assigned to the B-C bond, B-O bond and C = C bond stretching vibrations of the vinylboronate ester target product, respectively[52–54]. The IR peak intensities of the B-C bonds, B-O bonds and C = C bonds in the target product gradually increased with increasing reaction time. A comparison of the in situ FT-IR spectra of the microcrystalline $Cu_4NC$ and microcrystalline $Cu_8NC$ catalytic systems further revealed that the catalytic activity of the microcrystalline $Cu_4NC$ was the stronger than that of microcrystalline $Cu_8NC$. As shown in Supplementary Figs. 83 and 84, the characteristic Raman peak at 690 cm⁻¹ was attributed to the stretching vibration peak of the Cu-N bond of microcrystalline $Cu_4NC$[55,56]. The Raman peak intensities of the Cu-N bonds gradually decreased as the reaction proceeded, indicating that the Cu-N bonds may have dissociated during the hydroboration reaction. These changes in the Raman spectra were attributed to the interaction between the substrates and the microcrystalline $Cu_4NC$ catalyst, and further supported that the flexible Cu-N bonds of the copper NCs could dynamically dissociate from the copper cluster core to generate key intermediates.

To obtain a deeper understanding of the mechanism of copper NC-catalyzed hydroboration, DFT calculations were performed to explain why the catalytic activity of microcrystalline $Cu_4NC$ is stronger than that of microcrystalline $Cu_8NC$ (Figs. 5, 6, Supplementary Figs. 72–79, Supplementary Tables 2 and 3). As shown in Fig. 5 and Supplementary Fig. 78, the catalytic process of the hydroboration reaction catalyzed by copper NCs can be divided into three major steps: borylation to form Cu NCs-BPin, alkyne insertion and catalyst regeneration. In a basic proton environment, the two flexible Cu-N bonds of the copper NCs could dynamically dissociate from the copper cluster core to generate key intermediates A and B (Supplementary Figs. 72 and 73). As shown in Supplementary Fig. 72, the energy barrier leading to intermediate A was

only 2.9 kcal/mol and the energy of intermediate A was -19.3 kcal/mol, indicating that this process was both kinetically and thermodynamically favorable. The theoretical calculation results were consistent with the experimental results, which further illustrated that the Cu-N bonds were the flexible in this system. Compared with the Cu-N bonds, the Cu-S bonds were stable during this process, thus maintaining the catalyst integrity and efficient catalytic activity. The in situ-generated intermediate Cu NCs-OH and electrophile $B_2Pin_2$ afforded Int3 $Cu_4NC$-BPin and Int10 $Cu_8NC$-BPin through addition reactions, releasing energies of 22.0 kcal/mol and 19.6 kcal/mol, respectively. The intermediate Cu NCs-BPin with exposed catalytic sites adsorbed alkynes, and then, the insertion of alkynes occurred via transition states $TS_{4-5}$ and $TS_{11-12}$, which had energy barriers of 18.2 kcal/mol and 24.7 kcal/mol, respectively. Int2, Int10, $TS_{4-5}$ and $TS_{11-12}$ represent the intermediates and transition states of the rate-determining step in copper NC-catalyzed hydroboration. The activation energy of $Cu_4NC$-catalyzed hydroboration was 22.9 kcal/mol, corresponding to the free energy difference between $Int_2$ and $TS_{4-5}$ ($\Delta G_1 + \Delta G_2$), which was lower than the activation energy of $Cu_8NC$-catalyzed hydroboration ($\Delta G_3 = 24.7$ kcal/mol) (Figs. 5, 6 and Supplementary Figs. 72–79). In addition, to better explain the differences in the catalytic performance of $Cu_4NC$ and $Cu_8NC$ in the hydroboration reaction from a theoretical calculation perspective, several popular functionals were tested. In particular, the energy difference between the profiles for $Cu_4NC$ and $Cu_8NC$ calculated via the gold standard M06 functional was 12.8 kcal/mol (Supplementary Table 2). These results clearly illustrate that the catalytic activity of $Cu_4NC$ in the hydroboration reaction is better than that of $Cu_8NC$. In an alkaline proton solution, Int5 and Int12 could produce vinylboronate esters and key intermediates A and B via protonation, proving that the copper NCs were regenerated and could enter the next cycle. These theoretical calculation results further support and explain why the microcrystalline $Cu_4NC$ catalyst displayed good hydroboration performance.

As shown in Fig. 6, plausible mechanisms of the hydroboration reactions catalyzed by copper NCs were proposed based on the above-described in situ characterization, control experiments and theoretical calculations. The organic transformation commences with dynamic

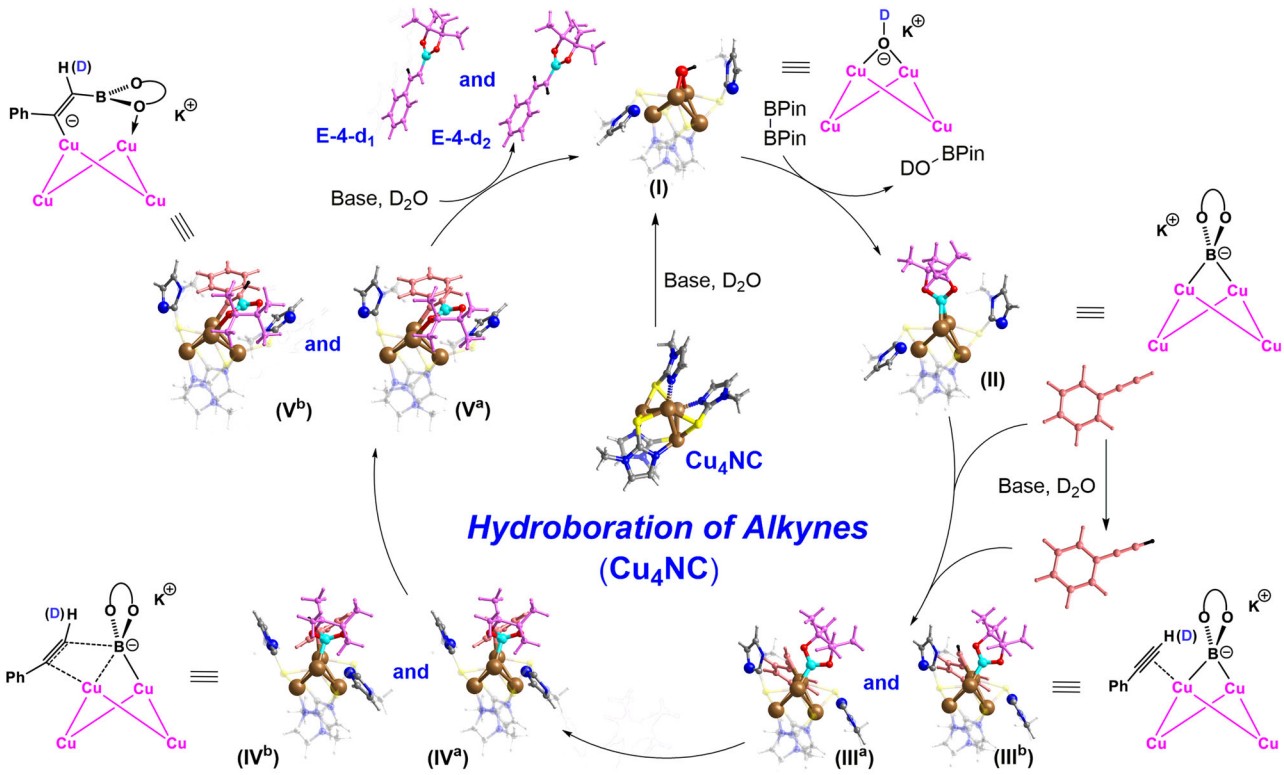

**Fig. 6 | A Proposed reaction mechanism.** A Proposed reaction mechanism for the formation of vinylboronate esters catalyzed by microcrystalline Cu₄NC.

dissociation of two flexible Cu-N bonds between the Cu₄NC metal core and the NHT ligands, releasing the catalytic center to enhance the catalytic activity and promote the generation of the key intermediate Cu₄NC-OD (I) in a deuterated alkaline protic solution. The intermolecular nucleophilic attack between the electrophile B₂Pin₂ and the nucleophile intermediate complex (I) generates the intermediates Cu₄NC-BPin (II) and DO-BPin. Owing to the presence of a deuterated alkaline protic solvent, some alkynes are deuterated by D₂O, providing alkyne mixtures of alkyne-d₁ compounds and normal alkynes during the hydroboration reaction. The exposed catalytic sites of the intermediate Cu₄NC-BPin (II), which originate from the dynamic dissociation of the Cu-N bonds, have sufficient space to adsorb the alkyne mixtures, resulting in the intermediates (IIIᵃ) and (IIIᵇ). The in situ-generated intermediate (III) can generate intermediates (IVᵃ) and (IVᵇ) by activating the alkyne mixtures via the exposed copper catalytic sites of Cu₄NC. Upon dissociation of the Cu-B bond and coordination of the flexible Cu-O bond between the Cu₄NC metal core and the BPin of intermediate (IV), the in situ-generated intermediate (IV) affords the intermediates (Vᵃ) and (Vᵇ) via addition. The key intermediate Cu₄NC-OD (I) is regenerated by protonation of the intermediate (V) in the deuterated alkaline protic solvent to produce the vinylboronate ester products **E-4d₁** and **E-4d₂**. As shown in Supplementary Fig. 76, the mechanism of the hydroboration process catalyzed by Cu₈NC is almost identical to that of the reaction catalyzed by Cu₄NC, which needs to proceed from the key intermediate Cu₈NC-OD (VI) to the intermediate (X), resulting in the desired vinylboronate ester products. When the hydroboration process is complete, the copper NC catalysts are recoordinated via the dissociated flexible and reversible Cu-N bonds between the metal core and the NHT ligands of the copper NCs, as revealed by the PXRD patterns, UV−vis spectra, ¹H NMR spectra, ESI–MS spectra and Raman spectra obtained to demonstrate the ultrahigh stability of the copper NCs and the rationality of the underlying mechanisms (Supplementary Figs. 31–38 and 85).

Microcrystalline Cu₄NC, as a catalyst, exhibits superb stability, good catalytic activity and regio-, stereo- and chemoselectivity, and is thus expected to become a practical catalyst for hydroboration reactions. To further investigate the physical and chemical properties of the microcrystalline Cu₄NC catalyst, preliminary kinetic studies of hydroboration were performed. An analysis of the linear fit of the $\ln(C_0/C)$ vs. reaction time (t) curve, revealed that the hydroboration transformations had a pseudo-first-order rate dependence on the concentration of phenylacetylenes (Supplementary Figs. 80 and 81). Using a $7 \times 10^{-4}$ mmol microcrystalline Cu₄NC catalyst, large-scale vinylboronate ester **E-4a** (12396.8 mg) was synthesized in 98% isolated yield under mild conditions for 12.5 h (Fig. 7a, Supplementary Table 1), which further confirmed the catalytic performance of microcrystalline Cu₄NC. Significantly, the boryl groups of vinylboronate esters are important building blocks of organic intermediates, which are beneficial for the conversion of organic functional groups through halogenation, arylation, etc. Disubstituted E-alkenyl halides (**E-7a** and **E-8a**) could be efficiently synthesized through halogenation reactions of vinylboronate ester **E-4a** (Fig. 7b, c)[57]. The (E)-(2-azidovinyl) benzene **E-10a** with a defined configuration was also generated in 92% yield via CuSO₄-mediated transformation of vinylboronate ester **E-4a** (Fig. 7e)[4,41]. Through the Suzuki cross-coupling reaction of vinylboronate ester **E-4a** and Ar-X (X: Br or I) compounds, functional group derivatizations of vinylboronate ester **E-4a** was achieved to provide the trans-stilbenes **E-9a** and **E-11a** (Fig. 7d and f)[4,41]. The flexibility of the boryl groups was fully utilized to promote the development of vinylboronate ester derivatizations. Through an in-depth study of the microcrystalline Cu₄NC catalyst, its good stability and reusability were recognized. The hydroboration reaction of alkynes was conducted for 5 cycles with the microcrystalline Cu₄NC catalyst. Nearly 100% yields were achieved for vinylboronate ester in several cyclic experiments (Supplementary Fig. 94).

## Discussion

In conclusion, DRDS copper NCs bearing NHT ligands with dynamic ligand effects were successfully developed in the gram scale and were further investigated as heterogeneous catalysts with high stability, atomically precise structures and dynamic dual catalytic sites. In

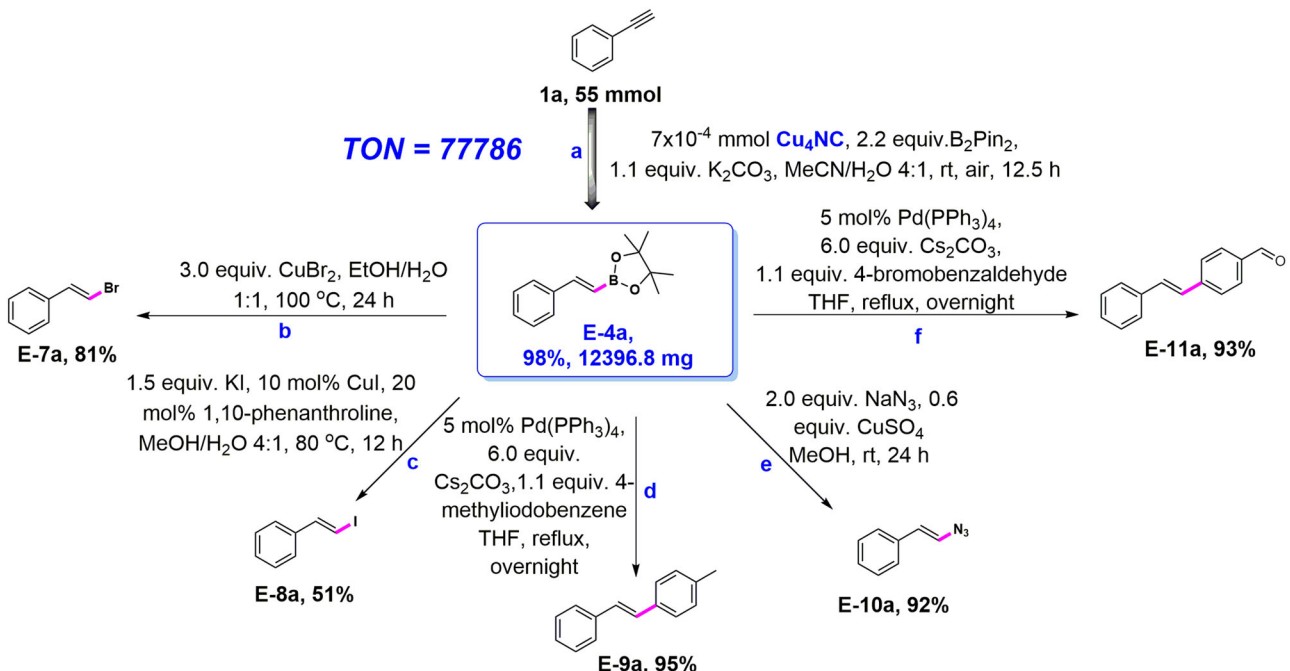

**Fig. 7 | Catalytic performance of Cu₄NC and derivatization of the B−C bond of vinylboronate ester E-4a. a** Large-scale synthesis of vinylboronate ester **E-4a**. **b**, **c** Synthesis of disubstituted E-alkenyl halides via halogenation reactions. **d** Construction of C − C bonds through Suzuki cross-coupling of vinylboronate ester and Ar-I. **e** Synthesis of (E)-(2-azidovinyl) benzene from a vinylboronate ester as substrate. **f** Exploration of the functional group derivatizations of trans-stilbenes via a method.

particular, the designed microcrystalline Cu₄NC with a well-defined structure displayed high catalytic performance for the hydroboration of alkynes with regio-, stereo- and chemoselectivity (up to 100%), achieving yields (up to 99%) under mild conditions (air atmosphere, protic solvent and room temperature). Importantly, microcrystalline Cu₄NC achieved a TON of up to 77786, which was higher than those reported for other catalysts. A combination of SCXRD analysis, control experiments, in situ characterization and theoretical calculations consistently revealed that the performance of Cu₄NC originated from the stable Cu-S bond and the reversible Cu-N bond, which promoted the formation of dynamic dual catalytic centres. This work emphasizes the importance of precisely constructing cluster catalysts with dual catalytic sites to achieve improved catalytic properties and elaborates the structure-activity relationship between the structure of cluster catalysts and their catalytic properties, thus offering a model for designing and constructing heterogeneous cluster catalysts for catalytic research, which can be expected to break ground in organic catalysis of heterogeneous copper clusters and promote the rapid development of cluster chemistry.

## Methods

All the reactions were carried out under ambient atmosphere (air) conditions unless otherwise noted. All commercial reagents and solvents were obtained from the commercial provider and used without further purification. ¹H NMR, ¹³C{¹H} NMR, ¹⁹F NMR and ¹¹B{¹H} NMR spectra were recorded on Bruker 600 MHz spectrometers. Chemical shifts were reported relative to internal tetramethylsilane (δ 0.00 ppm), CDCl₃ (δ 7.26 ppm) for ¹H NMR and CDCl₃ (δ 77.0 ppm) for ¹³C NMR. The nuclear magnetic spectra were analyzed using MestReNova software. Flash column chromatography was performed on 300−400 mesh silica gel.

### Instrumentation

Electrospray ionization mass spectrometry (ESI−MS) of the clusters was recorded using an AB Sciex X500R Q-TOF spectrometer. Powder

X-ray diffraction (PXRD) patterns of Cu₄NC and Cu₈NC were obtained using a Rigaku B/Max-RB X-ray diffractometer with Cu-Kα radiation (λ = 1.5418 Å) in air at room temperature. UV–vis absorption spectra were obtained by means of a Hitachi UH4150 UV–visible spectrophotometer. Energy dispersive spectroscopy (EDS) and elemental mapping measurements were carried out via Zeiss Sigma 500. The in situ IR spectra of all reactions were recorded on ReactIR 701 C230516193 spectrometer. The in situ Roman spectra of hydroboration catalyzed by microcrystalline Cu₄NC were recorded on a labRAM HR Evolution-HORIBA Raman system equipped with a CCD detector using a 532 nm He-Cd laser as the excitation source. Inductively coupled plasma mass spectrometer (ICP-MS) was performed on IRIS Advantage (Thermo). Elemental analyses (EA) were carried out with a Perkin-Elmer 240 elemental analyser. The X-ray diffraction datum of Cu₄NC and Cu₈NC were measured on a Rigaku XtaLAB Pro diffractometer (Supplementary Table 4).

### Materials

Hydroboration of alkynes was carried out under ambient atmosphere unless otherwise noted. Cu₄NC and Cu₈NC were synthesized under air atmosphere. The synthesis of deuterated phenylacetylene and 1-ethynyl-4-vinylbenzene substrates were carried out under a dry and oxygen-free N₂ atmosphere using standard Schlenk techniques. THF (HPLC grade) was dried over sodium/benzophenone and distilled under nitrogen prior to use. Methimazole (99%), B₂Pin₂ (99%), (Ph₃P)₂PdCl₂ (98%), 4-bromostyrene (98%), 2-methyl-2-propanethiol (99%), alkynes (97%-98%), ⁿBuLi (1.6 mol/l in hexane), methanol-d₄ (99.8%), various solvents (HPLC grade) and various copper salts (97%-98%) were directly purchased from companies of Energy Chemical, Bidepharm and Heowns without further purification.

### Synthesis of Cu₄NC

Under air atmosphere, the copper salt [Cu(MeCN)₄]PF₆ (1490.9 mg, 4.0 mmol) was dissolved in 100 mL of MeCN, to which 100 mL of methanol solution containing methimazole (456.7 mg, 4.0 mmol) was

added under vigorous stirring at room temperature. After the solution turned blue, the reaction was stirred for another 10 min. Then excess triethylamine (2.0 ml) was added to the stirring reaction, resulting in immediate formation of microcrystalline. The suspension was centrifuged, and the microcrystalline was collected. The microcrystalline was dissolved in DCM, and the resulting solution was diffused with MeCN to obtain colorless crystals of $Cu_4NC$ after 2 days at room temperature (Yield: 75 %, calculated based on methimazole ligand). **¹H NMR** (600 MHz, $CDCl_3$) δ 6.55 (d, $J = 20.6$ Hz, 8H), 3.57 (s, 12H). **¹³C{¹H} NMR** (151 MHz, $CDCl_3$) δ 151.1, 125.7, 119.9, 34.2. **HRMS(ESI)** m/z: [M+Cs]⁺ calcd for $C_{16}H_{20}CsCu_4N_8S_4^+$ 838.6913. found: 838.7003. Anal. Calc. for $Cu_4NC$ ($C_{16}H_{20}Cu_4N_8S_4$): C, 27.19; H, 2.85; N, 15.85; S, 18.14. Found: C, 27.23; H, 2.76; N, 15.75; S, 18.21.

### Synthesis of $Cu_8NC$

Under air atmosphere, $^tBuSCu$ (305.5 mg, 2.0 mmol) was dissolved in 100 mL of DCM. After sonication, the DCM solution of $^tBuSCu$ became clear, to which 100 mL of methanol solution containing methimazole (114.2 mg, 1.0 mmol) was added under stirring at room temperature. Then excess triethylamine (500.0 μL) was added to the stirring reaction, and the reaction stirred for another 10 min. The resulting pale-yellow solution was allowed to evaporate slowly at room temperature. After 3 days, yellow block crystals of $Cu_8NC$ were obtained in a yield of 91.0% (calculated based on $^tBuSCu$). **¹H NMR** (600 MHz, $CDCl_3$) δ 6.46 (dd, $J = 18.9, 1.4$ Hz, 8H), 3.58 (s, 12H), 1.36 (s, 36H). **¹³C{¹H} NMR** (151 MHz, $CDCl_3$) δ 150.6, 124.8, 119.5, 100.0, 48.0, 35.5, 34.0. **HRMS(ESI)** m/z: [M+Cs]⁺ calcd for $C_{32}H_{56}CsCu_8N_8S_8^+$ 1450.5766. found: 1450.5869. Anal. Calc. for $Cu_8NC$ ($C_{32}H_{56}Cu_8N_8S_8$): C, 29.17; H, 4.28; N, 8.50; S, 19.46. Found: C, 29.22; H, 4.24; N, 8.42; S, 19.52.

### General procedure for $Cu_4NC$-catalyzed the hydroboration reaction

Under air atmosphere, alkynes **1** (0.2 mmol, 1.0 eq), $B_2Pin_2$ **2** (0.44 mmol, 2.2 equiv.), microcrystalline $Cu_4NC$ catalysts (2.0 mol%), $K_2CO_3$ (0.44 mmol, 2.2 equiv.) and mixture solvent (2.0 mL, MeCN-$H_2O$, v/v 4/1) were added into the tube. The reaction mixture was stirred at room temperature for 1 h or 2 h. The reactions were monitored by TLC. When alkynes were consumed, the reactions were quenched and concentrated. The crude products were then purified by column chromatography to give the target products.

### General procedure for $Cu_4NC$-catalyzed the hydroboration reaction in 55 mmol scale

Under air atmosphere, phenylacetylene **1** (5615.5 mg, 55.0 mmol, 1.0 eq), $B_2Pin_2$ **2** (30721.9 mg, 121.0 mmol, 2.2 equiv.), microcrystalline $Cu_4NC$ catalysts (0.5 mg, $7 \times 10^{-4}$ mmol, $1.3 \times 10^{-3}$ mol%), $K_2CO_3$ (8361.7 mg, 60.5 mmol, 1.1 equiv.) and the mixture solvent (55.0 mL, MeCN-$H_2O$, v/v 4/1) were added into a 100 mL flask. The reaction mixture was stirred at room temperature for 12.5 h. The reactions were monitored by TLC. When alkynes were consumed, the reactions were quenched and concentrated. The crude products were then purified by column chromatography to give the (E)-4,4,5,5-tetramethyl-2-styryl-1,3,2-dioxaborolane **(E-4a)** products as colorless oil with an overall isolated yield: 98% (12396.8 mg).

### Recycling procedure for $Cu_4NC$-catalyzed the hydroboration reaction

Under air atmosphere, phenylacetylene **1** (20.4 mg, 0.2 mmol, 1.0 eq), $B_2Pin_2$ **2** (111.8 mg, 0.44 mmol, 2.2 equiv.), microcrystalline $Cu_4NC$ catalysts (2.8 mg, 2.0 mol%), $K_2CO_3$ (60.7 mg, 0.44 mmol, 2.2 equiv.) and the mixture solvent (2.0 mL, MeCN-$H_2O$, v/v 4/1) were added into a 10 mL flask. The reaction mixture was stirred at room temperature for 1 h. Conversions and yields were determined by ¹H NMR using 1,3,5-trimethoxybenzene as an internal standard. After the reaction is completed, the reaction mixture was centrifuged at 8000 rpm for 5 min. The

microcrystalline $Cu_4NC$ catalyst was filtered, washed with hexane and then dried in vacuum to recycle the microcrystalline $Cu_4NC$. The recycled microcrystalline $Cu_4NC$ catalyst was used again in the next recycle. In this way, the hydroboration reaction of alkynes was conducted for 5 cycles with the microcrystalline $Cu_4NC$ catalyst (Supplementary Fig. 94).

## Data availability

All other data are available from the corresponding author upon request. All data needed to evaluate the conclusions in the paper are present in the paper and/or the Supplementary Materials (including Supplementary Figs. 1–176, details of the chemicals, instrumentation, synthesis, characterization, DFT and X-ray crystal details for $Cu_4NC$ (CIF) and $Cu_8NC$ (CIF)). The source data for Figs. 5 and 6 and Supplementary Figs. 74–79 are provided with this paper. The X-ray crystallographic coordinates for the structures reported in this article have been deposited at the Cambridge Crystallographic Data Centre (CCDC), under deposition number CCDC 2330281 ($Cu_4NC$) 2330282 ($Cu_8NC$). These data can be obtained free of charge via www.ccdc.cam.ac.uk/data_request/cif. Source data are provided with this paper.

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

## Acknowledgements

This work was supported by the National Natural Science Foundation of China (No. 21825106, S.-Q.Z.; 92061201, S.-Q.Z.; 22201261, M.-M.Z.; 22309113, Z.W.; 22369008, F.L.), the China Postdoctoral Science

Foundation (2022M722864, T.J.), Zhongyuan Thousand Talents (Zhongyuan Scholars) Program of Henan Province (234000510007, S.-Q.Z.), Scientific and Technological Project of Yunnan Precious Metals Laboratory (YPML-2023050204, Z.W.) and Zhengzhou University.

## Author contributions

S.-Q.Z. and T.J. conceived the idea. T.J. synthesized nanocluster catalysts and conducted the hydroboration transformation experiments. J.A., Y.H., C.-F.S. and Y.-X.L. assisted with the experiments and characterizations. M.-M.Z. helped with the manuscript revising and data analysis. S.-Q.Z., R.-W.H. and T.J. discussed the results. Z.W., X.-G.L. and F.L. conducted theoretical calculations of the hydroboration. R.-W.H. and T.J. wrote the manuscript. S.-Q.Z. revised the manuscript. All the authors reviewed and contributed to this paper.

## Competing interests

The authors declare no competing interests.
