## [Peer Review File · Nature Communications]

Synergistic Dual Sites in Atomically Precise Copper Clusters for Highly Chemoselective and Efficient Hydroboration

Corresponding Author: Professor Shuang-Quan Zang

Version 0:

Reviewer comments:

Reviewer #1

(Remarks to the Author)

Zang and coworkers report two dynamically regulating dual-catalytic site copper cluster heterogeneous catalysts for hydroboration with high chemoselectivity for specific functional groups under mild conditions. The DRDS Cu₄NC achieved the TON of 77786, which was significantly higher than those of known catalytic systems. The mechanism of hydroboration catalyzed copper clusters is well demonstrated with sufficient experimental data and theoretical calculations. The results revealed that the dynamically regulating dual-catalytic sites of DRDS copper cluster catalysts played the key role in hydroboration reactions. The manuscript illuminates the structure–activity relationship of the DRDS copper cluster catalysts and it can provide a new reference to improve catalytic performance of cluster catalysts by fully leveraging synergistic interactions and dynamic ligand effects, which shows innovation on the design and the regulation of dual-catalytic site catalyst and breaks new ground in the construction and application of the clusters. I think the work is interesting and potentially impactful. Thus, I advise this manuscript can be published in Nature Communications after the following minor revisions.

1. Lack of NHT ligand as control group catalyst in Table 1. Some functional ligands are also great catalysts in many organic reactions. The catalytic activity of the NHT ligand in hydroboration should be provided.
2. In this work, terminal alkynes were used as important reactants in hydroboration reaction. How about the reaction with internal alkynes under the conditions in Fig. 3.
3. How about the reaction with the HBpin as the boron source under the optimal reaction conditions in Table 1.
4. In the last paragraph on the P13 and in Figure 4 (d and e), the loads of radical scavenger of BHT and o-cresol are different. Please confirm the loads of radical scavenger and modify them.
5. All figure legends should be presented in a uniform and consistent manner, which include a brief title and multiple detailed titles. I suggest the authors unify the form of the figure legends.
6. The format problem can be modified in some places, such as: “19.6. kcal/mol” in P15, the format of “7×10⁻⁴” and “7*10⁻⁴” should be in a uniform in the work.
7. Some recently examples about Cu NC and Cu catalyzed hydroboration should be cited: Angew. Chem. Int. Ed. 2023, 62, e202218369; Angew. Chem. Int. Ed. 2023, 62, e202306497; J. Mater. Chem. A, 2023, 11, 12777.

Reviewer #2

(Remarks to the Author)

In this article Zang, Wang and Huang report the synthesis of copper clusters and their use in protoboration of alkynes. The synthesis of the pre-catalysts is straightforward, simple and scalable and in addition these compounds exhibit a great air tolerance. The stabilization of the cluster is possible through the use of a deprotonable N-heterocyclic thione (methimazole) that binds the copper atoms through both the sulfur and nitrogen atoms, a coordination mode that apparently is very relevant to explain the catalytic behavior of the clusters. The two clusters synthesized are competent in the protoboration of alkynes, exhibiting a great chemo-, regio- and stereoselectivity. In addition, the reaction can be performed in air and one of the catalysts have proved extremely robust. The mechanism of the reaction has been analyzed by some labelling experiments (using either D₂O or deuterated substrates) and by DFT methods that have shown the relevance of both the ligand and the

proximity of the copper atoms to explain the selectivity observed. The simplicity of the catalyst, the procedures and the mechanistic study makes me think that this article might be suitable for publication in Nat. Commun. While this is a promising system for the efficient functionalization of substrates such as aromatic alkynes, I have some concerns that need to be addressed prior to publication.

1. First of all, in many occasions the utilization of the language is confusing and the English needs to be polished. I normally would not complain about this aspect of a paper, but this time I had some trouble trying to figure out what the authors really wanted to claim. Derived from this, expressions such as ‘...which can improve the catalytic performance and enhance the catalytic activity...’ can be found. What do they mean by this? What is the difference? On page 4, again the English needs to be improved. Figure 1 shows expressions such as ‘lowly catalytic activities’, ‘dried atmosphere’, ‘highly recover’...

2. The purity of the catalysts and the new compounds should be proved by either elemental analysis or High Resolution Mass Spectrometry. The copper clusters have been only characterized by NMR spectroscopy, PXRD and ESI-MS. The known borylated compounds should be either accompanied by a reference of its previous synthesis to compare the spectra or if totally new HRMS should be given.

3. The authors claim, in the abstract, that their catalyst exhibit and ultra-high TON of 77786. I don't believe this can be consider as ultra-high if we compared this to other catalytic processes whose TON are of the order of millions. It will be acceptable to say high TON. In addition, the statement that their catalyst is the best so far reported is questionable. The catalytic system given in reference 19 provided TOF numbers nearly 3 times better than those shown by the authors in this article (41743 h⁻¹ vs 16286 h⁻¹). Therefore, the sentence on page 7 “the TON values of the reported catalytic systems were very poor” should be removed, and in addition comparison of the TOF numbers with other catalytic systems should be provided.

4. Another relevant point that should be corrected is the fact that in the introduction the authors state that homogeneous transition metal complexes have been used under harsh reaction conditions. However, most of the reports used catalytic conditions at room temperature that are far from “harsh reaction conditions”. The fact that some of these processes are carried out under inert atmosphere cannot be considered as harsh. I encourage the authors to avoid over-emphasizing this type of statements. In addition, a puzzling claim the authors make throughout the manuscript is the heterogeneous character of their catalysts. This does not seem to be the case, as it is clearly demonstrated during the entire work that the species involved are in solution. In fact, they are synthesized in solution, and they can be crystallized. They can even be characterized by NMR or ESI-MS methods. Why do they claim this system is heterogeneous?

5. Page 5: I don't understand the phrase “flexible and reversible N-Cu coordinate bonds and stable S-Cu covalent bonds”. What is the meaning of coordinate bond? I don't see the difference between Cu-S and Cu-N bonds which very likely are both covalent in nature. In addition, the authors have coined or used the term “dynamically regulating dual-catalytic site”. The authors insist on the synergistic effect between multiple metallic atoms. In their case, what is the basis of such synergy? From my view this is just metal-ligand and metal-metal cooperativity in which, from one side, the Cu-N bond cooperates or assist in the deprotonation of H₂O, and the nearby copper atoms facilitate the insertion of the alkyne into the boryl fragment. Some references to this type of cooperation should be given.

6. With regard to the mechanism, the discussion in some points are confusing. First, the proposed mechanism in Figure 5 is very difficult to follow. The structures obtained by DFT calculation are too small. I encourage the authors to use Figure 64 in the SI that is much easier to follow and to send Figure 5 in the main text to the SI. Concerning the use of D₂O for the deuteration of phenylacetylene (pages 12-13), the authors state first that copper acetylides might be involved considering the incorporation of deuterium in the alkyne, but afterwards they proved that copper is not necessary for this exchange, contradicting the presence of copper acetylides. This should be clarified.

7. Regarding the mechanistic proposal, DFT calculations have been carried out. Although the energy barriers are somehow consistent with the experiments there is one step that is missing in the calculation. The calculated energy for the conversion of Cu₄NC or Cu₈NC into A or B should be provided, since this is the key step that proves the participation of the Cu-N bonds in the catalytic cycle. What are the energy barriers for this process? In addition, a simple experiment should be carried out to prove this step by reacting Cu₄NC or Cu₈NC with H₂O or D₂O. How important is the presence of the N-H generated in this step? Regardless of the isolation, an easy experiment such as an IR spectrum should shed light about the presence of a hydroxide group (with H₂O and D₂O). Additionally, if complexes A or B are sufficiently stable an experiment using a thione without N-H moieties should be carried out to prove its cooperative effect. The difference in energy between the calculated profiles for Cu₄NC and Cu₈NC is 1.8 kcal mol⁻¹. Given this very small energy difference, the authors should avoid terms expressions such as ‘clearly illustrated that Cu₄ largely exceeds Cu₈’. In fact, a different functional or basis set could easily invert these energies, making the Cu₈NC more favorable. Can the authors comment on this?

8. On page 8, the following can be read: ‘The result further proved that the structure of DRDS Cu₄NC and the distribution of dual-catalytic sites are more plausible than those of the other DRDS copper NCs.’ What does this mean? Sentences like these are frequently found in the manuscript and they are somewhat confusing and difficult to understand.

9. 1,3,5-trimethoxybenzene is utilized as an internal standard. Copper boryl species have been demonstrated to cleave aryl ethers (Tetrahedron Lett. 2017, 58, 17-20). Do the authors add the standard before or after the catalysis? If before, do they see any species that could come from trimethoxybenzene reactivity?

10. During the submission of this article a closely related system has been reported by Bakr, Rueping and co-workers (J. Am. Chem. Soc. 2024, 146, 16295–16305). I believe that the mechanistic difference between this system and the one reported herein justifies the inclusion of this reference in the main text.

11. On page 14, the authors claim that ‘The S-Cu covalent bonds were always very stable during this process to keep catalyst integrity and efficient catalytic activity’. How do they know this? What experiments have been carried out to demonstrate this claim?

12. Page 15: In a couple of occasions the authors mention that B₂pin₂ is the nucleophile, whereas intermediate complex(I) is the electrophile. B₂pin₂ is not a nucleophile, but an electrophile. Intermediate complex (I) (do they mean complex A?) is the nucleophile.

13. In the Supporting Information the ¹³C NMR spectra provided are ¹H decoupled. Therefore, this should be given as ¹³C{¹H}. In addition, no ¹¹B{¹H} NMR is provided for the new compounds.

Reviewer #3

(Remarks to the Author)

Reviewer #4

(Remarks to the Author)

In this report by Huang, Wang, and Zang 2 copper complexes with N-heterocyclic thione ligands are synthesised and their ability to undertake catalytic hydroboration of alkynes with a proton source and a diborane is explored experimentally and computationally. The reaction belongs to a class of extremely widely reported hydroborations with copper which have been well-known for decades (see section 3 of <https://pubs.rsc.org/en/content/articlelanding/2018/cs/c7cs00816c>), and represents at best a minor advance in this field.

The work is presented reasonably well, although a number of minor typos are present and some schemes are hard to follow (vide infra). The authors have performed a decently sized scope table, although only very limited examples of non-aromatic substituted alkynes are presented. I have no doubt that the reaction as described does work, and that it represents a useful advance in the area. Moreover, the compounds the authors claim are characterised appropriately - although I can't see evidence of bulk purity for the catalysts (CHN or calibrated NMR showing all compound added is in solution and not NMR silent). The larger concern, however, is that the work has serious flaws in its conclusions that would need to be remedied before publication.

There are some statements that should be clarified throughout such as where "resonant" (more usually "resonance") structures are discussed. These are discussed as if they have physical manifestation and are not simply ways of rationalising electronic structure. In the first sentence of "Catalyst Development and Characterization" the statement "The easily modifiable NHT ligands, including a resonant structure with a negative charge" isn't really true. The resonance structures of NHT ligands can be drawn as zwitterions, with the negative charge localised on the sulfur. This implies that the sulfur has more significant electron density associated with it than a standard thiocarbonyl. Which is fine, but needs to be discussed more carefully.

These issues are, however, minor in the face of the more significant issues in the work. Throughout the work, the copper "clusters" (nomenclature point really here - but is a small tetramer with a molecular weight of around 700 gmol⁻¹ really a cluster?), are referred to using two key concepts; one that this catalysis is heterogeneous, and two that they provide "DRDS" (dynamically regulating dual catalytic site) catalysis. At this stage, I can see no evidence provided by the authors to back up these claims.

The first point, that this reaction is heterogeneous may be true but I simply cannot tell.

- what evidence do the authors have that the reaction happens on the surface of a bulk material?

- are the authors, instead, contending that (relatively) small "clusters" are heterogeneous catalysts even if they go fully into solution? If so, that's a rather loose, and in my opinion inappropriate use of heterogeneous.

Thus, my first concern is whether the reaction is truly heterogeneous. Some hints that it may be are present in the manuscript including the line "DRDS Cu₄NC and DRDS Cu₈NC catalysts were also recovered and reused again." Now, the most reasonable way to do this would be to filter the reaction mixture, collect the filtrate, analyse it and show it to be Cu₄NC and Cu₈NC. I can see no evidence that the authors have done this in the manuscript, despite repeated claims of recovery. Perhaps the authors are referring to the recycling experiments present on page 46 of the supplementary? If so, they have written this section in a completely inappropriate way - repeated dosing is not recovery of the catalyst. If they have performed recovery experiments they should be in the ESI.

Significant evidence to the contrary is also provided - the authors provide solution state NMR of the catalysts! they thus form homogenous solutions with at least some solvent systems.

- if the reaction is heterogeneous in the traditional sense, is the DFT approach used in any way appropriate? My interpretation would be no, as it models a molecular system

Now, irrespective as to whether the statement that the reaction is "heterogeneous" is true or not (and I'm just asking for evidence there), it reflects a fundamental second flaw in the work. The authors provide absolutely no evidence as far as I can see that Cu₄NC is the actual catalytic vector! As a result, their entire mechanistic postulate sits on exceptionally shaky grounds and the DRDS argument lacks foundation. The authors provide a little evidence that the system does persist in solution throughout the catalysis (e.g. page 18 of the ESI), but to my eye there's more going on in the spectra than is discussed.

- What have the authors done to preclude the idea that the clusters come apart during the catalysis in solution, then when the substrate is consumed reoligomerise? In situ NMR of the reaction mixture may assist here

- What happens to the spectra of the materials when the substrates are added to the catalysts in stoichiometric studies?

A few minor comments on the larger claims of the synthetic work. The authors contend this may be viable on an industrial scale but provide no evidence for this claim. They should remove it, or justify it with experimental work. The authors vaunt the selectivity of the reaction for alkynes over alkenes - this is entirely unsurprising. Alkynes are a much easier substrate, this

is just an example of a relatively inactive catalyst. They also suggest selectivity over benzaldehyde, but they do the reaction in a water mixture! What's the evidence that the carbonyl remains intact in these conditions? Could it not have been hydrated to the geminal diol, which would then not be expected to be a viable substrate for hydroboration? On work up, this reversible hydration would then return the benzaldehyde.

Finally, a few minor comments on the DFT itself. Fig 5 is almost entirely incomprehensible - a slightly better drawn version is provided in the ESI but is still hard to follow. It's remarkable to me that the authors didn't cite Tilley's μ -copper boryl system which is the closest thing in the lit to their postulated intermediate (<https://pubs.rsc.org/en/content/articlelanding/2022/sc/d2sc00848c>). Given that Tilley can isolate such a compound, the authors should be able to do so for their systems if their mechanism has any credence. The DFT generally does seem reasonable - the level of theory is appropriate and the steps work, but at this stage the authors need to actually provide some evidence that there is a meaningful relationship between the catalysis as observed and the DFT conclusions.

In conclusion, this work needs significant improvement before publication, either through provision of more experiments or a much more careful write-up pointing out the speculative nature of many of the conclusions. Moreover, with these flaws answered it is at best a relatively prosaic step forward in this area of catalysis and would lack the broad appeal and impact for this journal.

Version 1:

Reviewer comments:

Reviewer #1

(Remarks to the Author)

The authors have sufficiently addressed all the comments. I now recommend the article for publication.

Reviewer #2

(Remarks to the Author)

The authors have made a great effort to improve the article by adding a series of experiments and explanations that have clarified most of my main concerns. Although I believe that there are still some points that would need further clarification, my opinion is that the results in the new version are clearly sufficient for being published in Nature Communications:

The utilization of English in the manuscript is still improvable, as incorrect conjugations are still present, and in some cases the utilization of mixed tenses (page 9: future and past tenses are mixed in the same paragraph).

The utilization of the word 'microcrystal' is utilized throughout the entire text. It should be 'microcrystalline', since it describes the copper clusters.

Page 5: By SCXRD analysis, the Cu-Cu distance is 2.662 Å and 2.663 Å in the Cu₄NC unit and Cu₈NC unit, respectively, which are slightly shorter than that of dicopper (I) μ -boryl complexes.¹⁵

This claim is incorrect. The distance described by the authors corresponds to the tert-butoxide dicopper species, instead of the boryl complexes. According to the reference, the boryl compounds possess copper-copper contacts of around 2.3 Å, much shorter than in the clusters described in this work. Therefore, this information is not valid for describing the proposed synergy between the copper centers.

Pages 8 and 9: the authors repeat the reasons for the heterogeneous character of their catalyst, making it redundant.

Page 16: the dissociation of the Cu-N bond, as suggested by Raman spectroscopy results, is described several times in the text, making it redundant. Shortening of this part would be clearly beneficial. Additionally, the authors should provide the experimental conditions for the Raman measurements, particularly if they have been recorded in solution or in the solid state. This is particularly relevant considering that if the measurement is made in solution, the complexes detected are soluble, providing evidence of the homogeneous character of the catalyst. Moreover, complexes Cu₄NC or Cu₈NC are in fact the pre-catalyst, being A or B (more stable than the pre-catalysts) the true catalysts whose solubility is unknown.

Page 17: again, a 1.8 kcal mol⁻¹ energy barrier difference between the Cu₄ and Cu₈ mechanism is not a robust result to claim the following: 'These results clearly illustrated that the catalytic activity of Cu₄NC is better than that of Cu₈NC in the hydroboration reaction'. I don't understand why the authors do not include the benchmark results, since the energy barrier difference can be as high as 12 kcal mol⁻¹, supporting their observations. The benchmark results should be included in the SI file, and their data discussed in the main text, in order to provide theoretical support to the experimental observations.

Supporting Information

Page 9: Figures S11 and S12 should include the description of parts 'a' and 'b'

Figures S91 and S92: what do the authors mean by dosing projects?

Reviewer #3

(Remarks to the Author)

Reviewer #4

(Remarks to the Author)

The authors have clearly done much to improve the work. The minor comments about claims in the manuscript are, generally speaking, covered by their revisions.

The nature of these systems as heterogeneous catalysts is now clear. This does rebut a number of the queries raised (for example, any sort of in situ monitoring by NMR is clearly therefore precluded). The Raman monitoring does provide some alternative to this, and provides evidence for the loss of the Cu-N bonds over the course of the reaction. Clearly these Raman data do therefore support the formation of species such as A.

1) It would be nice to show the Raman spectrum of $\text{Cu}_4\text{NC} + \text{K}_2\text{CO}_3$ in the absence of catalyst to provide further evidence of this.

2) Does the Raman spectrum of the recovered material show a complete return of these peaks?

The rationale for using DFT for this work is clearly considered on the authors part, although I do feel the authors are not fully considering the possibilities provided by a number of the Cu_4NC or Cu_8NC compounds being in close proximity on the surface of the insoluble material. Modelling a single molecular fragment of an insoluble material isn't the only alternative to taking a full heterogeneous approach, and the authors are in possession of understanding about the packing of the Cu_4NC and Cu_8NC compounds that would allow them to generate appropriate model systems to consider the possibility of more than a single cluster being involved in the reaction.

In terms of validating the Cu_4NC and Cu_8NC systems as the catalytic vector (and proving that the reaction is truly homogeneous), the authors provide evidence of insolubility of Cu_4NC , provide evidence of recovery without chemical change, provide evidence that the Cu(I) does not pass into solution, and provide evidence that Cu_4NC removed from the reaction at various time points shows consistent characteristics. What I can't see evidence of is evidence that shows that the supernatant lacks any catalytic ability. This should be easy to do by

3) removing the supernatant at a point prior to completion of the reaction and ensuring that no further conversion is observed within this solution.

4) removing the supernatant at the end of the reaction, and adding an additional aliquot of the substrates, and ensuring no conversion of the added material is observed.

At this stage, the balance of probabilities is that $\text{Cu}_4\text{CN}/\text{Cu}_8\text{CN}$ are the catalytic vector, but the formation of a soluble, catalytically competent, copper-free species under the reaction conditions still exists. It should be precluded.

Version 2:

Reviewer comments:

Reviewer #2

(Remarks to the Author)

The authors have done an excellent job addressing all of my concerns, and I believe the results presented here are therefore suitable for publication in Nat. Commun.

Just a minor typo, in some places (ESI) the Raman spectroscopy appears as "Roman" spectroscopy.

Reviewer #3

(Remarks to the Author)

Reviewer #4

(Remarks to the Author)

The authors have now answered all of my comments. I have 2 minor points that I suggest revising, but do not need to see the manuscript again.

Fig. 1b - "resonant" persists where "resonance" should be used

Fig. 5b - this is still very hard to parse.

Responses to Reviewers

Reviewer #1: (minor revisions)

Remarks: Zang and coworkers report two dynamically regulating dual-catalytic site copper cluster heterogeneous catalysts for hydroboration with high chemoselectivity for specific functional groups under mild conditions. The DRDS Cu₄NC achieved the TON of 77786, which was significantly higher than those of known catalytic systems. The mechanism of hydroboration catalyzed copper clusters is well demonstrated with sufficient experimental data and theoretical calculations. The results revealed that the dynamically regulating dual-catalytic sites of DRDS copper cluster catalysts played the key role in hydroboration reactions. The manuscript illuminates the structure–activity relationship of the DRDS copper cluster catalysts and it can provide a new reference to improve catalytic performance of cluster catalysts by fully leveraging synergistic interactions and dynamic ligand effects, which shows innovation on the design and the regulation of dual-catalytic site catalyst and breaks new ground in the construction and application of the clusters. I think the work is interesting and potentially impactful. Thus, I advise this manuscript can be published in Nature Communications after the following minor revisions.

Response: We thank the **Reviewer 1** for these positive comments very much. We have addressed the following issues as suggested.

Comment 1: Lack of NHT ligand as control group catalyst in Table 1. Some functional ligands are also great catalysts in many organic reactions. The catalytic activity of the NHT ligand in hydroboration should be provided.

Response: Thank the **Reviewer** for the insightful suggestion! This is a very good point. The NHT ligand as a catalyst for hydroboration reaction has been conducted under optimal conditions (Table 1, entry 19). The vinylboronate ester was not got, indicating that the NHT ligand has no catalytic ability in hydroboration. The result has been included in the revised manuscript.

Comment 2: In this work, terminal alkynes were used as important reactants in hydroboration reaction. How about the reaction with internal alkynes under the conditions in Fig. 3.

Response: Thanks a lot for the reviewer's suggestion. As suggested, we have tested the internal alkynes, bis(4-tolyl) acetylene, diphenylacetylene and 1-phenylpropyne in this reaction system. The hydroboration products were not observed under the conditions in Fig. 3, indicating a high activation barrier for internal alkynes. The corresponding discussions have been added into the revised manuscript for clarification.

Comment 3: How about the reaction with the HBpin as the boron source under the optimal reaction conditions in Table 1.

Response: This is a great point. Thanks very much for reviewer's suggestion! The HBpin as the boron source has been tried to hydroboration catalyzed by the Cu₄NC catalyst, no reaction was observed under optimal conditions (Table 1, entry 9). We have included the result in the revised manuscript.

Comment 4: In the last paragraph on the P13 and in Figure 4 (d and e), the loads of radical scavenger of BHT and o-cresol are different. Please confirm the loads of radical scavenger and modify them.

Response: Thanks for your reminder. We have carefully checked and corrected the corresponding mistake in the revised manuscript.

Comment 5: All figure legends should be presented in a uniform and consistent manner, which include a brief title and multiple detailed titles. I suggest the authors unify the form of the figure legends.

Response: Thank the **Reviewer** for the comment. We have made the revision as suggested. Thank you so much for helping improve the quality of manuscript.

Comment 6: The format problem can be modified in some places, such as: “19.6. kcal/mol” in P15, the format of “ 7×10^{-4} ” and “ $7 * 10^{-4}$ ” should be in a uniform in the work.

Response: Thanks for your advice. We have carefully checked and revised the format problem in the manuscript.

Comment 7: Some recently examples about Cu NC and Cu catalyzed hydroboration should be cited: Angew. Chem. Int. Ed. 2023, 62, e202218369; Angew. Chem. Int. Ed. 2023, 62, e202306497; J. Mater. Chem. A, 2023, 11, 12777.

Response: Thanks very much for the reviewer’s comment and suggestion. As suggested, we have cited these literatures in the revised manuscript.

Reviewer #2:

Remarks: In this article Zang, Wang and Huang report the synthesis of copper clusters and their use in protoboration of alkynes. The synthesis of the pre-catalysts is straightforward, simple and scalable and in addition these compounds exhibit a great air tolerance. The stabilization of the cluster is possible through the use of a deprotonable N-heterocyclic thione (methimazole) that binds the copper atoms through both the sulfur and nitrogen atoms, a coordination mode that apparently is very relevant to explain the catalytic behavior of the clusters. The two clusters synthesized are competent in the protoboration of alkynes, exhibiting a great chemo-, regio- and stereoselectivity. In addition, the reaction can be performed in air and one of the catalysts have proved extremely robust. The mechanism of the reaction has been analyzed by some labelling experiments (using either D₂O or deuterated substrates) and by DFT methods that have shown the relevance of both the ligand and the proximity of the copper atoms to explain the selectivity observed. The simplicity of the catalyst, the procedures and the mechanistic study makes me think that this article might be suitable for publication in Nat. Commun. While this is a promising system for the efficient functionalization of substrates such as aromatic alkynes, I have some concerns that need to be addressed prior to publication.

Response: We appreciate **Reviewer 2** for the positive comments and the following constructive suggestions to improve our manuscript. We have addressed the following issues as suggested.

Comment 1: First of all, in many occasions the utilization of the language is confusing and the English needs to be polished. I normally would not complain about this aspect of a paper, but this time I had some trouble trying to figure out what the authors really wanted to claim. Derived from this, expressions such as ‘...which can improve the catalytic performance and enhance the

catalytic activity...’ can be found. What do they mean by this? What is the difference? On page 4, again the English needs to be improved. Figure 1 shows expressions such as ‘lowly catalytic activities’, ‘dried atmosphere’, ‘highly recover’...

Response: Thank the Reviewer very much for the comment. We apologize for the inconvenience caused by the language. Following this suggestion, we have double-checked the manuscript and corrected the language. All changes have been highlighted in yellow in the revised manuscript.

Comment 2: The purity of the catalysts and the new compounds should be proved by either elemental analysis or High Resolution Mass Spectrometry. The copper clusters have been only characterized by NMR spectroscopy, PXRD and ESI-MS. The known borylated compounds should be either accompanied by a reference of its previous synthesis to compare the spectra or if totally new HRMS should be given.

Response: Thanks a lot for reviewer’s suggestion! As suggested, the copper clusters have been characterized by SCXRD (Supplementary Figures 5 and 6), elemental analysis, ESI-MS (Supplementary Figures 9 and 10), EDS-Mapping (Supplementary Figures 11 and 12), NMR spectroscopy (Supplementary Figures 13-22, 27-30), PXRD (Supplementary Figures 7 and 8), and UV-vis spectroscopy (Supplementary Figures 23-26) to prove the purity and physicochemical properties of the catalysts. On the basis of the ^1H and ^{13}C NMR spectroscopy, the known borylated compounds have been accompanied by the references of their previous synthesis to compare the spectra. The unreported vinylboronate ester compounds (**E-4q**, **E-4v**, **E-4y** and **E-4z**) have been also characterized by HRMS (ESI) and ^{11}B NMR spectroscopy (Supplementary Figures 146, 157, 164 and 167). These results have been included in the revised manuscript.

Elemental analysis of Cu_4NC and Cu_8NC :

Anal. Calc. for Cu_4NC ($\text{C}_{16}\text{H}_{20}\text{Cu}_4\text{N}_8\text{S}_4$): C, 27.19; H, 2.85; N, 15.85; S, 18.14. Found: C, 27.23; H, 2.76; N, 15.75; S, 18.21.

Anal. Calc. for Cu_8NC ($\text{C}_{32}\text{H}_{56}\text{Cu}_8\text{N}_8\text{S}_8$): C, 29.17; H, 4.28; N, 8.50; S, 19.46. Found: C, 29.22; H, 4.24; N, 8.42; S, 19.52.

ESI-MS of the unreported vinylboronate ester compounds (E-4q**, **E-4v**, **E-4y** and **E-4z**):**

HRMS (ESI) m/z : $[\text{M}+\text{H}]^+$ calcd for $\text{C}_{15}\text{H}_{20}\text{BO}_3^+$ 259.1500; found 259.1471 (**E-4q**).

HRMS (ESI) m/z : $[\text{M}+\text{H}]^+$ calcd for $\text{C}_{17}\text{H}_{21}\text{BNO}_2^+$ 282.1660; found 282.1662 (**E-4v**).

HRMS(ESI) m/z : $[\text{M}+\text{H}]^+$ calcd for $\text{C}_{18}\text{H}_{24}\text{BFeO}_2^+$ 339.1213. found: 339.1229 (**E-4y**).

HRMS (ESI) m/z : $[\text{M}+\text{H}]^+$ calcd for $\text{C}_{28}\text{H}_{37}\text{B}_2\text{O}_4^+$ 459.2872; found 459.2875 (**E-4z**).

Supplementary Figure 11 Morphology and energy dispersive spectroscopy (EDS) mapping results of microcrystal Cu_4NC .

Supplementary Figure 12. Morphology and energy dispersive spectroscopy (EDS) mapping results of microcrystal Cu_8NC .

-30.64

Supplementary Figure 146. $^{11}\text{B}\{^1\text{H}\}$ NMR spectrum of compound 4q.

-31.04

Supplementary Figure 157. $^{11}\text{B}\{^1\text{H}\}$ NMR spectrum of compound 4v.

-30.82

$^{11}\text{B}\{^1\text{H}\}$ NMR (193 MHz, CDCl_3)

Supplementary Figure 164. $^{11}\text{B}\{^1\text{H}\}$ NMR spectrum of compound 4y.

-31.23

$^{11}\text{B}\{^1\text{H}\}$ NMR (193 MHz, CDCl_3)

Supplementary Figure 167. $^{11}\text{B}\{^1\text{H}\}$ NMR spectrum of compound 4z.

Comment 3: The authors claim, in the abstract, that their catalyst exhibit and ultra-high TON of 77786. I don't believe this can be consider as ultra-high if we compared this to other catalytic processes whose TON are of the order of millions. It will be acceptable to say high TON. In addition, the statement that their catalyst is the best so far reported is questionable. The catalytic system given in reference 19 provided TOF numbers nearly 3 times better than those shown by the authors in this article (41743 h^{-1} vs 16286 h^{-1}). Therefore, the sentence on page 7 “the TON values of the reported catalytic systems were very poor” should be removed, and in addition comparison of the TOF numbers with other catalytic systems should be provided.

Response: Thanks for the reviewer's comment. As suggested, we have adjusted the description of TON in the manuscript to make it more reasonable and acceptable. And the sentence on page 7 “the TON values of the reported catalytic systems were very poor” has been removed. To better compare the performance of different catalysts as suggested, the TOF (and TON) of catalytic system composed with microcrystal Cu_4NC and other catalytic systems were summarized in Supplementary Table 3. It is worth noting that the catalytic system composed of microcrystal Cu_4NC can provide both a high TON of 77786 and a high TOF of 16286 h^{-1} . These results have been included in the revised manuscript and SI. Again, thanks to the reviewer very much for bringing this important point and helping us improve the manuscript.

Supplementary Table 3. Comparison of TON and TOF for **microcrystal Cu_4NC** with previous reports for the hydroboration of alkynes.

Entry	Catalyst	TON	TOF/ h^{-1}	Condition	Reference
1	Vanadium(III) catalys	4000	---	Homogeneous, rt, N_2 , 16h	24
2	JNM-4-Ns	41734 ^a	41734	Heterogeneous, rt, air, 20 min	29
3	$\text{Cu}_1\text{-O(I)/CeO}_2$	198	198	Heterogeneous, 90 °C, dried, N_2 , 1h	30
4	Co(OAc)_2 -ligand	18.2	0.76	Homogeneous, rt, dried, N_2 , 24h	31
5	Cu-PC-1	37.2	5.3	Homogeneous, rt, 405 nm, Ar, 7h	32
6	$\text{Cu-CuFe}_2\text{O}_4$	190	23.75	Heterogeneous, rt, air, 8 h	33
7	MgBu_2	12.14	0.67	Homogeneous, 80 °C, dried and N_2 , 18h	34
8	micro copper powder	9.6	0.4	Heterogeneous, rt, Ar, 11h or 24h	35
9	Pd_2dba_2 -ligand	46.5	---	Homogeneous, rt, 0.5h-2.5h	36
10	$\text{R}_3\text{Al-DABCO}$	8.9	4.45	Homogeneous, 110 °C, dried, N_2 , 2h	37
11	$(^{\text{Cy}}\text{APDI})\text{CoCH}_3$	27.67	4.61	Homogeneous, rt, dried, N_2 , 6h	38
12	$[\text{Pd(OAc)}_2]_3$ -ligand	18.6	1.55	Homogeneous, 80 °C, dried, Ar, 12h	39
13	LCuCl	23.5	1.96	Homogeneous, rt, dried, Ar, 12h	40
14	JNM-5	1182	394	Heterogeneous, rt, N_2 , 3 h	41
15	Cu_4NC	3680	736	Heterogeneous, rt, air, 5 h	42
16	Co(acac)_2 -ligand	198	16.5	Homogeneous, rt, N_2 , 12h	43
17	dom -NENU-3_2	99.5	24.88	Heterogeneous, 50 °C, N_2 , 4h	44
18	FeO/MgO	34	1.7	Heterogeneous, 160 °C, Ar, 20 h	45
19	microcrystal Cu_4NC	77786	16286	Heterogeneous, rt, air, 1h	This work

^a represents the absence of the definitive values in the article.

Comment 4: Another relevant point that should be corrected is the fact that in the introduction the authors state that homogeneous transition metal complexes have been used under harsh reaction conditions. However, most of the reports used catalytic conditions at room temperature that are far from “harsh reaction conditions”. The fact that some of these processes are carried out under inert atmosphere cannot be considered as harsh. I encourage the authors to avoid over-emphasizing this type of statements. In addition, a puzzling claim the authors make throughout the manuscript is the heterogeneous character of their catalysts. This does not seem to be the case, as it is clearly demonstrated during the entire work that the species involved are in solution. In fact, they are synthesized in solution, and they can be crystallized. They can even be characterized by NMR or ESI-MS methods. Why do they claim this system is heterogeneous?

Response: Thanks a lot for the reviewer’s comment. Following this suggestion, we have revised the description about reaction conditions of homogeneous transition metal catalysts to make it more reasonable and acceptable.

With regard to the issue of microcrystal Cu_4NC and Cu_8NC as heterogeneous catalysts, we did not make this clear and generated the unnecessary confusion. To further confirm this point, a series of experiments and characterization tests were carried out.

First, taking Cu_4NC as an example, through further research, we found that microcrystal Cu_4NC is completely insoluble in MeCN and H_2O solution, as shown in Supplementary Figures 16 and 17. The microcrystal Cu_4NC has a moderate solubility in DCM, CHCl_3 and DMSO, respectively. So Cu_4NC can be characterized by NMR or ESI-MS in CDCl_3 or DMSO, respectively. The hydroboration catalyzed by microcrystal Cu_4NC reacts in the mixture solvent (MeCN- H_2O). Due to the complete insolubility of microcrystal Cu_4NC in MeCN and H_2O solution, the microcrystal Cu_4NC is considered as heterogeneous catalysts in this reaction system.

Second, the hydroboration reaction catalyzed by microcrystal Cu_4NC was conducted. After the reaction was completed, the reaction mixture was centrifuged at 8000 rpm for 5 minutes. The microcrystal Cu_4NC catalyst was filtered, washed with hexane and then dried in vacuum to recycle the microcrystal Cu_4NC . Recycled microcrystal Cu_4NC catalyst was further characterized by ESI-MS (Supplementary Figure 37), NMR spectroscopy (Supplementary Figure 35), PXRD (Supplementary Figure 31), and UV-vis spectroscopy (Supplementary Figure 33), proving the microcrystal Cu_4NC manifests robustness after catalysis.

Third, we separated the catalyst from the supernatant of the reaction system by filtration, then the reaction solution was characterized by inductively coupled plasma mass spectrometric (ICP-MS) measurement, showing that no Cu^+ ions was found in the reaction solution, suggesting that Cu^+ ions were not leached from the metal cluster catalyst into solution over the course of the reaction.

Fourth, three parallel experiments of hydroboration catalyzed by microcrystal Cu_4NC also were carried out. The parallel catalytic experiments will be stopped at the 10th minute, the 30th minute and the 60th minute. The reaction mixture was centrifuged at 8000 rpm for 5 minutes. The microcrystal Cu_4NC catalyst was filtered, washed with hexane and then dried in vacuum to recycle the microcrystal Cu_4NC . Recycled microcrystal Cu_4NC catalyst was further characterized by PXRD (Supplementary Figure 93). It showed that the characteristic peaks of the catalyzed microcrystal Cu_4NC at different time (the 10th minute, the 30th minute and the 60th minute) matched well with the characteristic peaks of the new microcrystal Cu_4NC , confirming the structure of the recycled catalyst were maintained.

Finally, the recycled microcrystal Cu_4NC catalyst was used again in the next cycle. In this way,

the hydroboration reaction of alkynes was conducted for 5 cycles with the microcrystal Cu_4NC catalyst (Supplementary Figure 91).

Similarly, the research found that the microcrystal Cu_8NC can dissolve in DCM , CHCl_3 and DMSO , respectively. So Cu_8NC can be also characterized by NMR or ESI-MS in CDCl_3 or DMSO , respectively. Same as microcrystal Cu_4NC , the microcrystal Cu_8NC is also completely insoluble in H_2O and MeCN solution (Supplementary Figures 21 and 22). Since this hydroboration reaction works in the mixture solvent ($\text{MeCN-H}_2\text{O}$), the microcrystal Cu_8NC was also considered as heterogeneous catalysts in hydroboration. To further support this thesis, we have also made corresponding experiments similar to those of the microcrystal Cu_4NC catalyst.

All of these observations and experimental results suggested that microcrystal Cu_4NC and Cu_8NC can be considered as heterogeneous catalysts in the catalytic system of hydroboration. These explanations and results have been included in the revised manuscript.

Supplementary Figure 16. ^1H NMR spectrum of microcrystal Cu_4NC in CD_3CN . No characteristic peaks of Cu_4NC are found in ^1H NMR spectrum, proving that microcrystal Cu_4NC is completely insoluble in MeCN solution.

Supplementary Figure 17. ^1H NMR spectrum of microcrystal Cu_4NC in D_2O . No characteristic peaks of Cu_4NC are found in ^1H NMR spectrum, proving that microcrystal Cu_4NC is completely insoluble in H_2O solution.

Supplementary Figure 93. PXRD patterns of microcrystal Cu_4NC before catalysis and after the 10th minute, the 30th minute and the 60th minute catalysis.

Supplementary Figure 91. Recyclability of the microcrystal Cu_4NC catalyzed hydroboration reaction in term of yield (1 h).

Supplementary Figure 21. ^1H NMR spectrum of microcrystal Cu_8NC in CD_3CN . No characteristic peaks of Cu_8NC are found in ^1H NMR spectrum, proving that microcrystal Cu_8NC is completely insoluble in MeCN solution.

Supplementary Figure 22. ^1H NMR spectrum of microcrystal Cu_8NC in D_2O . No characteristic peaks of Cu_8NC are found in ^1H NMR spectrum, proving that microcrystal Cu_8NC is completely insoluble in H_2O solution.

Comment 5: Page 5: I don't understand the phrase "flexible and reversible N-Cu coordinate bonds and stable S-Cu covalent bonds". What is the meaning of coordinate bond? I don't see the difference between Cu-S and Cu-N bonds which very likely are both covalent in nature. In addition, the authors have coined or used the term "dynamically regulating dual-catalytic site". The authors insist on the synergistic effect between multiple metallic atoms. In their case, what is the basis of such synergy? From my view this is just metal-ligand and metal-metal cooperativity in which, from one side, the Cu-N bond cooperates or assist in the deprotonation of H_2O , and the nearby copper atoms facilitate the insertion of the alkyne into the boryl fragment. Some references to this type of cooperation should be given.

Response: Thank the reviewer so much for the comment. Cu-S and Cu-N bonds have the same process of forming bonds. S atom and N atom as electron donors donates electrons to electron acceptors of copper atom of cluster cores to form covalent bonds. As suggested, we have corrected this sentence in the revised manuscript.

The description on "the synergistic effect between multiple metallic atoms" is inappropriate, which has been revised in the manuscript as suggested. The cluster cores were constructed through metal-metal bond between different metal atoms, which is beneficial to regulating catalytic activity. The corresponding references have been cited in the revised manuscript.

The basis of dual site synergy was proposed for three reasons.

First, by SCXRD analysis, in the Cu_4NC unit and Cu_8NC unit, the Cu-Cu distance is 2.662 Å and 2.663 Å, respectively, which are shorter than that of dicopper (I) μ -boryl complexes (*Chem. Sci.* 2022, 13, 6619–6625), providing a foundation for the interaction between hydroxide ions and two

adjacent Cu atoms to form intermediate A or intermediate B.

Second, compared with Cu-S bonds, Cu-N bonds are weaker and easier to dissociate, decreasing the steric hindrance around the copper catalytic sites and exposing them to the substrate. This is supported by the following experiments, DFT calculations and *in situ* characterizations.

a. Through DFT-calculated binding energies of Cu-S and Cu-N bonds in Cu₄NC and Cu₈NC, we determined that the binding energies of Cu-S bonds are -20.7 kcal/mol and -24.2 kcal/mol, respectively. These values are lower than the binding energies of Cu-N bonds, which are -15.0 kcal/mol and -17.6 kcal/mol (Supplementary Figure 71). This indicates that the Cu-S bond is stronger than the Cu-N bond. Our findings align with previously reported results in the literature (*Chem. Rev.* 2024, 124, 7262-7378), which also demonstrates that the Cu-N bond is weaker compared to the Cu-S bond.

b. Taking Cu₄NC as an example, under basic condition, Cu-N bonds of the Cu₄NC are easily dissociated to form intermediate A. Mechanistic studies and density functional theory calculations illustrate that the two flexible Cu-N bonds of the copper NCs could dynamically dissociate from the copper cluster core to generate key intermediates A in a basic proton environment. As shown in Supplementary Figure 72, the energy barrier leading to intermediate A is only 2.9 kcal/mol and the energy of intermediate A is -19.3 kcal/mol, indicating that this process is both kinetically and thermodynamically favorable. In addition, the hydroboration reaction was monitored by the *in situ* Raman spectroscopy. We observed that the Raman peak intensities of Cu-N bonds gradually decreased as the reaction proceeds (Supplementary Figure 82), illustrating that the Cu-N bonds maybe undergo dissociation. The experimental result is consistent with the theoretical calculation, which further illustrated that Cu-N bonds have the flexible character in this system.

Third, after the catalytic reactions were completed, microcrystal Cu₄NC and Cu₈NC were centrifuged, filtered, washed and then dried to recycle the microcrystal Cu₄NC and Cu₈NC. Recycled microcrystal Cu₄NC and Cu₈NC catalyst were further characterized by ESI-MS (Supplementary Figure 37), NMR spectroscopy (Supplementary Figure 35), PXRD (Supplementary Figure 31), and UV-vis spectroscopy (Supplementary Figure 33), proving the microcrystal Cu₄NC and Cu₈NC manifests robustness after catalysis. The result illustrated that the dissociated Cu-N bonds could be re-constructed, showing that Cu-N bonds have the reversible character.

In summary, in this catalytic system, the flexible Cu-N bonds can be dissociated to expose catalytic sites, the shorter Cu-Cu distance can facilitate the interaction between hydroxide ions and two adjacent Cu atoms to form intermediate A or B, and the dissociated Cu-N bonds can be re-constructed to maintain the integrity of the catalyst structures. These experiments, DFT calculations and *in situ* characterizations support that Cu₄NC and Cu₈NC catalysts have the dynamically regulating dual-catalytic site.

Indeed, just as the reviewer 2 commented, metal-ligand and metal-metal cooperativity exist in the catalytic process. As shown in Supplementary Figure 72 and Figure 5, under basic condition, Cu-N bonds undergo dissociation and cooperate the insertion of hydroxide ions to form intermediate A. The excess K₂CO₃ can promote deprotonation of H₂O and generate a large amount of hydroxide ions, which is enough for the hydroboration reaction. However, under neutral condition, the reaction catalyzed by microcrystal Cu₄NC catalyst did not work (table 1 Entry 20), showing deprotonation of H₂O depended on excess K₂CO₃ rather than Cu-N bond cooperated in

the deprotonation of H₂O. And the active intermediate 3 Cu₄NC-BPin with Cu-Cu dual site also can adsorb and activate phenylacetylene molecule through feature of Lewis acidity, and then fully utilizes the Cu-Cu dual site synergistic effect to facilitate the insertion of the alkyne into the boryl fragment, forming the active intermediate 5. In these two processes, metal-ligand and metal-metal cooperativity surely exist, same as the reviewer's viewpoint.

Supplementary Figure 71. DFT-calculated the binding energies of Cu-S bonds and Cu-N bonds in Cu₄NC and Cu₈NC (ΔG in kcal/mol).

Supplementary Figure 72. DFT-computed Gibbs free energy profile for formation of Intermediate A. The relative Gibbs energies and electronic energies (in parentheses) are given in kcal/mol.

Supplementary Figure 82. Time-dependent *in situ* Raman spectra of the hydroboration reaction catalyzed by microcrystal Cu_4NC catalyst (from bottom to top: 1min, 10min, 15min, 20min). Reaction conditions: Phenylacetylene **1** (0.2 mmol, 1.0 equiv.), B_2Pin_2 **2** (0.44 mmol, 1.0 equiv.), Cu_4NC catal. (0.004 mmol) and K_2CO_3 (0.44 mmol, 1.0 equiv.) were added to the mixture solvent (2.0 mL, MeCN/ H_2O 4:1) under air atmosphere at room temperature and allowed to react for 1 h.

Figure R1. DFT-calculated Gibbs free energy profile (ΔG in kcal/mol) for the ways of deprotonation of H_2O (KOH comes from that excess K_2CO_3 can promote deprotonation of H_2O and generate a large amount of hydroxide ions.).

Fig. 5 | Theoretical calculations and proposed reaction mechanism. **a** DFT-calculated Gibbs free energy profile (ΔG in kcal/mol) for the hydroboration catalyzed by Cu_4NC . **b** The catalytic mechanism for the formation of the vinylboronate esters catalyzed by Cu_4NC .

Comment 6: With regard to the mechanism, the discussion in some points are confusing. First, the proposed mechanism in Figure 5 is very difficult to follow. The structures obtained by DFT calculation are too small. I encourage the authors to use Figure 64 in the SI that is much easier to follow and to send Figure 5 in the main text to the SI. Concerning the use of D₂O for the deuteration of phenylacetylene (pages 12-13), the authors state first that copper acetylides might be involved considering the incorporation of deuterium in the alkyne, but afterwards they proved that copper is not necessary for this exchange, contradicting the presence of copper acetylides. This should be clarified.

Response: Thanks for the reviewer's comment. In order to present the mechanism more clearly, as suggested, we have revised Figure 5 in the main text. The original Figure 5 in the main text has also been send to the supporting information.

As **Reviewer 2** suggested, we have clarified the deuteration of phenylacetylene in the revised manuscript.

In Figure 4b, when H₂O was replaced with D₂O, two kinds of deuterated products were observed. The D_a atom was introduced to the deuterated products during the elimination reaction process in hydroboration. However, the manner in which the D_b atom was introduced to the deuterated products remains unknown. There are two hypotheses here. The first hypothesis about the source of the β-site D_b, is that copper catalysts easily activate the C≡C bond of alkynes to form an intermediate π-metal-alkyne complex, which enhances the acidity of the alkyne proton, facilitating the formation of copper acetylides under basic condition. In this case, copper acetylides as active species participated in hydroboration. During the elimination reaction, deuterated products containing D_b atoms were easily obtained. The second hypothesis about the source of the β-site D_b, is that the partial phenylacetylene was directly deuterated to phenylacetylene-d₁ under the action of strong base K₂CO₃ and protic solvent H₂O.

To confirm which of the above hypotheses is the source of the β-site D_b, a series of control experiments were carried out. Using CH₃CN/D₂O as the mixture solvent and without B₂Pin₂, phenylacetylene (**1a**) and K₂CO₃ were added to the reaction system catalyzed by microcrystal Cu₄NC, providing phenylacetylene-d₁ in 96% yield (Supplementary Figures 60-63). Another control experiment was performed. Without adding microcrystal Cu₄NC catalyst, phenylacetylene (**1a**) and K₂CO₃ were directly added to the CH₃CN/D₂O mixture solvent under the reaction conditions, affording phenylacetylene-d₁ in 94% yield (Supplementary Figures 60, 67-69). These results indicated that the deuteration of phenylacetylene was independent of microcrystal Cu₄NC. In other words, the second hypothesis is more reasonable, in which the partial phenylacetylene was directly deuterated to phenylacetylene-d₁ under strong base K₂CO₃. These results have been included in the revised manuscript. Again, thank the reviewer a lot for bringing this important point and helping us improve the manuscript.

Fig. 5 | Theoretical calculations and proposed reaction mechanism. **a** DFT-calculated Gibbs free energy profile (ΔG in kcal/mol) for the hydroboration catalyzed by Cu_4NC . **b** The catalytic mechanism for the formation of the vinylboronate esters catalyzed by Cu_4NC .

Comment 7: Regarding the mechanistic proposal, DFT calculations have been carried out. Although the energy barriers are somehow consistent with the experiments there is one step that is missing in the calculation. The calculated energy for the conversion of Cu₄NC or Cu₈NC into A or B should be provided, since this is the key step that proves the participation of the Cu-N bonds in the catalytic cycle. What are the energy barriers for this process? In addition, a simple experiment should be carried out to prove this step by reacting Cu₄NC or Cu₈NC with H₂O or D₂O. How important is the presence of the N-H generated in this step? Regardless of the isolation, an easy experiment such as an IR spectrum should shed light about the presence of a hydroxide group (with H₂O and D₂O). Additionally, if complexes A or B are sufficiently stable an experiment using a thione without N-H moieties should be carried out to prove its cooperative effect. The difference in energy between the calculated profiles for Cu₄NC and Cu₈NC is 1.8 kcal mol⁻¹. Given this very small energy difference, the authors should avoid terms expressions such as ‘clearly illustrated that Cu₄ largely exceeds Cu₈. In fact, a different functional or basis set could easily invert these energies, making the Cu₈NC more favorable. Can the authors comment on this?

Response: Thank you for your comments.

In term of the calculated energy for the conversion of Cu₄NC or Cu₈NC into A or B, we have done theoretical calculations again.

In our experiment, we used K₂CO₃ as the base. Since K₂CO₃ can react with water to form KOH, we simplified our calculations by using KOH as the base. In the Cu₄NC, KOH undergoes a ligand substitution reaction to form intermediate A via an association mechanism. As shown in Supplementary Figure 72, the energy barrier leading to intermediate A is only 2.9 kcal/mol and the energy of intermediate A is -19.3 kcal/mol, indicating that this process is both kinetically and thermodynamically favorable. This suggests that the formation of intermediate A from the Cu₄NC is feasible and that intermediate A serves as a suitable starting point for this reaction. Similarly, formation of Cu₈NC-OH intermediate B from Cu₈NC is also kinetically and thermodynamically favorable, suggesting intermediate B is a suitable starting point (Supplementary Figure 73).

As stated in the previous response, in terms of the deprotonation of H₂O, the designed experiment was carried out. The hydroboration reactions catalyzed by microcrystal Cu clusters were worked in alkaline mixture solution (MeCN-H₂O) (Table 1 Entry 1). The excess K₂CO₃ can promote deprotonation of H₂O and generate hydroxide ions, which is enough for the catalyzed hydroboration. The control experiment was carried out to prove this step.

Without adding K₂CO₃ in the catalytic system, the reaction did not work (table 1 Entry 20), demonstrating deprotonation of H₂O depended on excess K₂CO₃ rather than Cu-N bond cooperated in the deprotonation of H₂O. At the same time, we also calculated the deprotonation of H₂O by Cu-N, however, it was found that it is energetically unfavorable (Figure R1). Instead, excess K₂CO₃ can promote deprotonation of H₂O and generate a large amount of hydroxide ions, which can facilitate the dissociation of N from Cu, providing a more stable intermediate Cu₄NC-KOH (Figure R1). Combining theoretical calculations and experimental research, the deprotonation of H₂O in this reaction system depended on excess K₂CO₃. The simple experiment by reacting Cu₄NC with H₂O cannot promote the deprotonation reaction of H₂O without adding K₂CO₃.

In-depth understanding of the mechanism of hydroboration catalyzed by microcrystal Cu₄NC or Cu₈NC was performed by *in situ* Fourier transform infrared (FT-IR) spectroscopy analysis and *in*

situ Raman spectroscopy analysis (Supplementary Figures 82-86). As shown in Supplementary Figures 84 and 86, the top view of the three-dimensional surface of *in situ* the FT-IR spectroscopy could provide the selected individual spectra of key signals for hydroboration reaction process catalyzed by microcrystal Cu₄NC or Cu₈NC catalyst. The IR peak intensities of B-B bonds (1124 cm⁻¹ for stretching vibration) and B-O bonds (1284 cm⁻¹ for stretching vibration) in B₂Pin₂ decrease gradually as the reaction progress (*Nat Commun.* 2023, 14, 6957; *RSC Adv.* 2023, 13, 16907–16914; *Carbon.* 2013, 54, 208–214), illustrating the progressive consumption of the B₂Pin₂ substrate. Meanwhile, the new IR peaks appear at 1148 cm⁻¹, 1352 cm⁻¹ and 1624 cm⁻¹, which can be assigned to the B-C bonds, B-O bonds and C=C bonds stretching vibrations of the vinylboronate ester target product, respectively (*Nat Commun.* 2023, 14, 6957; *RSC Adv.* 2023, 13, 16907–16914; *Carbon.* 2013, 54, 208–214). The IR peak intensities of the B-C bonds, B-O bonds and C=C bonds in target product grow gradually with reaction time. Compared the *in situ* FT-IR spectra of microcrystal Cu₄NC catalytic systems and microcrystal Cu₈NC catalytic systems, it further supported that microcrystal Cu₄NC is the stronger in catalytic activity than that of microcrystal Cu₈NC.

As shown in Supplementary Figure 82, the characteristic Raman peak at 690 cm⁻¹ was attributed to the stretching vibration peak of Cu-N bond of microcrystal Cu₄NC (*J. Am. Chem. Soc.* 2022, 144, 21046–21055; *J. Mol. Struct.* 2018, 1165, 79–83). It was observed that the Raman peak intensities of Cu-N bond gradually decrease as the reaction proceeds, illustrating that Cu-N bonds maybe dissociate during hydroboration reaction process. These changes in Raman spectra were attributed to the interaction between the substrates and the microcrystal Cu₄NC catalyst, which further support that the flexible Cu-N bonds of the copper NCs could dynamically dissociate from the copper cluster core to generate key intermediates.

As discussed above, in hydroboration, deprotonation of H₂O depended on excess K₂CO₃ rather than Cu-N bond cooperated in the deprotonation of H₂O (Table 1 Entries 1 and 20). At the same time, the microcrystal Cu₄NC achieved the TOF value up to 16286 h⁻¹ in the hydroboration reaction (Supplementary Table 1). Based on the actual situation, the complexes A is not stable and could not be isolated. As suggested by **Reviewer 2**, the NHT ligand was replaced by the cyclopentanethione (CAS: 38744-78-4) ligand without N-H moieties to construct the copper clusters. The cyclopentanethione is a monodentate ligand. Despite trying various approaches, we still did not achieve the desired structures of copper clusters. However, the above series of experiments have been added to illustrate this point.

Thank you for your insightful comment regarding the energy difference between the calculated profiles for Cu₄NC and Cu₈NC. We acknowledge that the energy difference of 1.8 kcal/mol is indeed small and that the choice of functional or basis set could potentially affect the relative stabilities of these species. As shown in Table R1, we tested some popular functionals in Table R1, the golden standard M06 functional enlarge the energy difference to 12.8 kcal/mol, (7.9 for dispersion corrected M06). B3LYP functional gives similar energy difference with PBE0 functional. In terms of basis set, 6-311+G** functional gives a larger difference of barrier difference. The above method can well fit our experiment. However, our previous calculations suggested that the functional PBE0 and the basis set def2-SVP are appropriate choices, as indicated by our earlier work which fit very well with the crystal structure (*J. Am. Chem. Soc.* 2023, 145, 6166–6176; *J. Am. Chem. Soc.* 2023, 145, 22310–22316). This combination provides

a balance between accuracy and computational speed, particularly for large systems such as the metal clusters studied in our work. Therefore, we employed PBE0/def2-SVP in the current paper. We appreciate your suggestion and will include a discussion on the potential impact of different functionals or basis sets on our results (Table R1). This will provide a more balanced and thorough interpretation of the data.

Table R1. Energy difference between the barrier of Cu₄NC and Cu₈NC tested for different functionals and basis set.

Method	Energy difference of barrier
PBE0-D3/def2svp (This work)	1.8 kcal/mol
M06-D3/def2svp	7.9 kcal/mol
M06/def2svp	12.8 kcal/mol
B3LYP-D3/def2svp	1.6 kcal/mol
PBE0/6-311+G**	8.4 kcal/mol

Supplementary Figure 72. DFT-computed Gibbs free energy profile for formation of Intermediate A. The relative Gibbs energies and electronic energies (in parentheses) are given in kcal/mol.

Supplementary Figure 73. DFT-computed Gibbs free energy profile for formation of Intermediate B. The relative Gibbs energies and electronic energies (in parentheses) are given in kcal/mol.

Figure R1. DFT-calculated Gibbs free energy profile (ΔG in kcal/mol) for the ways of deprotonation of H_2O (KOH comes from that excess K_2CO_3 can promote deprotonation of H_2O and generate a large amount of hydroxide ions.).

The *in situ* Raman Experiment

Supplementary Figure 82. Time-dependent *in situ* Raman spectra of the hydroboration reaction catalyzed by microcrystal Cu_4NC catalyst (from bottom to top: 1min, 10min, 15min, 20min). Reaction conditions: Phenylacetylene **1** (0.2 mmol, 1.0 equiv.), B_2Pin_2 **2** (0.44 mmol, 1.0 equiv.), Cu_4NC catal. (0.004 mmol) and K_2CO_3 (0.44 mmol, 1.0 equiv.) were added to the mixture solvent (2.0 mL, MeCN/ H_2O 4:1) under air atmosphere at room temperature and allowed to react for 1 h.

Supplementary Figure 83. Time-dependent *in situ* FT-IR spectra of the hydroboration reaction catalyzed by microcrystal Cu_4NC catalyst. Reaction conditions: Phenylacetylene **1** (0.2 mmol, 1.0 equiv.), B_2Pin_2 **2** (0.44 mmol, 2.2 equiv.), Cu_4NC catal. (0.004 mmol, 2.0 mol%) and K_2CO_3 (0.44 mmol, 2.2 equiv.) were added to the mixture solvent (2.0 mL, MeCN/ H_2O 4:1) under air atmosphere at room temperature and allowed to react for 1 h.

Supplementary Figure 84. The top view of the three-dimensional surface of the *in situ* FTIR spectra for hydroboration process catalyzed by microcrystal Cu_4NC catalyst.

Supplementary Figure 85. Time-dependent *in situ* FT-IR spectra of the hydroboration reaction catalyzed by microcrystal Cu_8NC catalyst. Reaction conditions: Phenylacetylene 1 (0.2 mmol, 1.0 equiv.), B_2Pin_2 2 (0.44 mmol, 2.2 equiv.), Cu_8NC catal. (0.004 mmol, 2.0 mol%) and K_2CO_3 (0.44 mmol, 2.2 equiv.) were added to the mixture solvent (2.0 mL, MeCN/ H_2O 4:1) under air atmosphere at room temperature and allowed to react for 1 h.

Supplementary Figure 86. The top view of the three-dimensional surface of the *in situ* FTIR spectra for hydroboration process catalyzed by microcrystal Cu_8NC catalyst.

Comment 8: On page 8, the following can be read: ‘The result further proved that the structure of DRDS Cu_4NC and the distribution of dual-catalytic sites are more plausible than those of the other DRDS copper NCs.’ What does this mean? Sentences like these are frequently found in the manuscript and they are somewhat confusing and difficult to understand.

Response: Thank the reviewer so much for the comment. We are sorry for the unclear descriptions. This description, “*the structure of DRDS Cu_4NC and the distribution of dual-catalytic sites are more plausible than those of the other DRDS copper NCs*”, is inappropriate. As Reviewer 2 suggested, we corrected this sentence in the revised manuscript. Compared to that of microcrystal Cu_8NC , the microcrystal Cu_4NC has a higher catalytic activity in hydroboration, originating from that microcrystal Cu_4NC has a lower activation energy toward hydroboration (Figure 5 and Supplementary Figures 74-79). We have carefully examined the manuscript. The corresponding situation has been corrected in the revised manuscript.

Comment 9: 1,3,5-trimethoxybenzene is utilized as an internal standard. Copper boryl species have been demonstrated to cleave aryl ethers (Tetrahedron Lett. 2017, 58, 17-20). Do the authors add the standard before or after the catalysis? If before, do they see any species that could come from trimethoxybenzene reactivity?

Response: A very good point! Thank reviewer so much for this important comment. In the hydroboration reaction, we added 1,3,5-trimethoxybenzene as the standard after the catalysis. Thank you for recommending this literature (Tetrahedron Lett. 2017, 58, 17–20). Indeed, this literature reported that copper boryl reagents only could cleave aryl allyl ethers. 1,3,5-trimethoxybenzene belongs to aryl alkyl ether. In this literature, the aryl alkyl ethers have not been demonstrated to be cleaved. The copper catalytic system of this literature is different from our copper cluster catalytic system.

To further prove that 1,3,5-trimethoxybenzene can be utilized as an internal standard, the control experiment was carried out. The 1,3,5-trimethoxybenzene replaced phenylacetylene as substrate was added into the copper cluster catalytic system. When the reaction time reaches 1 hour, the mesitylene as the standard was added into the reaction system. The reaction was monitored by the ^1H NMR spectrum. The result showed that the content of 1,3,5-trimethoxybenzene did not decrease (Supplementary Figures 87-90). It further illustrated that 1,3,5-trimethoxybenzene is an inert substrate in our catalytic system. The experiment showed that 1,3,5-trimethoxybenzene can be utilized as an internal standard in our catalytic system. These results have been included in the revised manuscript.

Supplementary Figure 87. The *in situ* ^1H NMR spectrum of deprotection of 1,3,5-trimethoxybenzene catalyzed by microcrystal Cu_4NC catalyst.

Supplementary Figure 88. The *in situ* ¹H NMR spectrum of deprotection of 1,3,5-trimethoxybenzene catalyzed by microcrystal Cu_4NC catalyst. Top: the control group. Bottom: the experimental group.

Supplementary Figure 89. The *in situ* ¹H NMR spectrum of deprotection of 1,3,5-trimethoxybenzene catalyzed by microcrystal Cu_8NC catalyst.

Supplementary Figure 90. The *in situ* 1H NMR spectrum of deprotection of 1,3,5-trimethoxybenzene catalyzed by microcrystal Cu_8NC catalyst. Top: the control group. Bottom: the experimental group.

Comment 10: During the submission of this article a closely related system has been reported by Bakr, Rueping and co-workers (J. Am. Chem. Soc. 2024, 146, 16295–16305). I believe that the mechanistic difference between this system and the one reported herein justifies the inclusion of this reference in the main text.

Response: Thanks a lot for the reviewer's suggestion. As suggested, we have added the paper in the revised manuscript.

Comment 11: On page 14, the authors claim that 'The S-Cu covalent bonds were always very stable during this process to keep catalyst integrity and efficient catalytic activity'. How do they know this? What experiments have been carried out to demonstrate this claim?

Response: Thanks for the reviewer's comment. This sentence has been appropriately revised to "Compared to the Cu-N bonds, the Cu-S bonds were stable during this process to keep catalyst integrity and efficient catalytic activity". And the explanations have been added in the revised manuscript.

Compared with Cu-N bonds, the Cu-S bonds are stable. Through DFT-calculated binding energies of Cu-S and Cu-N bonds in Cu_4NC and Cu_8NC , we determined that the binding energies of Cu-S bonds are -20.7 kcal/mol and -24.2 kcal/mol, respectively. These values are lower than the binding energies of Cu-N bonds, which are -15.0 kcal/mol and -17.6 kcal/mol (Supplementary Figure 71). This indicates that the Cu-S bond is stronger than the Cu-N bond. Our findings align with previously reported results in the literature (Chem. Rev. 2024, 124, 7262-7378), which also demonstrates that the Cu-N bond is weaker compared to the Cu-S bond.

Mechanistic studies and density functional theory calculations illustrate that the Cu-N bonds of the copper NCs could dissociate from the copper cluster core to generate key intermediates A in a

basic proton environment. As shown in Supplementary Figure 72, the energy barrier leading to intermediate A is only 2.9 kcal/mol and the energy of intermediate A is -19.3 kcal/mol, indicating that this process is both kinetically and thermodynamically favorable. In this process, compared with Cu-N bonds, the Cu-S bonds were stable.

Supplementary Figure 71. DFT-calculated binding energies of Cu-S bonds and Cu-N bonds in Cu_4NC and Cu_8NC (ΔG in kcal/mol).

Supplementary Figure 72. DFT-computed Gibbs free energy profile for formation of Intermediate A. The relative Gibbs energies and electronic energies (in parentheses) are given in kcal/mol.

Comment 12: Page 15: In a couple of occasions the authors mention that B₂pin₂ is the nucleophile, whereas intermediate complex(I) is the electrophile. B₂pin₂ is not a nucleophile, but an electrophile. Intermediate complex (I) (do they mean complex A?) is the nucleophile.

Response: Thank reviewer a lot for this very important reminder. The B₂Pin₂ is an electrophile in the catalytic system. Intermediate complex (I) in Figure 5b is complex A, which is the nucleophile. We have revised the manuscript as suggested.

Comment 13: In the Supporting Information the ¹³C NMR spectra provided are ¹H decoupled. Therefore, this should be given as ¹³C{¹H}. In addition, no ¹¹B{¹H} NMR is provided for the new compounds.

Response: Thanks for the reviewer's comment. We have revised the ¹³C NMR spectra of compounds as suggested. The new vinylboronate ester compounds (**E-4q**, **E-4v**, **E-4y** and **E-4z**) have been also characterized by HRMS(ESI) and ¹¹B spectroscopy (Supplementary Figures 146, 157, 164 and 167). These results have been included in the revised manuscript.

ESI-MS of the unreported vinylboronate ester compounds (E-4q, E-4v, E-4y and E-4z):

HRMS (ESI) m/z: [M+H]⁺ calcd for C₁₅H₂₀BO₃⁺ 259.1500; found 259.1471 (**E-4q**).

HRMS (ESI) m/z: [M+H]⁺ calcd for C₁₇H₂₁BNO₂⁺ 282.1660; found 282.1662 (**E-4v**).

HRMS(ESI) m/z: [M+H]⁺ calcd for C₁₈H₂₄BFeO₂⁺ 339.1213; found: 339.1229 (**E-4y**).

HRMS (ESI) m/z: [M+H]⁺ calcd for C₂₈H₃₇B₂O₄⁺ 459.2872; found 459.2875 (**E-4z**).

Supplementary Figure 146. ¹¹B{¹H} NMR spectrum of compound **4q**.

-31.04

$^{11}\text{B}\{^1\text{H}\}$ NMR (193 MHz, CDCl_3)

Supplementary Figure 157. $^{11}\text{B}\{^1\text{H}\}$ NMR spectrum of compound 4v.

-30.82

$^{11}\text{B}\{^1\text{H}\}$ NMR (193 MHz, CDCl_3)

Supplementary Figure 164. $^{11}\text{B}\{^1\text{H}\}$ NMR spectrum of compound 4y.

Supplementary Figure 167. $^{11}\text{B}\{^1\text{H}\}$ NMR spectrum of compound 4z.**Reviewer #3:**

Remarks: I co-reviewed this manuscript with one of the reviewers who provided the listed reports. This is part of the Nature Communications initiative to facilitate training in peer review and to provide appropriate recognition for Early Career Researchers who co-review manuscripts.

Response: Thank the reviewer 3 so much for the comment. Thank you very much for helping us improve the manuscript.

Reviewer #4:

Remarks: In this report by Huang, Wang, and Zang 2 copper complexes with N-heterocyclic thione ligands are synthesised and their ability to undertake catalytic hydroboration of alkynes with a proton source and a diborane is explored experimentally and computationally. The reaction belongs to a class of extremely widely reported hydroborations with copper which have been well-known for decades (see section 3 of <https://pubs.rsc.org/en/content/articlelanding/2018/cs/c7cs00816c>), and represents at best a minor advance in this field. The work is presented reasonably well, although a number of minor typos are present and some schemes are hard to follow (vide infra). The authors have performed a decently sized scope table, although only very limited examples of non-aromatic substituted alkynes are presented. I have no doubt that the reaction as described does work, and that it represents a useful advance in the area. Moreover, the compounds the authors claim are characterised appropriately - although I can't see evidence of bulk purity for the catalysts (CHN or

calibrated NMR showing all compound added is in solution and not NMR silent). The larger concern, however, is that the work has serious flaws in its conclusions that would need to be remedied before publication.

Response: We thank the **Reviewer 4** for these positive suggestions and the following constructive suggestions to improve our manuscript. We have double-checked the manuscript and corrected the minor typos. The evidence of bulk purity for the catalysts have been added in the revised manuscript. We have addressed the following issues as suggested.

In our work, we have made the following progress in related field.

First, as mentioned by the **Reviewer 4**, indeed, hydroboration catalyzed by copper has been reported. However, the reported catalytic systems usually are restricted by specific reaction conditions (a dry or inert atmosphere, etc). And turnover number (TON) of the reported catalytic system usually exhibit less than 5000. But the newly microcrystal Cu_4NC catalyst displayed outstanding catalytic performance for the hydroboration of alkynes, achieving excellent regio-, stereo- and chemoselectivity (up to 100%) and an outstanding yield (up to 99%) under mild conditions (air atmosphere and room temperature). Importantly, microcrystal Cu_4NC achieves the high TON value up to 77786, which was higher than those of reported catalysts (Supplementary Table 3). It is worth noting that the mechanism of hydroboration has also been studied by SCXRD analysis, theoretical calculations, *in situ* FT-IR spectroscopy, *in situ* Raman spectroscopy and control experiments, further illustrating the structure-activity relationship of catalysts. These results represent that microcrystal Cu_4NC catalyst further promotes development of hydroboration reaction.

Second, the progress of hydroboration reaction mainly originated from development of new copper cluster catalysts. Metal nanoclusters (NCs) are representative nanostructured material catalysts (*Chem. Rev.* 2020, 120, 2, 526–622), which have the advantage of atomically precise crystallographic structures, high surface-to-volume ratio, molecular purity, abundant active sites. However, it is still a challenge that the dynamic coordination properties of the cluster shell ligands can be effectively utilized to achieve precise dual-catalytic site control while maintaining the integral cluster structure. In previous catalytic researches, the ligands of NCs always were removed to expose more catalytic sites, leading to that cluster structures are changed and clusters are aggregated, which decreased their structural uniformity. At the same time, copper NCs with long-term stability in organic reaction systems remain unrealized, which restricts the applicability of copper NCs in catalysis. We developed two dynamically regulating dual-catalytic site copper clusters, Cu_4NC and Cu_8NC supported by an N-heterocyclic thione (NHT) ligand in the gram scale, which remain still high stability after organic catalysis. This work promotes the combination of cluster chemistry and organic catalysis and provide a novel reference idea for enhancing the activity of cluster catalysts.

As suggested, the microcrystal copper clusters have been characterized by SCXRD (Supplementary Figures 5 and 6), elemental analysis, PXRD (Supplementary Figures 7 and 8), EDS-Mapping (Supplementary Figures 11 and 12), ESI-MS (Supplementary Figures 9 and 10), NMR spectroscopy (Supplementary Figures 13-22, 27-30) and UV-vis spectroscopy (Supplementary Figures 23-26) to prove the purity and physicochemical properties of the catalysts. The evidence of bulk purity for the catalysts have been deeply discussed on Comment 2. These results have been included in the revised manuscript.

Elemental analysis of Cu₄NC and Cu₈NC:

Anal. Calc. for Cu₄NC (C₁₆H₂₀Cu₄N₈S₄): C, 27.19; H, 2.85; N, 15.85; S, 18.14. Found: C, 27.23; H, 2.76; N, 15.75; S, 18.21.

Anal. Calc. for Cu₈NC (C₃₂H₅₆Cu₈N₈S₈): C, 29.17; H, 4.28; N, 8.50; S, 19.46. Found: C, 29.22; H, 4.24; N, 8.42; S, 19.52.

In term of the NMR measurement, through further research, we found that microcrystal Cu₄NC and Cu₈NC are completely insoluble in MeCN and H₂O solution (Supplementary Figures 16, 17, 21 and 22). The microcrystal Cu₄NC and Cu₈NC have the moderate solubility in DCM and CHCl₃, respectively. So Cu₄NC and Cu₈NC can be characterized by NMR in CDCl₃. The hydroboration catalyzed by microcrystal Cu₄NC or Cu₈NC reacts in the mixture solvent (MeCN-H₂O). Due to the complete insolubility of microcrystal Cu₄NC and Cu₈NC in MeCN and H₂O solution, the microcrystal Cu₄NC and Cu₈NC is considered as heterogeneous catalysts in this reaction system, which has been closely illustrated on Comment 2 and Comment 3.

To make sure the change of the microcrystal Cu₄NC when the substrates are added to the catalysts in stoichiometric studies, the NMR experiments were carried out. As mentioned above, the Cu₄NC have a moderate solubility in CHCl₃. As reviewer suggested, 1.0 eq. microcrystal Cu₄NC, 2.0 eq. phenylacetylene, 4.0 eq. B₂Pin₂ and 4.0 eq. K₂CO₃ were added into the mixture solvent (CDCl₃-D₂O). The *in situ* ¹H NMR was monitored at the 5th minute, the 30th minute and the 60th minute, showing that characteristic peaks of Cu₄NC have no change and the products are not found (Supplementary Figure 94). The other NMR experiment, that 10 mol% microcrystal Cu₄NC, 1.0 eq. phenylacetylene, 2.2 eq. B₂Pin₂ and 2.2 eq. K₂CO₃ were added into the mixture solvent (CD₃CN-D₂O), were carried out and was monitored at the 0 minute, the 3rd minute, the 6th minute, the 10th minute and the 20th minute. It showed that the vinylboronate ester was got with excellent yields. Since Cu₄NC is completely insoluble in MeCN and H₂O, the characteristic peaks of Cu₄NC are not found (Figure R2). It has been closely illustrated on Comment 2

Supplementary Figure 16. ^1H NMR spectrum of microcrystal Cu_4NC in CD_3CN . No characteristic peaks of Cu_4NC are found in ^1H NMR spectrum, proving that microcrystal Cu_4NC is completely insoluble in MeCN solution.

Supplementary Figure 17. ^1H NMR spectrum of microcrystal Cu_4NC in D_2O . No characteristic peaks of Cu_4NC are found in ^1H NMR spectrum, proving that microcrystal Cu_4NC is completely insoluble in H_2O solution.

Supplementary Figure 21. ^1H NMR spectrum of microcrystal Cu_8NC in CD_3CN . No characteristic peaks of Cu_8NC are found in ^1H NMR spectrum, proving that microcrystal Cu_8NC is completely insoluble in MeCN solution.

Supplementary Figure 22. ^1H NMR spectrum of microcrystal Cu_8NC in D_2O . No characteristic peaks of Cu_8NC are found in ^1H NMR spectrum, proving that microcrystal Cu_8NC is completely insoluble in H_2O solution.

Supplementary Figure 94. Time-dependent *in situ* ^1H NMR spectra of the hydroboration reaction catalyzed by microcrystal Cu_4NC catalyst in the mixture solvent ($\text{CDCl}_3\text{-H}_2\text{O}$). Reaction conditions: Phenylacetylene 1 (2.0 equiv.), B_2Pin_2 2 (4.0 equiv.), Cu_4NC catal. (1.0 equiv.) and K_2CO_3 (4.0 equiv.) were added to the mixture solvent (2.0 mL, $\text{CDCl}_3\text{-H}_2\text{O}$) under air atmosphere at room temperature and allowed to react for 1 h. No product was found.

Figure R2. Time-dependent *in situ* ^1H NMR spectra of the hydroboration reaction catalyzed by microcrystal Cu_4NC catalyst in the mixture solvent ($\text{CD}_3\text{CN-H}_2\text{O}$). Reaction conditions: Phenylacetylene 1 (1.0 equiv.), B_2Pin_2 2 (2.2 equiv.), Cu_4NC catal. (10% mol) and K_2CO_3 (2.2 equiv.) were added to the mixture solvent (2.0 mL, $\text{CD}_3\text{CN-H}_2\text{O}$) under air atmosphere at room temperature and allowed to react for 20 min.

Supplementary Figure 11. Morphology and energy dispersive spectroscopy (EDS) mapping results of microcrystal Cu_4NC .

Supplementary Figure 12. Morphology and energy dispersive spectroscopy (EDS) mapping results of microcrystal Cu_8NC .

Comment 1: There are some statements that should be clarified throughout such as where "resonant" (more usually "resonance") structures are discussed. These are discussed as if they have physical manifestation and are not simply ways of rationalising electronic structure. In the first sentence of "Catalyst Development and Characterization" the statement "The easily modifiable NHT ligands, including a resonant structure with a negative charge" isn't really true. The resonance structures of NHT ligands can be drawn as zwitterions, with the negative charge localised on the sulfur. This implies that the sulfur has more significant electron density associated with it than a standard thiocarbonyl. Which is fine, but needs to be discussed more carefully.

Response: Thank the reviewer so much for the comment. As suggested, we have revised about expression of resonance structures in the manuscript. And we added elucidation resonance structures of NHT ligands in the discussion.

The NHT ligands are classic ligand stabilizers and are also air- and moisture-stable molecules. The resonance structures of NHT ligands can be drawn as zwitterions, with the negative charge located on the sulfur atom, which elucidates that the sulfur anion has more significant electron density associated than a standard thiocarbonyl. The constructed conditions of Cu_4NC and Cu_8NC are under basic condition. When the NHT ligands are under basic conditions, NHT ligands can undergoes tautomerization, changing a thione form structure to an enol-like form structure (Figure 1b, compound D), which implies that the electron density of sulfur anion will be further improved and endows it highly strong interaction with copper. It is beneficial that S anion as electron donors donates electrons to electron acceptors of copper cluster cores to construct stable Cu-S bonds.

a The transition-metal as homogeneous catalysts catalyzed the hydroboration of alkynes.

-Low recovery ratio and low catalytic activity (low TON) for homogeneous catalyst

-Required specific conditions (such as a dry and inert atmosphere, high temperature, etc.)

[M] = Rh(I), Ir(III), Cu(I/II), Fe(II), Co(II),
Mg(II) and Ti(IV) salts, etc.

b **This work:** The Cu_4NC and the Cu_8NC as heterogeneous catalysts catalyzed the hydroboration of alkynes.

Ligand: N-heterocyclic thione (NHT)

- N-heterocyclic thione-stabilized copper nanoclusters

- High turnover number of Cu_4NC (77786)

- 28 examples of hydroboration, up to 99% yield

- Excellent regioselectivities, stereoselectivities and chemoselectivities

- C-B coupling under air atmosphere at room temperature

- High recovery ratio for microcrystal Cu_4NC and Cu_8NC catalysts

Fig. 1| Representative synthesis strategies for vinylboronate esters. a The transition-metal catalyzed the hydroboration of alkynes. **b** This work: The Cu_4NC and Cu_8NC as heterogeneous catalysts catalyzed the hydroboration of alkynes. R_1 represents kinds of functional group.

Comment 2: These issues are, however, minor in the face of the more significant issues in the work. Throughout the work, the copper "clusters" (nomenclature point really here - but is a small tetramer with a molecular weight of around 700 gmol^{-1} really a cluster?), are referred to using two key concepts; one that this catalysis is heterogeneous, and two that they provide "DRDS" (dynamically regulating dual catalytic site) catalysis. At this stage, I can see no evidence provided by the authors to back up these claims.

Response: Thanks for the reviewer's comment.

The term cluster defined by F. A. Cotton in the early 1960s feature the important metal–metal bonds (*Acc. Chem. Res.* 1969, 2, 240–247). As an important member of the metal-cluster family, ligand protected metal clusters composed of a few to hundreds of metal atoms that are coordinated by a definite ligand shell (*Chem. Rev.* 2017, 117, 8208–8271; *Acc. Chem. Res.* 2015, 48, 1570–1579). The molecular-like purity of metal nanoclusters can make high-quality crystals available, and their atomic-packing structures can be determined by single-crystal X-ray crystallography (*Acc. Chem. Res.* 2023, 56, 1528–1538).

For ligand-protected metal clusters, a few to hundreds of metal atoms are held together mainly, or at least to a significant extent, by bonds directly between metal atoms, even though sometimes some non-metal atoms auxiliary glue them, where the glued metal entities are stabilized by a definite organic ligand shell (*Chem. Rev.* 2020, 120, 2, 526–622). Based on the single-crystal X-ray diffraction (SCXRD) structural analysis, the metal core structure of Cu_4NC is constructed by four copper atoms held together by metal-metal bonds between copper metal atoms. And the Cu_4NC metal core are stabilized by organic ligand shell. Therefore, the Cu_4NC meets the definition of cluster, which is really cluster.

The microcrystal Cu_4NC and Cu_8NC are heterogeneous catalysts in the hydroboration reactions. With regard to the issue of microcrystal Cu_4NC and Cu_8NC as heterogeneous catalysts, we did not make this clear and generated the unnecessary confusion. To further confirm this point, a series of experiments and characterization tests were carried out.

First, taking Cu_4NC as an example, through further research, we found that microcrystal Cu_4NC is completely insoluble in MeCN and H_2O solution (Supplementary Figures 16–17). The microcrystal Cu_4NC has a moderate solubility in DCM, CHCl_3 and DMSO, respectively. So Cu_4NC can be characterized by NMR or ESI-MS in CDCl_3 or DMSO, respectively. The hydroboration catalyzed by microcrystal Cu_4NC reacts in the mixture solvent (MeCN- H_2O). Due to the complete insolubility of microcrystal Cu_4NC in MeCN and H_2O solution, the microcrystal Cu_4NC is considered as heterogeneous catalysts in this reaction system.

Second, the hydroboration reaction catalyzed by microcrystal Cu_4NC was conducted. After the reaction was completed, the reaction mixture was centrifuged at 8000 rpm for 5 minutes. The microcrystal Cu_4NC catalyst was filtered, washed with hexane and then dried in vacuum to recycle the microcrystal Cu_4NC . Recycled microcrystal Cu_4NC catalyst was further characterized by ESI-MS (Supplementary Figure 37), NMR spectroscopy (Supplementary Figure 35), PXRD (Supplementary Figure 31), and UV-vis spectroscopy (Supplementary Figure 33), proving the microcrystal Cu_4NC manifests robustness after catalysis.

Third, we separated the catalyst from the supernatant of the reaction system by filtration, then the reaction solution was characterized by inductively coupled plasma mass spectrometric (ICP-MS) measurement, showing that no Cu^+ ions was found in the reaction solution, suggesting that Cu^+

ions were not leached from the metal cluster catalyst into solution over the course of the reaction.

Fourth, three parallel experiments of hydroboration catalyzed by microcrystal Cu_4NC also were carried out. The parallel experiments will be stopped at the 10th minute, the 30th minute and the 60th minute. The reaction mixture was centrifuged. The microcrystal Cu_4NC catalyst was filtered, washed and then dried in vacuum to recycle the microcrystal Cu_4NC . Recycled microcrystal Cu_4NC catalyst was further characterized by PXRD (Supplementary Figure 93). It showed that the characteristic peaks of the catalyzed microcrystal Cu_4NC at different time (the 10th minute, the 30th minute and the 60th minute) matched well with the characteristic peaks of the new microcrystal Cu_4NC , confirming the structure of the recycled catalyst were maintained.

Finally, the recycled microcrystal Cu_4NC catalyst was used again in the next cycle. In this way, the hydroboration reaction of alkynes was conducted for 5 cycles with the microcrystal Cu_4NC catalyst (Supplementary Figure 91).

Similarly, the research found that the microcrystal Cu_8NC can dissolve in DCM, CHCl_3 and DMSO, respectively. So Cu_8NC can be also characterized by NMR or ESI-MS in CDCl_3 or DMSO, respectively. Same as microcrystal Cu_4NC , the microcrystal Cu_8NC is also completely insoluble in H_2O and MeCN solution (Supplementary Figures 21 and 22). Since this hydroboration reaction works in the mixture solvent ($\text{MeCN-H}_2\text{O}$), the microcrystal Cu_8NC is considered as heterogeneous catalyst in hydroboration. To further support this point, similarly, we have also made corresponding experiments similar to those of the microcrystal Cu_4NC catalyst (Supplementary Figures 21, 22, 32, 34, 36 and 38).

All of these observations and experimental results suggested that microcrystal Cu_4NC and Cu_8NC can be considered as heterogeneous catalysts in the catalytic system of hydroboration.

The dynamically regulating dual-catalytic sites (DRDS) were proposed for three reasons.

First, by SCXRD analysis, in the Cu_4NC unit and Cu_8NC unit, the Cu-Cu distance is 2.662 Å and 2.663 Å, respectively, which are shorter than that of dicopper (I) μ -boryl complexes (*Chem. Sci.* 2022, 13, 6619–6625), providing a foundation for the interaction between hydroxide ions and two adjacent Cu atoms to form intermediate A or intermediate B.

Second, compared with Cu-S bonds, Cu-N bonds are weaker and easier to dissociate, decreasing the steric hindrance around the copper catalytic sites and exposing them to the substrate. This is supported by the following experiments, DFT calculations and *in situ* characterizations.

a. Through DFT-calculated binding energies of Cu-S and Cu-N bonds in Cu_4NC and Cu_8NC , we determined that the binding energies of Cu-S bonds are -20.7 kcal/mol and -24.2 kcal/mol, respectively. These values are lower than the binding energies of Cu-N bonds, which are -15.0 kcal/mol and -17.6 kcal/mol (Supplementary Figure 71). This indicates that the Cu-S bond is stronger than the Cu-N bond. Our findings align with previously reported results in the literature (*Chem. Rev.* 2024, 124, 7262-7378), which also demonstrates that the Cu-N bond is weaker compared to the Cu-S bond.

b. Taking Cu_4NC as an example, under basic condition, Cu-N bonds of the Cu_4NC are easily dissociated to form intermediate A. Mechanistic studies and density functional theory calculations illustrate that the two flexible Cu-N bonds of the copper NCs could dynamically dissociate from the copper cluster core to generate key intermediates A in a basic proton environment. As shown in Supplementary Figure 72, the energy barrier leading to intermediate A is only 2.9 kcal/mol and the energy of intermediate A is -19.3 kcal/mol, indicating that this process is both kinetically and

thermodynamically favorable. In addition, the hydroboration reaction was monitored by the *in situ* Raman spectroscopy. We observed that the Raman peak intensities of Cu-N bonds gradually decreased as the reaction proceeds (Supplementary Figure 82), illustrating that the Cu-N bonds may undergo dissociation. The experimental result is consistent with the theoretical calculation, which further illustrated that Cu-N bonds have the flexible character in this system.

Third, after the catalytic reactions were completed, microcrystal Cu_4NC and Cu_8NC were centrifuged, filtered, washed and then dried to recycle the microcrystal Cu_4NC and Cu_8NC . Recycled microcrystal Cu_4NC and Cu_8NC catalyst were further characterized by ESI-MS (Supplementary Figure 37), NMR spectroscopy (Supplementary Figure 35), PXRD (Supplementary Figure 31), and UV-vis spectroscopy (Supplementary Figure 33), proving the microcrystal Cu_4NC and Cu_8NC manifests robustness after catalysis. The result illustrated that the dissociated Cu-N bonds could be re-constructed, showing that Cu-N bonds have the reversible character.

In summary, in this catalytic system, the flexible Cu-N bonds can be dissociated to expose catalytic sites, the shorter Cu-Cu distance can facilitate the interaction between hydroxide ions and two adjacent Cu atoms to form intermediate A or B, and the dissociated Cu-N bonds can be re-constructed to maintain the integrity of the catalyst structures. These experiments, DFT calculations and *in situ* characterizations support that Cu_4NC and Cu_8NC catalysts have the dynamically regulating dual-catalytic site.

Supplementary Figure 16. ^1H NMR spectrum of microcrystal Cu_4NC in CD_3CN . No characteristic peaks of Cu_4NC are found in ^1H NMR spectrum, proving that microcrystal Cu_4NC is completely insoluble in MeCN solution.

Supplementary Figure 17. ^1H NMR spectrum of microcrystal Cu_4NC in D_2O . No characteristic peaks of Cu_4NC are found in ^1H NMR spectrum, proving that microcrystal Cu_4NC is completely insoluble in H_2O solution.

Supplementary Figure 93. PXRD patterns of microcrystal Cu_4NC before catalysis and after the 10th minute, the 30th minute and the 60th minute catalysis.

Supplementary Figure 91. Recyclability of the microcrystal Cu_4NC catalyzed hydroboration reaction in term of yield (1 h).

Supplementary Figure 21. ^1H NMR spectrum of microcrystal Cu_8NC in CD_3CN . No characteristic peaks of Cu_8NC are found in ^1H NMR spectrum, proving that microcrystal Cu_8NC is completely insoluble in MeCN solution.

Supplementary Figure 22. ^1H NMR spectrum of microcrystal Cu_8NC in D_2O . No characteristic peaks of Cu_8NC are found in ^1H NMR spectrum, proving that microcrystal Cu_8NC is completely insoluble in H_2O solution.

Supplementary Figure 71. DFT-calculated the binding energies of Cu-S bonds and Cu-N bonds in Cu_4NC and Cu_8NC (ΔG in kcal/mol).

Supplementary Figure 72. DFT-computed Gibbs free energy profile for formation of Intermediate A. The relative Gibbs energies and electronic energies (in parentheses) are given in kcal/mol.

Supplementary Figure 82. Time-dependent *in situ* Raman spectra of the hydroboration reaction catalyzed by microcrystal Cu_4NC catalyst (from bottom to top: 1min, 10min, 15min, 20min). Reaction conditions: Phenylacetylene **1** (0.2 mmol, 1.0 equiv.), B_2Pin_2 **2** (0.44 mmol, 1.0 equiv.), Cu_4NC catal. (0.004 mmol) and K_2CO_3 (0.44 mmol, 1.0 equiv.) were added to the mixture solvent (2.0 mL, MeCN/ H_2O 4:1) under air atmosphere at room temperature and allowed to react for 1 h.

Comment 3: The first point, that this reaction is heterogeneous may be true but I simply cannot tell.

- what evidence do the authors have that the reaction happens on the surface of a bulk material?
- are the authors, instead, contending that (relatively) small "clusters" are heterogeneous catalysts even if they go fully into solution? If so, that's a rather loose, and in my opinion inappropriate use of heterogeneous.

Thus, my first concern is whether the reaction is truly heterogeneous. Some hints that it may be are present in the manuscript including the line "DRDS Cu₄NC and DRDS Cu₈NC catalysts were also recovered and reused again." Now, the most reasonable way to do this would be to filter the reaction mixture, collect the filtrate, analyse it and show it to be Cu₄NC and Cu₈NC. I can see no evidence that the authors have done this in the manuscript, despite repeated claims of recovery. Perhaps the authors are referring to the recycling experiments present on page 46 of the supplementary? If so, they have written this section in a completely inappropriate way - repeated dosing is not recovery of the catalyst. If they have performed recovery experiments they should be in the ESI.

Significant evidence to the contrary is also provided - the authors provide solution state NMR of the catalysts! they thus form homogeneous solutions with at least some solvent systems.

Response: Thank the reviewer so much for the comment.

The microcrystal Cu₄NC and Cu₈NC are heterogeneous catalysts in this hydroboration reactions. With regard to the issue of microcrystal Cu₄NC and Cu₈NC as heterogeneous catalysts, we are sorry for the unclear descriptions. To further confirm this point, a series of experiments and characterization tests were carried out.

First, taking Cu₄NC as an example, through further research, we found that microcrystal Cu₄NC is completely insoluble in MeCN and H₂O solution (Supplementary Figures 16 and 17). The microcrystal Cu₄NC has a moderate solubility in DCM, CHCl₃ and DMSO, respectively. So Cu₄NC can be characterized by NMR or ESI-MS in CDCl₃ or DMSO, respectively. The hydroboration catalyzed by microcrystal Cu₄NC reacts in the mixture solvent (MeCN-H₂O). Due to the complete insolubility of microcrystal Cu₄NC in MeCN and H₂O solution, the microcrystal Cu₄NC is considered as heterogeneous catalysts in this reaction system.

Second, the hydroboration reaction catalyzed by microcrystal Cu₄NC was conducted. After the reaction was completed, the reaction mixture was centrifuged. The microcrystal Cu₄NC catalyst was filtered, washed with hexane and then dried in vacuum to recycle the microcrystal Cu₄NC. Recycled microcrystal Cu₄NC catalyst was further characterized by ESI-MS (Supplementary Figure 37), NMR spectroscopy (Supplementary Figure 35), PXRD (Supplementary Figure 31), and UV-vis spectroscopy (Supplementary Figure 33), proving the microcrystal Cu₄NC manifests robustness after catalysis.

Third, we separated the catalyst from the supernatant of the reaction system by filtration, then the reaction solution was characterized by inductively coupled plasma mass spectrometric (ICP-MS) measurement, showing that no Cu⁺ ions was found in the reaction solution, suggesting that Cu⁺ ions were not leached from the metal cluster catalyst into solution over the course of the reaction.

Fourth, three parallel experiments of hydroboration catalyzed by microcrystal Cu₄NC also were carried out. The parallel experiments will be stopped at the 10th minute, the 30th minute and the 60th minute. The reaction mixture was centrifuged. The microcrystal Cu₄NC catalyst was filtered, washed and then dried in vacuum to recycle the microcrystal Cu₄NC. Recycled microcrystal

Cu₄NC catalyst was further characterized by PXRD (Supplementary Figure 93). It showed that the characteristic peaks of the catalyzed microcrystal Cu₄NC at different time (the 10th minute, the 30th minute and the 60th minute) matched well with the characteristic peaks of the new microcrystal Cu₄NC, confirming the structure of the recycled catalyst were maintained.

Finally, the recycled microcrystal Cu₄NC catalyst was used again in the next cycle. In this way, the hydroboration reaction of alkynes was conducted for 5 cycles with the microcrystal Cu₄NC catalyst (Supplementary Figure 91).

Similarly, the research found that the microcrystal Cu₈NC can dissolve in DCM, CHCl₃ and DMSO, respectively. So Cu₈NC can be also characterized by NMR or ESI-MS in CDCl₃ or DMSO, respectively. Same as microcrystal Cu₄NC, the microcrystal Cu₈NC is also completely insoluble in H₂O and MeCN solution (Supplementary Figure 21 and 22). Since this hydroboration reaction works in the mixture solvent (MeCN-H₂O), the microcrystal Cu₈NC is considered as heterogeneous catalyst in hydroboration. To further support this point, similarly, we have also made corresponding experiments similar to those of the microcrystal Cu₄NC catalyst (Supplementary Figures 21, 22, 32, 34, 36 and 38).

All of these observations and experimental results suggested that microcrystal Cu₄NC and Cu₈NC can be considered as heterogeneous catalysts in the catalytic system of hydroboration.

Taking Cu₄NC as an example, it can be seen that the hydroboration reaction happens on the surface of a bulk material from five aspects.

First, microcrystal Cu₄NC is completely insoluble in MeCN and H₂O solution (Supplementary Figures 16 and 17). The hydroboration reaction catalyzed by microcrystal Cu₄NC took place in the mixture solvent (MeCN-H₂O). It proves that hydroboration reaction happens on the surface of microcrystal Cu₄NC.

Second, the hydroboration reaction catalyzed by microcrystal Cu₄NC was conducted. After the reaction was completed, the reaction mixture was filtered to recycle the microcrystal Cu₄NC. The recyclable microcrystal Cu₄NC can remain intact after catalytic reactions, which were demonstrated by the PXRD measurements (Supplementary Figure 31).

Third, after the reaction was completed, we separated the catalyst from the reaction system by filtration, then the reaction solution was characterized by ICP-MS measurement, showing that no Cu⁺ ions was found in the reaction solution, suggesting that Cu⁺ ions were not leached from the metal cluster catalyst into solution over the course of the reaction.

Fourth, three parallel experiments of hydroboration catalyzed by microcrystal Cu₄NC also were carried out. The parallel experiments will be stopped at the 10th minute, the 30th minute and the 60th minute. The reaction mixture was centrifuged. The microcrystal Cu₄NC catalyst was filtered, washed and then dried in vacuum to recycle the microcrystal Cu₄NC. Recycled microcrystal Cu₄NC catalyst was further characterized by PXRD (Supplementary Figure 93). It showed that the characteristic peaks of the catalyzed microcrystal Cu₄NC at different time (the 10th minute, the 30th minute and the 60th minute) matched well with the characteristic peaks of the new microcrystal Cu₄NC, confirming the structure of the recycled catalyst were maintained.

Finally, the recycled microcrystal Cu₄NC catalyst was used again in the next cycle. In this way, the hydroboration reaction of alkynes was conducted for 5 cycles with the microcrystal Cu₄NC catalyst (Supplementary Figure 91).

All of these results indicated that hydroboration reaction happens on the surface of a bulk material.

Regarding the issue that (relatively) small "clusters" are heterogeneous catalysts even if they go fully into solution, we tend to contend that the clusters are heterogeneous catalysts when the clusters are completely insoluble in reaction solution. If the clusters go fully into reaction solution, we tend to contend that the clusters are homogeneous catalysts. As mentioned before, the microcrystal Cu_4NC and Cu_8NC are completely insoluble in reaction solution ($\text{MeCN-H}_2\text{O}$) before and after the reaction, which are characterized by various spectra, proving the microcrystal Cu_4NC and Cu_8NC are heterogeneous catalysts in the catalytic system of hydroboration.

In term of the issue about recovery experiments, we have repeated the recovery experiments. When we studied the recovery experiments, we found that the weak interaction between recycled microcrystal Cu_4NC catalyst and filter paper can lead to unnecessary loss of microcrystal Cu_4NC catalyst, thereby limiting the number of cycles of catalysis. To eliminate the influence of the weak interaction, the repeated dosing projects were carried out. In this way, the hydroboration reaction of alkynes was conducted for more than 20 cycles (Supplementary Figure 92). Indeed, as mentioned by the reviewer, "repeated dosing is not recovery of the catalyst", we have repeated the recycling experiments as suggested.

Under air atmosphere, alkynes **1** (0.2 mmol, 1.0 eq), B_2Pin_2 **2** (0.44 mmol, 2.2 equiv.), Cu_4NC catalysts (2.0 mol%), K_2CO_3 (0.44 mmol, 2.2 equiv.) and the mixture solvent (2.0 mL, $\text{MeCN-H}_2\text{O}$, v/v 4/1) were added into a 10 mL flask. The reaction mixture was stirred at room temperature for 1 h. Conversions and yields were determined by ^1H NMR using 1,3,5-trimethoxybenzene as an internal standard. After the reaction is completed, the reaction mixture was centrifuged at 8000 rpm for 5 minutes. The microcrystal Cu_4NC catalyst was filtered, washed with hexane and then dried in vacuum to recycle the microcrystal Cu_4NC . The recycled microcrystal Cu_4NC catalyst was used again in the next recycle. In this way, the hydroboration reaction of alkynes was conducted for 5 cycles with the microcrystal Cu_4NC catalyst (Supplementary Figure 91). In the revised manuscript, we explained the above experimental procedures clearly.

In term of issue about solution state NMR of catalysts, as mentioned before, the microcrystal Cu_4NC and Cu_8NC has a moderate solubility in DCM, CHCl_3 , respectively. So Cu_4NC and Cu_8NC can be characterized by NMR in CDCl_3 , respectively. But the microcrystal Cu_4NC and Cu_8NC are completely insoluble in reaction solution ($\text{MeCN-H}_2\text{O}$) before and after the reaction and are characterized by various spectra, proving the microcrystal Cu_4NC and Cu_8NC are heterogeneous catalysts in the catalytic system of hydroboration.

Supplementary Figure 16. ^1H NMR spectrum of microcrystal Cu_4NC in CD_3CN . No characteristic peaks of Cu_4NC are found in ^1H NMR spectrum, proving that microcrystal Cu_4NC is completely insoluble in MeCN solution.

Supplementary Figure 17. ^1H NMR spectrum of microcrystal Cu_4NC in D_2O . No characteristic peaks of Cu_4NC are found in ^1H NMR spectrum, proving that microcrystal Cu_4NC is completely insoluble in H_2O solution.

Supplementary Figure 93. PXRD patterns of microcrystal Cu₄NC before catalysis and after the 10th minute, the 30th minute and the 60th minute catalysis.

Supplementary Figure 91. Recyclability of the microcrystal Cu₄NC catalyzed hydroboration reaction in term of yield (1 h).

Supplementary Figure 21. ^1H NMR spectrum of microcrystal Cu_8NC in CD_3CN . No characteristic peaks of Cu_8NC are found in ^1H NMR spectrum, proving that microcrystal Cu_8NC is completely insoluble in MeCN solution.

Supplementary Figure 22. ^1H NMR spectrum of microcrystal Cu_8NC in D_2O . No characteristic peaks of Cu_8NC are found in ^1H NMR spectrum, proving that microcrystal Cu_8NC is completely insoluble in H_2O solution.

Supplementary Figure 91. Recyclability of the microcrystal Cu_4NC catalyzed hydroboration reaction in term of yield (1 h).

Comment 4: If the reaction is heterogeneous in the traditional sense, is the DFT approach used in any way appropriate? My interpretation would be no, as it models a molecular system.

Response: Thank you for your valuable feedback and for raising this important point. We appreciate the opportunity to clarify the suitability of our approach. While DFT is indeed traditionally used for molecular systems, it can also be effectively applied to clusters with the weak noncovalent interactions, such as the one in our study. Here are the reasons supporting this. Firstly, unlike the COF, MOF or MoS_2 these solid catalyst with strong chemical bond, in which atoms arrange orderly, the noncovalent interactions between the clusters are very weak. And the free NHT ligand have no catalytic activity in our system (Table 1, entry 19). If we still treat it in the periodic boundary conditions, just like traditional heterogeneous catalyst materials, we can't reproduce the structures of clusters well. However, the interactions and electronic structure within the cluster are crucial for understanding its properties, especially the catalysis, and DFT provides a robust framework for this analysis.

Projector Augmented-Wave (PAW), when implemented in plane-wave codes, is optimized for periodic systems. For non-periodic systems like isolated molecules or clusters, the use of a large supercell to avoid interactions between periodic images can lead to inefficiencies (*Physical Review B*. 1994, 50, 17953–17978). This requires a significant number of plane waves to describe the system accurately, increasing computational cost. And the plane-wave basis set may introduce artifacts in the charge density, particularly in regions far from the nucleus, which can affect the accuracy of properties that depend on the electron density distribution. Non-periodic systems might have surfaces or edges that are not adequately modeled by PAW in a plane-wave basis set. The treatment of these surfaces can be less accurate compared to all-electron methods or localized basis sets. Although a large vacuum region is often introduced around the molecule or cluster to minimize interactions between periodic images, some residual interactions might still exist,

potentially leading to inaccuracies. In addition, achieving convergence in the total energy and forces for non-periodic systems can be more challenging due to the need to accurately represent the isolated system in a periodic framework.

Herein, we use Gaussian-based DFT with the def2-SVP basis set, which is particularly well-suited for systems with the weak noncovalent interactions. This basis set strikes an excellent balance between computational efficiency and accuracy, making it ideal for studying clusters. The localized nature of the Gaussian basis functions helps minimize computational artifacts related to artificial periodicity, ensuring a more accurate representation of the cluster. For instance, in catalysts like Cu_4NC or Cu_8NC , the interactions between clusters are relatively weak, which justifies the examination of single clusters in isolation. This approach effectively models the catalysis at the active centers, which typically involve only a few atoms. Additionally, by treating these as molecular systems, we can apply more accurate hybrid functionals instead of relying solely on the Generalized Gradient Approximation (GGA) with Perdew-Burke-Ernzerhof (PBE) functional. This not only enhances the precision of our simulations but also provides a more detailed understanding of the catalytic processes at the atomic level. In literature, there are also many articles treating the cluster as molecular systems (*Angew. Chem. Int. Ed.* 2023, 62, e202216735; *Angew. Chem. Int. Ed.* 2021, 60, 10573–10576; *Angew. Chem. Int. Ed.* 2021, 60, 26136–2614; *J. Am. Chem. Soc.* 2022, 144, 10844–10853; *J. Am. Chem. Soc.* 2021, 143, 1768–1772; *Chem* 2022, 8, 2380–2392; *J. Am. Chem. Soc.* 2024, 146, 16295–16305; *Angew. Chem. Int. Ed.* 2018, 57, 9775–9779; *J. Am. Chem. Soc.* 2020, 142, 4141–4153).

Comment 5: Now, irrespective as to whether the statement that the reaction is "heterogeneous" is true or not (and I'm just asking for evidence there), it reflects a fundamental second flaw in the work. The authors provide absolutely no evidence as far as I can see that Cu_4NC is the actual catalytic vector! As a result, their entire mechanistic postulate sits on exceptionally shaky grounds and the DRDS argument lacks foundation. The authors provide a little evidence that the system does persist in solution throughout the catalysis (e.g. page 18 of the ESI), but to my eye there's more going on in the spectra than is discussed.

- What have the authors done to preclude the idea that the clusters come apart during the catalysis in solution, then when the substrate is consumed reoligomerise? In situ NMR of the reaction mixture may assist here
- What happens to the spectra of the materials when the substrates are added to the catalysts in stoichiometric studies?

Response: Thanks for the reviewer's comment. The microcrystal Cu_4NC and Cu_8NC can be used as heterogeneous catalysts in the catalytic system of hydroboration, which has been closely illustrated on Comment 3.

From the following four aspects, it can be concluded that the microcrystal Cu_4NC is catalytic vector of hydroboration.

First, compared the control experiment Entry 1 and Entry 21 in **Table 1**, microcrystal Cu_4NC was used as catalyst for the hydroboration, providing single vinylboron products with good yield of 98% (table 1 Entry 1), the hydroboration cannot be achieved under the lack of copper catalysts condition (table 1 Entry 21).

Second, after the reaction was completed, the reaction mixture was centrifuged. The microcrystal Cu_4NC catalyst was filtered, washed and then dried in vacuum to recycle the microcrystal Cu_4NC .

Recycled microcrystal Cu_4NC catalyst was further characterized by ESI-MS (Supplementary Figure 37), NMR spectroscopy (Supplementary Figure 35), PXRD (Supplementary Figure 31), and UV-vis spectroscopy (Supplementary Figure 33), proving the microcrystal Cu_4NC manifests robustness after catalysis.

Third, three parallel experiments of hydroboration catalyzed by microcrystal Cu_4NC also were carried out. The parallel experiments will be stopped at the 10th minute, the 30th minute and the 60th minute. The reaction mixture was centrifuged at 8000 rpm for 5 minutes. The microcrystal Cu_4NC catalyst was filtered, washed with hexane and then dried in vacuum to recycle the microcrystal Cu_4NC . Recycled microcrystal Cu_4NC catalyst was further characterized by PXRD (Supplementary Figure 93). It showed that the characteristic peaks of the catalyzed microcrystal Cu_4NC at different time (the 10th minute, the 30th minute and the 60th minute) matched well with the characteristic peaks of the new microcrystal Cu_4NC , confirming the structure of the recycled catalyst were maintained.

Fourth, the recycled microcrystal Cu_4NC catalyst was used again in the next cycle. In this way, the hydroboration reaction of alkynes was conducted for 5 cycles with the microcrystal Cu_4NC catalyst (Supplementary Figure 91).

In a word, these results indicated that the microcrystal Cu_4NC is catalytic vector of hydroboration.

The dynamically regulating dual-catalytic site (DRDS) were proposed for three reasons.

First, by SCXRD analysis, in the Cu_4NC unit and Cu_8NC unit, the Cu-Cu distance is 2.662 Å and 2.663 Å, respectively, which are shorter than that of dicopper (I) μ -boryl complexes (*Chem. Sci.* 2022, 13, 6619–6625), providing a foundation for the interaction between hydroxide ions and two adjacent Cu atoms to form intermediate A or intermediate B.

Second, compared with Cu-S bonds, Cu-N bonds are weaker and easier to dissociate, decreasing the steric hindrance around the copper catalytic sites and exposing them to the substrate. This is supported by the following experiments, DFT calculations and *in situ* characterizations.

a. Through DFT-calculated binding energies of Cu-S and Cu-N bonds in Cu_4NC and Cu_8NC , we determined that the binding energies of Cu-S bonds are -20.7 kcal/mol and -24.2 kcal/mol, respectively. These values are lower than the binding energies of Cu-N bonds, which are -15.0 kcal/mol and -17.6 kcal/mol (Supplementary Figure 71). This indicates that the Cu-S bond is stronger than the Cu-N bond. Our findings align with previously reported results in the literature (*Chem. Rev.* 2024, 124, 7262-7378), which also demonstrates that the Cu-N bond is weaker compared to the Cu-S bond.

b. Taking Cu_4NC as an example, under basic condition, Cu-N bonds of the Cu_4NC are easily dissociated to form intermediate A. Mechanistic studies and density functional theory calculations illustrate that the two flexible Cu-N bonds of the copper NCs could dynamically dissociate from the copper cluster core to generate key intermediates A in a basic proton environment. As shown in Supplementary Figure 72, the energy barrier leading to intermediate A is only 2.9 kcal/mol and the energy of intermediate A is -19.3 kcal/mol, indicating that this process is both kinetically and thermodynamically favorable. In addition, the hydroboration reaction was monitored by the *in situ* Raman spectroscopy. We observed that the Raman peak intensities of Cu-N bonds gradually decreased as the reaction proceeds (Supplementary Figure 82), illustrating that the Cu-N bonds maybe undergo dissociation. The experimental result is consistent with the theoretical calculation, which further illustrated that Cu-N bonds have the flexible character in this system.

Third, after the catalytic reactions were completed, microcrystal Cu_4NC and Cu_8NC were centrifuged, filtered, washed and then dried to recycle the microcrystal Cu_4NC and Cu_8NC . Recycled microcrystal Cu_4NC and Cu_8NC catalyst were further characterized by ESI-MS (Supplementary Figure 37), NMR spectroscopy (Supplementary Figure 35), PXRD (Supplementary Figure 31), and UV-vis spectroscopy (Supplementary Figure 33), proving the microcrystal Cu_4NC and Cu_8NC manifests robustness after catalysis. The result illustrated that the dissociated Cu-N bonds could be re-constructed, showing that Cu-N bonds have the reversible character.

In summary, in this catalytic system, the flexible Cu-N bonds can be dissociated to expose catalytic sites, the shorter Cu-Cu distance can facilitate the interaction between hydroxide ions and two adjacent Cu atoms to form intermediate A or B, and the dissociated Cu-N bonds can be re-constructed to maintain the integrity of the catalyst structures. These experiments, DFT calculations and *in situ* characterizations support that Cu_4NC and Cu_8NC catalysts have the dynamically regulating dual-catalytic site.

I am very sorry for causing you confusion that “*The authors provide a little evidence that the system does persist in solution throughout the catalysis (e.g. page 18 of the ESI), but to my eye there's more going on in the spectra than is discussed.*”

As mentioned above, the microcrystal Cu_4NC and Cu_8NC have a moderate solubility in DCM, CHCl_3 and DMSO, respectively. So microcrystal Cu_4NC and Cu_8NC can be characterized by NMR in CDCl_3 . The microcrystal Cu_4NC and Cu_8NC can be characterized by ESI-MS in DMSO. The microcrystal Cu_4NC and Cu_8NC can be characterized by UV-Vis spectra in DCM. However, we also found that microcrystal Cu_4NC is completely insoluble in MeCN and H_2O solution (Supplementary Figures 16 and 17). The hydroboration catalyzed by microcrystal Cu_4NC and Cu_8NC reacts in the mixture solvent (MeCN- H_2O). The Supplementary Figures on the page 18 of the ESI are that the recycled microcrystal Cu_4NC and Cu_8NC were characterized by NMR spectroscopy in CDCl_3 (Supplementary Figures 35 and 36). The NMR spectra of the recycled microcrystal Cu_4NC and Cu_8NC were performed aiming to characterize the structures of the clusters, where the solvent used is CDCl_3 instead of MeCN- H_2O solvent used in the catalytic system. It should be noted that the solvent of our catalytic system is mixture solvent (MeCN- H_2O). The specific method for NMR characterization of microcrystal Cu_4NC catalyst is shown below: Taking Cu_4NC as an example, the hydroboration reaction catalyzed by microcrystal Cu_4NC was carried out. After the reaction is completed, the reaction mixture was centrifuged. The microcrystal Cu_4NC catalyst was filtered, washed with hexane and then dried in vacuum to recycle the microcrystal Cu_4NC . Recycled microcrystal Cu_4NC catalyst was further characterized by NMR spectroscopy in CDCl_3 . At the same time, some characteristic peaks of hexane and H_2O in ^1H NMR spectra of the recycling the microcrystal Cu_4NC have been assigned (Supplementary Figure 35).

To preclude the idea that “*the clusters come apart during the catalysis in solution, then when the substrate is consumed reoligomerise*”, **the control experiments were carried out.**

Taking Cu_4NC as an example, as mentioned above, the microcrystal Cu_4NC have a moderate solubility in DCM and CHCl_3 , respectively. In the mixture solvent (DCM- H_2O or CHCl_3 - H_2O), the hydroboration catalyzed by microcrystal Cu_4NC did not react (Table 1 Entries 17 and 18). When the catalytic system is in the mixture solvent (MeCN- H_2O), the hydroboration can react

(Table 1 Entry 1). But microcrystal Cu_4NC is completely insoluble in MeCN and H_2O (Supplementary Figures 16 and 17). The *in situ* NMR of the reaction mixture could not assist here. To verify the stability of the catalyst and to verify that the crystal state of microcrystal Cu_4NC has been maintained in hydroboration, three parallel experiments of hydroboration catalyzed by microcrystal Cu_4NC also were carried out. The parallel experiments will be stopped at the 10th minute, the 30th minute and the 60th minute. The reaction mixture was centrifuged at 8000 rpm for 5 minutes. The microcrystal Cu_4NC catalyst was filtered, washed with hexane and then dried in vacuum to recycle the microcrystal Cu_4NC . Recycled microcrystal Cu_4NC catalyst was further characterized by PXRD (Supplementary Figure 93). It showed that the characteristic peaks of the recycle microcrystal Cu_4NC at different time (the 10th minute, the 30th minute and the 60th minute) matched well with the characteristic peaks of the new microcrystal Cu_4NC . In a word, the results illustrated that the hydroboration happens on the surface of microcrystal Cu_4NC , and microcrystal Cu_4NC is stable in hydroboration process.

To make sure the change of the microcrystal Cu_4NC when the substrates are added to the catalysts in stoichiometric studies, the NMR experiments were carried out. As mentioned above, the Cu_4NC have a moderate solubility in CHCl_3 . However, in the mixture solvent ($\text{CHCl}_3\text{-H}_2\text{O}$), the hydroboration catalyzed by microcrystal Cu_4NC did not react. As suggested by reviewer, 1.0 eq. microcrystal Cu_4NC , 2.0 eq. phenylacetylene, 4.0 eq. B_2Pin_2 and 4.0 eq. K_2CO_3 were added into the mixture solvent ($\text{CDCl}_3\text{-D}_2\text{O}$). The *in situ* ^1H NMR was monitored at the 5th minute, the 30th minute and the 60th minute, showing that characteristic peaks of Cu_4NC have no change and the products are not found (Supplementary Figure 94). The other NMR experiment, that 10 mol% microcrystal Cu_4NC , 1.0 eq. phenylacetylene, 2.2 eq. B_2Pin_2 and 2.2 eq. K_2CO_3 were added into the mixture solvent ($\text{CD}_3\text{CN-D}_2\text{O}$), were carried out and was monitored at the 0 minute, the 3rd minute, the 6th minute, the 10th minute and the 20th minute. It showed that the vinylboronate ester was got with excellent yields. Since Cu_4NC is completely insoluble in MeCN and H_2O , the characteristic peaks of Cu_4NC are not found (Figure R2). In addition, we also monitored the reaction process by *in situ* Fourier transform infrared (FT-IR) spectroscopy and *in situ* Raman spectroscopy measurements as above.

Supplementary Figure 93. PXRD patterns of microcrystal Cu_4NC before catalysis and after the 10th minute, the 30th minute and the 60th minute catalysis.

Supplementary Figure 91. Recyclability of the microcrystal Cu_4NC catalyzed hydroboration reaction in term of yield (1 h).

Supplementary Figure 71. DFT-calculated the binding energies of Cu-S bonds and Cu-N bonds in Cu_4NC and Cu_8NC (ΔG in kcal/mol).

Supplementary Figure 72. DFT-computed Gibbs free energy profile for formation of Intermediate A. The relative Gibbs energies and electronic energies (in parentheses) are given in kcal/mol.

The *in situ* Raman Experiment

Supplementary Figure 82. Time-dependent *in situ* Raman spectra of the hydroboration reaction catalyzed by microcrystal Cu_4NC catalyst (from bottom to top: 1min, 10min, 15min, 20min). Reaction conditions: Phenylacetylene **1** (0.2 mmol, 1.0 equiv.), B_2Pin_2 **2** (0.44 mmol, 1.0 equiv.), Cu_4NC catal. (0.004 mmol) and K_2CO_3 (0.44 mmol, 1.0 equiv.) were added to the mixture solvent (2.0 mL, MeCN/ H_2O 4:1) under air atmosphere at room temperature and allowed to react for 1 h.

Supplementary Figure 16. ^1H NMR spectrum of microcrystal Cu_4NC in CD_3CN . No characteristic peaks of Cu_4NC are found in ^1H NMR spectrum, proving that microcrystal Cu_4NC is completely insoluble in MeCN solution.

Supplementary Figure 17. ^1H NMR spectrum of microcrystal Cu_4NC in D_2O . No characteristic peaks of Cu_4NC are found in ^1H NMR spectrum, proving that microcrystal Cu_4NC is completely insoluble in H_2O solution.

Supplementary Figure 35. ^1H NMR spectra of Cu_4NC after catalysis (top: recycled microcrystal Cu_4NC), Cu_4NC before catalysis (middle: new microcrystal Cu_4NC) and methimazole ligand (bottom) in CDCl_3 .

Supplementary Figure 21. ^1H NMR spectrum of microcrystal Cu_8NC in CD_3CN . No characteristic peaks of Cu_8NC are found in ^1H NMR spectrum, proving that microcrystal Cu_8NC is completely insoluble in MeCN solution.

Supplementary Figure 22. ^1H NMR spectrum of microcrystal Cu_8NC in D_2O . No characteristic peaks of Cu_8NC are found in ^1H NMR spectrum, proving that microcrystal Cu_8NC is completely insoluble in H_2O solution.

Supplementary Figure 94. Time-dependent *in situ* ^1H NMR spectra of the hydroboration reaction catalyzed by microcrystal Cu_4NC catalyst in the mixture solvent ($\text{CDCl}_3\text{-H}_2\text{O}$). Reaction conditions: Phenylacetylene **1** (2.0 equiv.), B_2Pin_2 **2** (4.0 equiv.), Cu_4NC catal. (1.0 equiv.) and K_2CO_3 (4.0 equiv.) were added to the mixture solvent (2.0 mL, $\text{CDCl}_3\text{-H}_2\text{O}$) under air atmosphere at room temperature and allowed to react for 1 h. No product was found.

Figure R2. Time-dependent *in situ* ^1H NMR spectra of the hydroboration reaction catalyzed by microcrystal Cu_4NC catalyst in the mixture solvent ($\text{CD}_3\text{CN}-\text{H}_2\text{O}$). Reaction conditions: Phenylacetylene 1 (1.0 equiv.), B_2Pin_2 2 (2.2 equiv.), Cu_4NC catal. (10% mol) and K_2CO_3 (2.2 equiv.) were added to the mixture solvent (2.0 mL, $\text{CD}_3\text{CN}-\text{H}_2\text{O}$) under air atmosphere at room temperature and allowed to react for 20 min.

Comment 6: A few minor comments on the larger claims of the synthetic work. The authors contend this may be viable on an industrial scale but provide no evidence for this claim. They should remove it, or justify it with experimental work. The authors vaunt the selectivity of the reaction for alkynes over alkenes - this is entirely unsurprising. Alkynes are a much easier substrate, this is just an example of a relatively inactive catalyst. They also suggest selectivity over benzaldehyde, but they do the reaction in a water mixture! What's the evidence that the carbonyl remains intact in these conditions? Could it not have been hydrated to the geminal diol, which would then not be expected to be a viable substrate for hydroboration? On work up, this reversible hydration would then return the benzaldehyde.

Response: This is a good point. Thank reviewer for this great suggestion! We have revised manuscript as suggested to remove the claim of that the microcrystal Cu_4NC may be viable on the chemical industry.

The complex organic compounds, especially drug molecules and natural products, usually contain functional groups of alkynes and functional groups of alkenes, simultaneously. When only the functional groups of alkynes need to be derived to other functional groups, the character of high chemoselectivity for specific functional groups ($\text{C}\equiv\text{C}$) is important to the catalysts. From this point of view, the microcrystal Cu_4NC also has a certain value.

To prove that the carbonyl of benzaldehyde remains intact in our catalytic system, we carried out the NMR experiment. As reviewer suggested, 2.0 mol% microcrystal Cu_4NC , 1.0 eq. *p*-tolualdehyde, 2.2 eq. B_2Pin_2 and 2.2 eq. K_2CO_3 were added into the mixture solvent ($\text{CH}_3\text{CN}-\text{H}_2\text{O}$). The *in situ* ^1H NMR was monitored at the 0 minute, the 3rd minute, 5th minute, the

10th minute, 15th minute, the 20th minute, 30th minute, the 40th minute, 50th minute and the 60th minute, showing that characteristic peaks of *p*-tolualdehyde have no change (Supplementary Figure 41). The result illustrated that *p*-tolualdehyde is inert substrate to our catalytic system, indicating that the carbonyl remains intact our catalytic system.

Supplementary Figure 41. Time-dependent *in situ* ¹H NMR spectra of the reaction catalyzed by microcrystal Cu₄NC catalyst. Reaction conditions: *p*-tolualdehyde (0.2 mmol, 1.0 equiv.), B₂Pin₂ 2 (0.44 mmol, 2.2 equiv.), Cu₄NC catal. (0.004 mmol, 2.0 mol%) and K₂CO₃ (0.44 mmol, 2.2 equiv.) were added to the mixture solvent (2.0 mL, MeCN/H₂O 4:1) under air atmosphere at room temperature and allowed to react for 1 h.

Comment 7: Finally, a few minor comments the DFT itself. Fig 5 is almost entirely incomprehensible - a slightly better drawn version is provided in the ESI but is still hard to follow. It's remarkable to me that the authors didn't cite Tilley's μ -copper boryl system which is the closest thing in the lit to their postulated intermediate (<https://pubs.rsc.org/en/content/articlelanding/2022/sc/d2sc00848c>). Given that Tilley can isolate such a compound, the authors should be able to do so for their systems if their mechanism has any credence. The DFT generally does seem reasonable - the level of theory is appropriate and the steps work, but at this stage the authors need to actually provide some evidence that there is a meaningful relationship between the catalysis as observed and the DFT conclusions.

Response: Thanks for the reviewer's comment. In order to present the mechanism more clearly, as suggested, we have revised Figure 5 in the main text. The original Figure 5 in the main text has been send to the SI. As suggested, we have cited the literature in the revised manuscript.

As suggested, we have cited the Tilley's paper in the revised manuscript. The Tilley's μ -copper boryl system were prepared with the dinucleating ligand DPFN (2,7-bis(fluoro-di(2-pyridyl))

methyl)-1,8-naphthyridine) in THF solvent. The robust ligand allowed for the isolation of the thermally stable dicopper (I) boryl complexes and the stable dicopper (I) boryl complexes have a high solubility in THF, which is beneficial to isolating the dicopper (I) boryl complexes.

As mentioned above, the microcrystal Cu_4NC and Cu_8NC have a moderate solubility in DCM, CHCl_3 . So Cu_4NC and Cu_8NC can be characterized by NMR in CDCl_3 . Taking Cu_4NC as an example, the hydroboration catalyzed by microcrystal Cu_4NC did not react in the mixture solvent ($\text{DCM-H}_2\text{O}$ or $\text{CHCl}_3\text{-H}_2\text{O}$) (table 1 Entries 17 and 18), illustrating the microcrystal Cu_4NC -boryl system has not been produced in the mixture solvent ($\text{DCM-H}_2\text{O}$ or $\text{CHCl}_3\text{-H}_2\text{O}$). However, in our catalytic system, the solvent used are the mixture solvent ($\text{MeCN-H}_2\text{O}$). We found that microcrystal Cu_4NC are completely insoluble in MeCN and H_2O (Supplementary Figures 16 and 17). The microcrystal Cu_4NC and Cu_8NC are heterogeneous catalysts in the catalytic system of hydroboration, which has been closely illustrated on Comment 2 and Comment 3. Based on the actual situation, the microcrystal Cu_4NC -boryl intermediate could not be isolated.

The relationship between the catalysis as observed and the DFT conclusions were illuminated for three aspects.

First, by SCXRD analysis, in the Cu_4NC unit and Cu_8NC unit, the Cu-Cu distance is 2.662 Å and 2.663 Å, respectively, which are shorter than that of dicopper (I) μ -boryl complexes (*Chem. Sci.* 2022, 13, 6619–6625), providing a foundation for the interaction between hydroxide ions and two adjacent Cu atoms to form intermediate A or intermediate B. Meanwhile, mechanistic studies and density functional theory calculations illustrate that the two flexible Cu-N bonds of the copper NCs could dynamically dissociate from the copper cluster core to generate key intermediates A in a basic proton environment.

Second, through DFT-calculated binding energies of Cu-S and Cu-N bonds in Cu_4NC and Cu_8NC , we determined that the binding energies of Cu-S bonds are -20.7 kcal/mol and -24.2 kcal/mol, respectively. These values are lower than the binding energies of Cu-N bonds, which are -15.0 kcal/mol and -17.6 kcal/mol (Supplementary Figure 71). This indicates that the Cu-S bond is stronger than the Cu-N bond. Our findings align with previously reported results in the literature (*Chem. Rev.* 2024, 124, 7262-7378), which also demonstrates that the Cu-N bond is weaker compared to the Cu-S bond.

Taking Cu_4NC as an example, under basic condition, Cu-N bonds of the Cu_4NC are easily dissociated to form intermediate A. As shown in Supplementary Figure 72, the energy barrier leading to intermediate A is only 2.9 kcal/mol and the energy of intermediate A is -19.3 kcal/mol, indicating that this process is both kinetically and thermodynamically favorable. In addition, the hydroboration reaction was monitored by the *in situ* Raman spectroscopy. We observed that the Raman peak intensities of Cu-N bonds gradually decreased as the reaction proceeds (Supplementary Figure 82), illustrating that the Cu-N bonds maybe undergo dissociation. The experimental result is consistent with the theoretical calculation, which further illustrated that Cu-N bonds have the flexible character in this system.

Third, theory calculations show that the hydroboration catalyzed by microcrystal Cu_4NC or Cu_8NC have undergone key intermediates A or B. In alkaline proton solutions, Int5 or Int12 could produce vinylboronate esters and key intermediates A or B by protonation, proving that the copper NCs were regenerated and could enter the next cycle. While the reaction was completed, N atoms reversibly coordinated to Cu atoms, restoring the original structure of the cluster. The flexibility

ensures the stability and reversibility of this copper cluster catalyst.

In-depth understanding of the mechanism of hydroboration catalyzed by microcrystal Cu_4NC or Cu_8NC was performed by *in situ* Fourier transform infrared (FT-IR) spectroscopy analysis and *in situ* Raman spectroscopy analysis (Supplementary Figures 82-86). As shown in Supplementary Figures 84 and 86, the top view of the three-dimensional surface of *in situ* the FT-IR spectroscopy could provide the selected individual spectra of key signals for hydroboration reaction process catalyzed by microcrystal Cu_4NC or Cu_8NC catalyst. The IR peak intensities of B-B bonds (1124 cm^{-1} for stretching vibration) and B-O bonds (1284 cm^{-1} for stretching vibration) in B_2Pin_2 decrease gradually as the reaction progress (*Nat Commun.* 2023, 14, 6957; *RSC Adv.* 2023, 13, 16907–16914; *Carbon.* 2013, 54, 208–214), illustrating the progressive consumption of the B_2Pin_2 substrate. Meanwhile, the new IR peaks appear at 1148 cm^{-1} , 1352 cm^{-1} and 1624 cm^{-1} , which can be assigned to the B-C bonds, B-O bonds and C=C bonds stretching vibrations of the vinylboronate ester target product, respectively (*Nat Commun.* 2023, 14, 6957; *RSC Adv.* 2023, 13, 16907–16914; *Carbon.* 2013, 54, 208–214). The IR peak intensities of the B-C bonds, B-O bonds and C=C bonds in target product grow gradually with reaction time. Compared the FT-IR spectroscopy of microcrystal Cu_4NC catalytic systems and microcrystal Cu_8NC catalytic systems, it further supported that microcrystal Cu_4NC is the stronger in catalytic activity than that of microcrystal Cu_8NC .

As shown in Supplementary Figure 82, the characteristic Raman peak at 690 cm^{-1} was attributed to the stretching vibration peak of Cu-N bond of microcrystal Cu_4NC (*J. Am. Chem. Soc.* 2022, 144, 21046–21055; *J. Mol. Struct.* 2018, 1165, 79–83). It was observed that the Raman peak intensities of Cu-N bond gradually decrease as the reaction proceeds, illustrating that Cu-N covalent bonds maybe dissociate during hydroboration reaction process. These changes in Raman spectra were attributed to the interaction between the substrates and the microcrystal Cu_4NC catalyst, which further support that the flexible Cu-N bonds of the copper NCs could dynamically dissociate from the copper cluster core to generate key intermediates.

In a word, combining experimental research and theoretical calculations, the mechanism of the reaction was deeply elucidated from multiple dimensions. These results have been added in the revised manuscript.

Supplementary Figure 16. ^1H NMR spectrum of microcrystal Cu_4NC in CD_3CN . No characteristic peaks of Cu_4NC are found in ^1H NMR spectrum, proving that microcrystal Cu_4NC is completely insoluble in MeCN solution.

Supplementary Figure 17. ^1H NMR spectrum of microcrystal Cu_4NC in D_2O . No characteristic peaks of Cu_4NC are found in ^1H NMR spectrum, proving that microcrystal Cu_4NC is completely insoluble in H_2O solution.

Supplementary Figure 71. DFT-calculated the binding energies of Cu-S bonds and Cu-N bonds in Cu₄NC and Cu₈NC (ΔG in kcal/mol).

Supplementary Figure 72. DFT-computed Gibbs free energy profile for formation of Intermediate A. The relative Gibbs energies and electronic energies (in parentheses) are given in kcal/mol.

The *in situ* Raman Experiment

Supplementary Figure 82. Time-dependent *in situ* Raman spectra of the hydroboration reaction catalyzed by microcrystal Cu₄NC catalyst (from bottom to top: 1min, 10min, 15min, 20min). Reaction conditions: Phenylacetylene 1 (0.2 mmol, 1.0 equiv.), B₂Pin₂ 2 (0.44 mmol, 1.0 equiv.), Cu₄NC catal. (0.004 mmol) and K₂CO₃ (0.44 mmol, 1.0 equiv.) were added to the mixture solvent (2.0 mL, MeCN/H₂O 4:1) under air atmosphere at room temperature and allowed to react for 1 h.

The *in situ* Fourier Transform Infrared (FT-IR) Experiment

Supplementary Figure 83. Time-dependent *in situ* FT-IR spectra of the hydroboration reaction catalyzed by microcrystal Cu₄NC catalyst. Reaction conditions: Phenylacetylene 1 (0.2 mmol, 1.0 equiv.), B₂Pin₂ 2 (0.44 mmol, 2.2 equiv.), Cu₄NC catal. (0.004 mmol, 2.0 mol%) and K₂CO₃ (0.44 mmol, 2.2 equiv.) were added to the mixture solvent (2.0 mL, MeCN/H₂O 4:1) under air atmosphere at room temperature and allowed to react for 1 h.

Supplementary Figure 84. The top view of the three-dimensional surface of the *in situ* FTIR spectra for hydroboration process catalyzed by microcrystal Cu_4NC catalyst.

Supplementary Figure 85. Time-dependent *in situ* FT-IR spectra of the hydroboration reaction catalyzed by microcrystal Cu_8NC catalyst. Reaction conditions: Phenylacetylene 1 (0.2 mmol, 1.0 equiv.), B_2Pin_2 2 (0.44 mmol, 2.2 equiv.), Cu_8NC catal. (0.004 mmol, 2.0 mol%) and K_2CO_3 (0.44 mmol, 2.2 equiv.) were added to the mixture solvent (2.0 mL, MeCN/ H_2O 4:1) under air atmosphere at room temperature and allowed to react for 1 h.

Supplementary Figure 86. The top view of the three-dimensional surface of the *in situ* FTIR spectra for hydroboration process catalyzed by microcrystal Cu_3NC catalyst.

In conclusion, this work needs significant improvement before publication, either through provision of more experiments or a much more careful write-up pointing out the speculative nature of many of the conclusions. Moreover, with these flaws answered it is at best a relatively prosaic step forward in this area of catalysis and would lack the broad appeal and impact for this journal.

Response: We thank the reviewer's helpful suggestion. As suggested, we have carefully revised and supplemented the article to resolve and correct flaws. We truly hope that the work is approved by you, providing a positive respond for publishing this manuscript on *Nature Communications*.

Responses to Reviewers

Reviewer #1:

Remarks: The authors have sufficiently addressed all the comments. I now recommend the article for publication.

Response: We thank the **Reviewer 1** for the positive comment very much.

Reviewer #2:

Remarks: The authors have made a great effort to improve the article by adding a series of experiments and explanations that have clarified most my main concerns. Although I believe that there are still some points that would need further clarification, my opinion is that the results in the new version are clearly sufficient for being published in Nature Communications:

Response: We appreciate **Reviewer 2** for the positive comments and the following constructive suggestions to improve our manuscript. We have addressed the following issues as suggested.

The utilization of English in the manuscript is still improvable, as incorrect conjugations are still present, and in some cases the utilization of mixed tenses (page 9: future and past tenses are mixed in the same paragraph).

The utilization of the word ‘microcrystal’ is utilized throughout the entire text. It should be ‘microcrystalline’, since it describes the copper clusters.

Response: Thank the **Reviewer 2** very much for the comment. We apologize for the inconvenience caused by the language. Following this suggestion, we have double-checked the manuscript and corrected the language. As suggested, we have polished the language of the manuscript through **Springer Nature Author Services**. And we have revised the expression of microcrystalline in the manuscript.

Page 5: By SCXRD analysis, the Cu-Cu distance is 2.662 Å and 2.663 Å in the Cu₄NC unit and Cu₈NC unit, respectively, which are slightly shorter than that of dicopper (I) μ-boryl complexes¹⁵. This claim is incorrect. The distance described by the authors corresponds to the tert-butoxide

dicopper species, instead of the boryl complexes. According to the reference, the boryl compounds possess copper-copper contacts of around 2.3 Å, much shorter than in the clusters described in this work. Therefore, this information is not valid for describing the proposed synergy between the copper centers.

Response: Thank the **Reviewer 2** very much for this important reminder. As **Reviewer 2** suggested, we have clarified this section in the revised manuscript. “SCXRD analysis revealed that the Cu-Cu distances were 2.662 Å and 2.663 Å in the Cu₄NC unit and Cu₈NC unit, respectively, which were slightly shorter than that of the tert-butoxide dicopper species. This shorter distance is beneficial for the formation of key intermediates A and B to achieve synergistic effects on dual catalytic sites.”

Pages 8 and 9: the authors repeat the reasons for the heterogeneous character of their catalyst, making it redundant.

Page 16: the dissociation of the Cu-N bond, as suggested by Raman spectroscopy results, is described several times in the text, making it redundant. Shortening of this part would be clearly beneficial. Additionally, the authors should provide the experimental conditions for the Raman measurements, particularly if they have been recorded in solution or in the solid state. This is particularly relevant considering that if the measurement is made in solution, the complexes detected are soluble, providing evidence of the homogeneous character of the catalyst. Moreover, complexes Cu₄NC or Cu₈NC are in fact the pre-catalyst, being A or B (more stable than the pre-catalysts) the true catalysts whose solubility is unknown.

Response: Thanks for the reviewer’s comment. As **Reviewer 2** suggested, the repeated descriptions on pages 8 and 9 regarding the reasons for the heterogeneous character of catalyst, have been simplified in the revised manuscript. Similarly, the repeated descriptions on page 16 regarding the dissociation of the Cu-N bond suggested by Raman spectroscopy results, have been simplified in the revised manuscript. The experimental conditions for the Raman measurements have been added to the revised Supplementary information (Supplementary Figure 82). And the *in situ* Raman spectra of hydroboration catalyzed by microcrystalline Cu₄NC were recorded in the solid state.

To further confirm that the key intermediate A or B is completely insoluble in the reaction solvent system (MeCN-H₂O), corresponding experiments and characterization tests were carried out. First, the hydroboration reaction catalyzed by microcrystalline Cu₄NC was conducted. After 5 minutes, the microcrystalline Cu₄NC was removed from the catalytic system. Continuous stirring of supernatant was further monitored by the *in situ* ¹H NMR at the 5th minute, 20th minute, 30th minute and 60th minute (Supplementary Figure 98), showing that no further conversion was observed within this solution. Next, the other control experiment was carried out. When the reaction was completed, the microcrystalline Cu₄NC was removed from the catalytic system. The additional aliquot of the substrates was added to the supernatant, which was further monitored by the *in situ* ¹H NMR at the 1th minute, 20th minute, 30th minute and 60th minute (Supplementary Figure 99). The result showed that no conversion of the added material was observed. Finally, the supernatants were characterized by inductively coupled plasma mass spectrometric (ICP-MS) measurement, showing that no Cu⁺ ions were found in the reaction supernatant, suggesting that Cu⁺ ions were not leached from the metal cluster catalyst into solution over the course of the reaction. These results suggested that key intermediate A is completely insoluble in the reaction solvent system (MeCN-H₂O).

Supplementary Figure 82. The test method of the Raman spectra.

Supplementary Figure 98. Time-dependent *in situ* ^1H NMR spectra of the hydroboration reaction. Reaction conditions: Phenylacetylene 1 (1.0 equiv.), B_2Pin_2 2 (2.2 equiv.), Cu_4NC catal. (10% mol) and K_2CO_3 (2.2 equiv.) were added to the mixture solvent (2.0 mL, $\text{CD}_3\text{CN}-\text{H}_2\text{O}$) under air atmosphere at room temperature and allowed to react for 5 min. The catalyst was separated from the supernatant of the reaction system by filtration. The supernatant was stirred for 1 h.

Supplementary Figure 99. Time-dependent *in situ* ^1H NMR spectra of the hydroboration reaction. Reaction conditions: Phenylacetylene 1 (1.0 equiv.), B_2Pin_2 2 (2.2 equiv.), Cu_4NC catal. (10% mol) and K_2CO_3 (2.2 equiv.) were added to the mixture solvent (2.0 mL, $\text{CD}_3\text{CN}-\text{H}_2\text{O}$) under air atmosphere at room temperature and allowed to react for 1 h. The catalyst was separated from the supernatant of the reaction system by filtration. Phenylacetylene 1 (1.0 equiv.), B_2Pin_2 2 (2.2 equiv.) and K_2CO_3 (2.2 equiv.) were added to the supernatant. The reaction was stirred for 1 h.

Page 17: again, a 1.8 kcal mol⁻¹ energy barrier difference between the Cu₄ and Cu₈ mechanism is not a robust result to claim the following: ‘These results clearly illustrated that the catalytic activity of Cu₄NC is better than that of Cu₈NC in the hydroboration reaction’. I don’t understand why the authors do not include the benchmark results, since the energy barrier difference can be as high as 12 kcal mol⁻¹, supporting their observations. The benchmark results should be included in the SI file, and their data discussed in the main text, in order to provide theoretical support to the experimental observations.

Response: Thank the **Reviewer 2** very much for the comment. As suggested by **Reviewer 2**, the benchmark results have been added into the revised Supplementary information (Supplementary Table 2). The benchmark results have been discussed in the revised manuscript, which could provide theoretical support to the experimental observations.

Supporting Information, Page 9: Figures S11 and S12 should include the description of parts ‘a’ and ‘b’. Figures S91 and S92: what do the authors mean by dosing projects?

Response: Thank the **Reviewer 2** very much for this important reminder. As **Reviewer 2** suggested, the description of parts ‘a’ and ‘b’ in Figures S11 and S12, has been added into the revised Supplementary information.

The repeated dosing experiments were conducted to further demonstrate that microcrystalline Cu₄NC has excellent catalytic activity and stability, which was added into the revised Supplementary information.

Reviewer #3:

Remarks: I co-reviewed this manuscript with one of the reviewers who provided the listed reports. This is part of the Nature Communications initiative to facilitate training in peer review and to provide appropriate recognition for Early Career Researchers who co-review manuscripts.

Response: Thank the **reviewer 3** so much for the comment. Thank you very much for helping us improve the manuscript.

Reviewer #4:

Remarks: The authors have clearly done much to improve the work. The minor comments about claims in the manuscript are, generally speaking, covered by their revisions.

Response: We thank the **Reviewer 4** for the positive comment very much.

The nature of these systems as heterogeneous catalysts is now clear. This does rebut a number of the queries raised (for example, any sort of in situ monitoring by NMR is clearly therefore precluded). The Raman monitoring does provide some alternative to this, and provides evidence for the loss of the Cu-N bonds over the course of the reaction. Clearly these raman data do therefore support the formation of species such as A.

1) It would be nice to show the Raman spectrum of Cu₄NC + K₂CO₃ in the absence of catalyst to provide further evidence of this.

2) Does the Raman spectrum of the recovered material show a complete return of these peaks?

Response: We appreciate **Reviewer 4** for the positive comments and the following constructive suggestions to improve our manuscript. We have addressed the following issues as suggested. The *in situ* Raman spectra of Cu₄NC + K₂CO₃ was monitored. We observed that the Raman peak intensities of the Cu-N bonds also gradually decreased as the reaction proceeded (Supplementary Figure 84), illustrating that the Cu-N bonds may have dissociated under basic condition. These Raman data further support the formation of the key intermediate A. As suggested by **Reviewer 4**, the Raman spectra of the recovered material was further characterized by the Raman spectroscopy

(Supplementary Figure 85). The peak intensity of the Raman spectra of the recovered material displays good agreement with that of the new catalyst material (Supplementary Figure 85), proving that the Raman spectra of the recovered material show a complete return of these peaks.

Supplementary Figure 84. Time-dependent *in situ* Raman spectra of the $\text{Cu}_4\text{NC} + \text{K}_2\text{CO}_3$ (from bottom to top: 1 min, 10 min, 15 min, 20 min, 30 min). Reaction conditions: the mixture solution ($\text{MeCN}/\text{H}_2\text{O}$) of K_2CO_3 was dropped onto microcrystalline Cu_4NC catalyst to react under air atmosphere at room temperature.

Supplementary Figure 85. The Raman spectra of the microcrystalline Cu_4NC before and after catalysis. The new $\text{Cu}_4\text{NC-1}$ and the new $\text{Cu}_4\text{NC-2}$ represent the Raman spectra of microcrystalline Cu_4NC catalysts prepared from different batches. The recovered $\text{Cu}_4\text{NC-1}$ and the recovered $\text{Cu}_4\text{NC-2}$ represent the Raman spectra of microcrystalline Cu_4NC catalysts recovered from different batches.

The rationale for using DFT for this work is clearly considered on the authors part, although I do feel the authors are not fully considering the possibilities provided by a number of the Cu_4NC or Cu_8NC compounds being in close proximity on the surface of the insoluble material. Modelling a single molecular fragment of an insoluble material isn't the only alternative to taking a full heterogeneous approach, and the authors are in possession of understanding about the packing of the Cu_4NC and Cu_8NC compounds that would allow them to generate appropriate model systems to consider the possibility of more than a single cluster being involved in the reaction.

Response: We sincerely thank the reviewer for their insightful comments on our application of Density Functional Theory (DFT) to study heterogeneous reactions using a cluster approach. We acknowledge and strongly agree on the importance of considering the packing of Cu clusters in

this context, and we greatly appreciate your valuable insights.

To address this, we calculated the rate-determining steps for the Cu_4NC monomer, Cu_4NC dimer, and Cu_4NC trimer to assess the influence of packing. Our DFT calculations indicated that the barriers for the Cu_4NC monomer (22.9 kcal/mol), Cu_4NC dimer (20.6 kcal/mol), and Cu_4NC trimer (22.0 kcal/mol) are quite close (Supplementary Table 3). Furthermore, the transition states for these clusters are very similar (Supplementary Table 3), suggesting that the cluster approach is indeed feasible. We believe that the critical influence of the active sites is the primary reason for this consistency, while the packing of the clusters is relatively less important. And our current focus on single clusters allows us to effectively model the active centers involved in the reaction.

Additionally, by treating these clusters as molecular systems, we can apply more accurate hybrid functionals instead of relying solely on the Generalized Gradient Approximation (GGA) with the Perdew-Burke-Ernzerhof (PBE) functional. This not only enhances the precision of our simulations but also provides greater computational efficiency, resulting in a more detailed understanding of the catalytic processes at the atomic level.

Thank you once again for your valuable feedback.

Supplementary Table 3. Theoretical calculations regarding rate-determining intermediates and the rate-determining transition states for the hydroboration catalyzed by Cu_4NC monomer, dimer and trimer.

States	The barrier (kcal/mol)	Structures of TS ₄₋₅
Cu_4NC monomer	22.9 kcal/mol	 Cu_4NC monomer C-B:1.88 Å Cu-C:2.05 Å Cu1-B:2.20 Å Cu2-B:2.23 Å
Cu_4NC dimer	20.6 kcal/mol	 Cu_4NC dimer C-B:1.91 Å Cu-C:2.05 Å Cu1-B:2.21 Å Cu2-B:2.21 Å

Cu ₄ NC trimer	22.0 kcal/mol	 Cu₄NC trimer C-B:1.89 Å Cu-C:2.04 Å Cu1-B:2.24 Å Cu2-B:2.26 Å
---------------	---

In terms of validating the Cu₄NC and Cu₈NC systems as the catalytic vector (and proving that the reaction is truly homogeneous), the authors provide evidence of insolubility of Cu₄NC, provide evidence of recovery without chemical change, provide evidence that the Cu(I) does not pass into solution, and provide evidence that Cu₄NC removed from the reaction at various time points shows consistent characteristics. What I can't see evidence of is evidence that shows that the supernatant lacks any catalytic ability. This should be easy to do by

3) removing the supernatant at a point prior to completion of the reaction and ensuring that no further conversion is observed within this solution.

4) removing the supernatant at the end of the reaction, and adding an additional aliquot of the substrates, and ensuring no conversion of the added material is observed.

At this stage, the balance of probabilities is that Cu₄CN/Cu₈CN are the catalytic vector, but the formation of a soluble, catalytically competent, copper-free species under the reaction conditions still exists. It should be precluded.

Response: We appreciate **Reviewer 4** for the positive comments and the following constructive suggestions to improve our manuscript. We have addressed the following issues as suggested. Corresponding experiments and characterization tests were carried out. First, the hydroboration reaction catalyzed by microcrystalline Cu₄NC was conducted. After 5 minutes, the microcrystalline Cu₄NC was removed from the catalytic system. Continuous stirring of supernatant was further monitored by the *in situ* ¹H NMR at the 5th minute, 20th minute, 30th minute and 60th minute (Supplementary Figure 98), showing that no further conversion was observed within this solution. Second, the other control experiment was carried out. When the reaction was completed, the microcrystalline Cu₄NC was removed from the catalytic system. The additional aliquot of the substrates was added to the supernatant, which was further monitored by the *in situ* ¹H NMR at the 1th minute, 20th minute, 30th minute and 60th minute (Supplementary Figure 99). The result showed that no conversion of the added material was observed. All of these observations and experimental results suggested that the supernatant lacks any catalytic ability. The possibility of forming a soluble, catalytically competent, and copper-free species under reaction conditions has been precluded.

Supplementary Figure 98. Time-dependent *in situ* ^1H NMR spectra of the hydroboration reaction. Reaction conditions: Phenylacetylene 1 (1.0 equiv.), B_2Pin_2 2 (2.2 equiv.), Cu_4NC catal. (10% mol) and K_2CO_3 (2.2 equiv.) were added to the mixture solvent (2.0 mL, $\text{CD}_3\text{CN-H}_2\text{O}$) under air atmosphere at room temperature and allowed to react for 5 min. The catalyst was separated from the supernatant of the reaction system by filtration. The supernatant was stirred for 1 h.

Supplementary Figure 99. Time-dependent *in situ* ^1H NMR spectra of the hydroboration reaction. Reaction conditions: Phenylacetylene 1 (1.0 equiv.), B_2Pin_2 2 (2.2 equiv.), Cu_4NC catal. (10% mol) and K_2CO_3 (2.2 equiv.) were added to the mixture solvent (2.0 mL, $\text{CD}_3\text{CN-H}_2\text{O}$) under air atmosphere at room temperature and allowed to react for 1 h. The catalyst was separated from the supernatant of the reaction system by filtration. Phenylacetylene 1 (1.0 equiv.), B_2Pin_2 2 (2.2 equiv.) and K_2CO_3 (2.2 equiv.) were added to the supernatant. The reaction was stirred for 1 h.

Responses to Reviewers

Reviewer #2:

Remarks: The authors have done an excellent job addressing all of my concerns, and I believe the results presented here are therefore suitable for publication in Nat. Commun.

Just a minor typo, in some places (ESI) the Raman spectroscopy appears as "Roman" spectroscopy.

Response: We thank the **Reviewer 2** for the positive comment very much. And we have revised the expression of Raman in the Supplementary information as suggested.

Reviewer #3:

Remarks: I co-reviewed this manuscript with one of the reviewers who provided the listed reports. This is part of the Nature Communications initiative to facilitate training in peer review and to provide appropriate recognition for Early Career Researchers who co-review manuscripts.

Response: Thank the **reviewer 3** so much for the comment. Thank you very much for helping us improve the manuscript.

Reviewer #4:

Remarks: The authors have now answered all of my comments. I have 2 minor points that I suggest revising, but do not need to see the manuscript again.

Fig. 1b - "resonant" persists where "resonance" should be used.

Fig. 5b - this is still very hard to parse.

Response: We thank the **Reviewer 4** for the positive comment very much. As suggested by the **Reviewer 4**, we have revised the expression of resonance in Fig. 1b and we have revised Fig. 5b in the main text.